# Breast Cancer: A Molecularly Heterogenous Disease Needing Subtype-Specific Treatments

**DOI:** 10.3390/medsci8010018

**Published:** 2020-03-23

**Authors:** Ugo Testa, Germana Castelli, Elvira Pelosi

**Affiliations:** Department of Oncology, Istituto Superiore di Sanità, Regina Elena 299, 00161 Rome, Italy; germana.castelli@iss.it (G.C.); elvira.pelosi@iss.it (E.P.)

**Keywords:** cancer, breast cancer, cancer genomics, biomarkers, target therapy

## Abstract

Breast cancer is the most commonly occurring cancer in women. There were over two-million new cases in world in 2018. It is the second leading cause of death from cancer in western countries. At the molecular level, breast cancer is a heterogeneous disease, which is characterized by high genomic instability evidenced by somatic gene mutations, copy number alterations, and chromosome structural rearrangements. The genomic instability is caused by defects in DNA damage repair, transcription, DNA replication, telomere maintenance and mitotic chromosome segregation. According to molecular features, breast cancers are subdivided in subtypes, according to activation of hormone receptors (estrogen receptor and progesterone receptor), of human epidermal growth factors receptor 2 (HER2), and or BRCA mutations. In-depth analyses of the molecular features of primary and metastatic breast cancer have shown the great heterogeneity of genetic alterations and their clonal evolution during disease development. These studies have contributed to identify a repertoire of numerous disease-causing genes that are altered through different mutational processes. While early-stage breast cancer is a curable disease in about 70% of patients, advanced breast cancer is largely incurable. However, molecular studies have contributed to develop new therapeutic approaches targeting HER2, CDK4/6, PI3K, or involving poly(ADP-ribose) polymerase inhibitors for BRCA mutation carriers and immunotherapy.

## 1. Introduction

Breast cancer is a dramatically important health problem and it is one of the main causes of women death. Using the cancer registers of 187 countries it was estimated that the global breast cancer incidence increased from 641,000 cases in 1980 to 1,643,000 cases in 2010, with an annual increase rate of approximately 3% [1]. Breast cancer killed, in these countries, more than 400,000 women in 2010 [1]. These numbers indicate that there is an absolute need to improve our understanding of the cellular and molecular basis of this tumor, to improve its prevention and therapy. In 2019, about 268,000 new cases of invasive breast cancer are expected to be diagnosed in USA, along with about 63,000 new cases of non-invasive breast cancer; approximately 1% of the breast cancers diagnosed in women is expected to be diagnosed in men [2,3]. Breast cancer rates in USA began to decrease in the year 2000, probably in relation with the reduced use of hormone replacement therapy by women [2]. It was estimated that one in eight USA women will display invasive breast cancer over the course of their lifetime [2,3]. A women’s risk of developing breast cancer nearly doubles if she has first-degree relatives who have been diagnosed with breast cancer [2]. About 5–10% of breast cancers can be linked to gene mutations inherited from one’s mother or father, such as BRCA1 or BRCA2 mutations [2]. The peak of breast cancer incidence occurs for women between the ages of 65 and 80 years. However, invasive breast cancer incidence is also frequently observed in young women (<50 years) and its incidence is increasing 0.2% per year [3]. Considerable progresses have been made over the past 50 years in the evaluation and treatment of patients with breast cancer, leading to a nearly 40% decrease in mortality of this disease (due to prevention strategies and the improvement of medical treatment) [2].

## 2. Mammary Stem Cells

Breast epithelium forms a ductal network that is embedded into an adipose tissue, connecting the nipple through numerous collecting ducts to a complex system of mammary lobes, which are the structures that are responsible for milk production during pregnancy and lactation. The mammary gland undergoes very extensive changes during development after birth, consisting in its glandular expansion during puberty to full tubule-alveolar differentiation during lactation. Two main cellular lineages are present within the mammary gland: (a) luminal cells, forming the internal layer of ducts and alveoli: these cells express hormone receptors (HR, ER/PR) and they are characterized by the expression of a set of cytokeratins, including CK8, 18 and 19; (b) basal myoepithelial cells, capable of contractile activity and localized between luminal cells and the basement membrane: these cells are characterized by the expression of CK5 and 14 and of smooth muscle actin.

Early studies showed that any portion of an intact murine mammary gland containing epithelium could generate an entire mammary epithelial tree on transplantation into an epithelium-free mammary fat pad. This capacity was ascribed to the presence in the mammary gland of mammary stem cells that are thought to be self-renewing and to reside at the apex of a cellular hierarchy. The development of cell isolation procedures, as well as of in vivo assay into immunodeficient mice (xenografting of human breast cells into the cleared fat pad or under the renal capsule), have greatly contributed to the characterization of human mammary stem cells. These studies have led to the isolation of cells exhibiting the properties of mammary stem cells while using a combination of various cell-surface markers (CD44, CD24, CD29, CD49f, and EpCAM) [4]. The human breast is formed, starting from a flask-like epithelial structure known as the mammary primordium (this structure develops at week 14 of gestation): central and peripheral primary bud cells can be identified at this stage. At a later stage of development, solid cords of epithelial cells (secondary epithelial outgrowths) migrate in the surrounding mesenchyme, starting from growing primary bud. At these stages of development, fetal mammary stem cells control the growth of mammary gland, displaying properties that are different from those observed for adult mammary stem cells. High CD44 and CD49f expression characterizes a basal epithelial compartment containing all the fetal mammary stem cell activity [5]. Interestingly, during late embryogenesis, fetal mammary rudiments are highly enriched in stem cells [5]. Gene expression, transplantation, and in vitro studies predicted the existence of autocrine and paracrine mechanisms in these fetal mammary stem cells, involving ERB and FGF signaling pathways [5]. The gene expression profiles from mammary stem cells and associated fetal stromal cells displayed a significant similarity with the basal-like and HER2^+^ breast cancer subtypes [5]. Analysis of normal human breast tissue shows a hierarchical organization, involving non-clonogenic luminal cells, and differentiated (EpCAM^+^CD49f^+^ALDH^-^) and undifferentiated (EpCAM^+^CD49f^+^ALDH^+^) luminal progenitors; all of the progenitor populations are highly plastic and can generate all mammary cell types [6].

In mouse, luminal mammary cells can be subdivided into ER^+^ and ER^-^. The cells represent two distinct lineages and their development, homeostasis, and regeneration is distinct and is supported by two different stem cell populations in the adult mammary gland [7,8].

Two different models have been proposed to explain the relationship and the development of various lineages existing in the mammary epithelium. One model assumes that different ductal and lobular progenitors exist, both of which are capable of giving rise to both basal and luminal cells. A second model proposes that early during development the basal and the luminal cell lineages are completely separated.

At variance of mammary fetal progenitors that are multipotent, two separate populations of unipotent progenitor cells maintain the adult mammary gland [9]. In fact, van Keymeulen et al. have used an inducible genetic lineage strategy, allowing to explore the multi- or uni-potency of mammary progenitors during various stages of development: embryogenesis, after birth, during puberty, and lactation. During embryonic life all of the mammary gland cells were derived from a CK14^+^ multipotent progenitor cell [9]. In contrast, the studies carried out on mammary glands after birth provided evidence regarding the existence of facultative luminal (CK8+) and basal/myoepithelial (CK14+) unipotent progenitors. The luminal progenitors are able to generate mature ductal luminal cells or mature alveolar (milk-secreting) luminal cells; the unipotent basal/myoepithelial progenitors generate myoepithelial cells. However, under appropriate conditions, the adult CD24^+^CD29^+(high)^ fraction was able to revert to a multipotent embryonic-like activity, generating both luminal and basal cells [9]. Therefore, it was concluded that, in normal adult mammary gland, luminal and basal/myoepithelial lineages both contain long-lived unipotent stem cells displaying extensive renewing capacities, as shown by their ability to clonally expand during morphogenesis and to undergo massive expansion during pregnancy.

Other studies have shown that adult mammary glands contain a Wnt-responsive cell population that is enriched for stem cells. Using a combination of cell culture and in vivo transplantation experiments, it was shown that Wnt proteins act as important self-renewal factors for mammary gland stem cells [10]. In addition, it was shown that Axin 2^+^ cells (a target of Wnt signaling) have the properties of mammary stem cells [10]. Taking advantage of the identification of Axin2^+^ cells as a mammary stem cell responsive to Wnt, cell tracking experiments have been carried out to define the staminal potential of these cells during ontogenic development [11]. In the embryo, the Axin2^+^ cells mark the luminal lineage, while after birth these cells become exclusively committed to the basal cell fate [11]; later, in adult life Axin2 marks cells corresponding to multipotent progenitors generating both mammary lineages and unipotent progenitors, generating both mammary lineages and unipotent progenitors, which are committed to each of the two lineages [11]. These observations indicate that dynamic changes of stem/progenitor cells occur in the mammary gland during development [11]. In line with these observations, the protein C receptor, a Wnt target in the mammary gland, marks a unique population of mouse multipotent mammary stem cells [12].

Some studies have provided evidence regarding the existence of bipotent progenitors in adult mammary tissues, but other investigators did not confirm these findings. Thus, through clonal cell-fate mapping studies, evidence was provided for the existence of bipotent mammary stem cells, as well as of distinct long-lived progenitor cells. The cellular dynamics of these cellular elements at various developmental stages support a model in which both stem and progenitor cells drive breast morphogenesis during puberty, whereas bipotent progenitors drive the homeostasis of adult mammary gland [13]. Cai and coworkers identified a quiescent mammary epithelial cell population expressing high levels of Bcl11b (B-Cell Lymphoma/Leukemia 11B, a zinc finger protein), which were located at the interface between luminal and basal cells [14]. The loss of Bcl11b leads to an exhaustion of ductal epithelium and the loss of epithelial cell regenerative capacity; gain- and loss-of-function studies indicate that Bcl11b induces cells to enter the G_0_ phase [14]. LGR5 is only a marker of bipotent mammary progenitors in embryonic cells, while in adult mammary LGR5-positive cells are restricted to the myoepithelial lineage [15].

Through asymmetric divisions, these stem cells generate a more differentiated cell progeny unable to self-renew. It was commonly believed that the differentiated cell progeny was unable to revert to a stem-like condition. However, experiments carried out on cultures of mammary epithelial cells have reported an unexpected plasticity, in that differentiated mammary epithelial cells are able to convert a stem-like state, according the stochastic process [16]. This conversion occurs in both normal and transformed mammary epithelial cell populations [16].

Recent studies have shown that a hierarchy of mammary stem/progenitors exists within the mammary epithelium and their basic biology, survival, and proliferation is controlled by signals that are generated both locally and systemically. Among these various signals, hormone signaling plays a key role. The development of mammary gland is controlled by the concerted action of both systemic hormones and growth factors. In this context, a key role is played by steroid hormones estrogen and progesterone that are mitogens for mammary epithelial cells. The effects of these two hormones are mediated through two specific nuclear receptors, the estrogen receptor (ER) and the progesterone receptor (PR), respectively. Studies that have been carried out on developing mouse mammary tissue have clarified the physiological role of these two hormone receptors, with the estrogen receptor being involved in the regulation of duct formation and morphogenesis and the progesterone receptor in the regulation of duct branch formation. Furthermore, PR plays a key role during pregnancy, allowing for tertiary side branching and alveologenesis. The effects of estrogen and progesterone on mammary gland are largely mediated through paracrine mechanisms. Recent studies have shown that estrogen and progesterone markedly affect mammary stem cell function. Thus, it was shown that (i) ovarectomy markedly reduces the mammary stem cell pool; (ii) mammary stem cell activity increases in mice that were treated with estrogen plus progesterone; (iii) mammary stem cell pool markedly increases during maximal progesterone levels at the luteal diestrus phase of the mouse; (iv) treatment with aromatase inhibitors decreases the mammary stem cell pool; and, (v) pregnancy determines a marked increase in mammary stem cell pool, through a mechanism, which is mediated by progesterone and involving a paracrine mediator, RANK Ligand [17,18]. A more recent study has defined the role of RANK as a key paracrine mediator of the effects of progesterone on mammary stem cells. In fact, it was shown that progesterone administration markedly increases the levels of RANK Ligand (Receptor Activator of NF-kB Ligand) [19]. The genetic inactivation of the RANK L receptor impairs the effects of progesterone on mammary stem cells [19]. Importantly, the inhibition of the RANK/RANKL system in mammary gland markedly decreases the incidence and delays the onset of progesterone-driven mammary cancer [19].

The effect of autocrine and paracrine signals on mammary stem cell fate is ultimately mediated by a fine control of gene expression, which is mainly controlled through the coordinated effects of a network of transcription factors. In this context, a regulatory network orchestrated by the transcription factors Slug and Sox9, plays a key role in the determination of the mammary stem cell fate [20]. The inhibition of either Slug or Sox9 determines a block of mammary stem cell activity [20]. On the other hand, the enforced transient expression of exogenous Slug and Sox9 into differentiated luminal cells determines their conversion to long-term repopulating mammary cells [20].

The ErbB family of receptor tyrosine kinase and their ligands are important regulators of mammary gland development. This family consists of four members: HER1/ErbB1/EGFR; HER2/ErbB2/Neu; HER3/ErbB3; and, HER4/ErbB4. ErB TRKs are required for normal breast development, being particularly important being the role of ErbB2 and ErbB3. In fact, the loss of ErbB2 in mammary epithelium delays ductal elongation and disorganizes terminal end buds of the mammary gland; in contrast, loss of ErbB3, whose expression is highest in luminal mammary cells and lowest in basal stem cells, impaired AKT and MAPK kinase signaling in luminal cells, with the consequent loss of luminal cell proliferation and survival: interestingly, the loss of ErbB3 concomitantly induced an expansion of basal cells, thus suggesting that the normal function of this receptor tyrosine kinase is required for maintaining the balance between luminal and basal breast epithelium [21].

A very important problem in the context of the study of normal and malignant mammary stem cells is related to the development of suitable and reproducible in vivo assays to evaluate mammary stemness. Two types of assays have been proposed in this context. Both of these assays were based on the evaluation of the capacity to regenerate mammary gland structures from mammary human epithelial cells transplanted into highly immunodeficient mice. One of the assays is based on the evaluation of the capacity to colonize the precleared mammary fat pad of immunodeficient mice to create a suitable environment before the transplantation in these sites of human mammary epithelial cells [22,23]. An alternative strategy has been developed that is based on suspending human mammary epithelial cells, together with irradiated human fibroblasts in a collagen gel, which is subsequently implanted under the kidney capsule of estrogen- and progesterone-treated NOD/SCID mice [24].

Advances in next generation sequencing and handling procedures of single cells allowed for the possibility of exploring cellular heterogeneity at the single cell level and reconstruct lineage hierarchies while using single-cell RNA sequencing. Single-cell transcriptomic analysis of stem cell state allows for defining ontogenic stages and lineage specification programs occurring in early murine mammary gland development [25]. This study showed that: (i) individual mammary stem cells co-express genes associated with differentiated mammary lineages; (ii) mammary stem cells constitute a single distribution of heterogeneous transcriptional states, without discrete subpopulations [25]. These findings suggest that stem cell capacity is distributed across heterogeneous cell profiles [25]. Nguyen et al. reached the same conclusions through the study of single cell transcriptomic in human mammary epithelial cell populations, including one basal and two luminal cell types, identified as secretory L1 and hormone-responsive L2-type cells [26]. Temporal reconstruction of differentiation trajectories indicates the existence of one continuous lineage hierarchy that connects the basal lineage to the two differentiated luminal cell branches [26].

Other unicellular sequencing studies have characterized multipotent embryonic mammary progenitors, showing that these cells express a unique hybrid basal and luminal signature and the factors that are associated with the different lineages [27]. Early during embryonic development, embryonic multipotent mammary cells become lineage-restricted [28]. Finally, single-cell landscape in human mammary cells revealed the existence of bipotent-like cells that are associated with breast cancer risk and outcome [29].

## 3. Molecular Abnormalities of Invasive Breast Cancer

Breast cancer is a highly heterogeneous disease for its histology, epidemiology, and molecular properties. Six molecular subtypes of breast cancer have been identified according to their gene expression profiles and their identification and classification was of fundamental importance for our understanding of tumor genesis and progression: normal breast-like, luminal A and B, basal-like, claudin-low, and HER2/ERB2 overexpressing. The origin of luminal A and B tumors seems to be the mammary duct luminal epithelium, with consistent hormone receptor expression. Basal-like cancers form a heterogeneous group of breast cancers, which probably arise from progenitor cells different from those involved in other breast cancers. HER2/ERB2 overexpressing breast cancers represent a group of aggressive breast cancers that are associated with poor prognosis. Finally, claudin-low are a peculiar group of aggressive breast cancers that are characterized by negative expression of ER, PR, and HER2 (triple-negative), and by the acquisition of mesenchymal/sarcomatoid and/or squamous metaplasia of malignant breast epithelium. This classification also reflects a different metastatic potential of these various breast cancer types. Bone is the most frequent metastatic site for all breast cancer subtypes, with the exception of basal-like. Luminal A tumors are those with the lowest tendency to metastasize; luminal/HER2 and HER2-positive breast cancers were more metastatic than luminal A cancers, particularly at the level of brain, liver, and lung metastases; the basal-like tumors displayed a higher tendency to metastasize at the level of brain and lung, but a lower tendency at the level of liver and bone; finally, triple-negative tumors metastasize at the level of all sites [30].

Other investigators have grouped and classified breast cancers according to the expression of the important functional markers estrogen receptor (ER), progesterone receptor (PR), and HER2, allowing the identification of tumor subtypes with different outcomes [31]. These markers may be also used to additionally characterize the molecular subtypes: luminal A subtype is defined as ER^+^ and/or PR^+^, HER2^-^; luminal B subtype is defined as ER^+^ and/or PR^+^, HER2^+^; basal-like subtype is defined as ER^-^, PR^-^, HER2^-^; and, HER2 subtype is defined as ER^-^, PR^-^, HER2^+^. Thus, luminal A breast cancers are highly ER^+^ and PR^+^, HER2^-^, have usually low proliferative rates and a low Ki67 index, have a NST (no special type), tubular cribiform or classic lobular histology and have a good prognosis. Luminal B breast cancers can be subdivided into HER2^-^ and HER2^+^: the HER2^-^ tumors are usually ER^+^ (lower expression than in luminal A tumors), have high proliferation rates, a high Ki67 index, a micropapillary and lobular pleimorphic histology, and exhibit an intermediate perognosis; luminal B, HER2^+^ breast cancers are usually ER^+^, PR^+^, have a high Ki67 index and an intermediate prognosis. HER2-enriched non-luminal breast cancers have NST histology, a high Ki67 index, an aggressive tumor phenotype, and an intermediate prognosis. Triple-negative breast cancers (TNBCs) largely correspond to basal-like and claudin-low subtypes, have a NST histology or a special histology (metaplastic, adenoid cystic, medullary-like), a high Ki67 index, and a poor prognosis.

These three clinically adopted markers for the classification of primary breast cancers are used to help decisions regarding therapy in the metastatic setting. The ER, PR, and HER status often changes during disease progression; in fact, a recent study that was carried out on a large cohort of patients estimated that at relapse 32%, 41%, and 15% of patients change their ER, PR, and HER2 status, respectively [32]. Importantly, women with ER-positive tumors that changed to ER-negative tumors had a significantly 48% increased risk of death when compared with women with stable ER-positive tumors [32].

The treatment of breast cancer during the last years was based on many of the classification criteria that were previously mentioned. The real impact of some of these parameters was now analyzed through the meta-analysis of many clinical trials reporting data in large numbers of patients, thus allowing for reaching some important conclusions. Concerning the hormonal status, the largely more important parameter is the presence of estrogen receptor on tumor cells. In ER-positive breast cancers, the allocation of five-years of treatment with the estrogen inhibitor tamoxifen significantly reduced (of about a third) disease recurrence and disease-related mortality [33]. On the other hand, the meta-analysis of radiotherapy studies that were carried out on more than 10000 patients showed that radiotherapy to the conserved breast halves the rate at which disease recurs and reduces the breast cancer death by about a sixth [34]. However, this proportional benefit varies considerably between patients with different disease characteristics [34]. The meta-analysis of the survival impact of various neo-adjuvant chemotherapy regimens provided evidence that, generally, chemotherapy reduces of about one-third breast cancer mortality as compared to non-chemotherapy and that anthracycline-based regimens are more efficacious than taxane-based regimens or cyclophosphamide-based regimens. Importantly, the risk reductions that are induced by chemotherapy were affected little by age, nodal status, tumor diameter or differentiation, estrogen receptor status, or tamoxifen use [35].

Comparative genomic array studies allowed for identifying three main types of genomic alterations in breast cancer: (a) tumors with few genetic rearrangements (mainly characterized by gain of chromosome 1q and/or loss of 16q); (b) tumors with complex genetic alterations; and, (c) tumors with packed, high-level implicons. Advances in genome sequencing of breast cancers allowed for identifying the full spectrum of mutations present in a small number of breast cancers. One of the large-scale sequencing studies, which was carried out by Stephens and coworkers [36], provided evidence about the existence of different types of alterations, according to three different patterns: (a) interchromosomal translocations with copy number alterations involving large DNA fragments or whole chromosome arms; (b) complex, interchromosomal translocations involving shorter regions with high-level amplifications; and (c) small, interchromosomal segmental alterations, such as deletions, duplications, and/or inversions, called “mutator phenotype”.

The great surprise deriving from the detailed sequencing studies of breast cancer was the observation that individual tumors were unique, each harboring a large collection of individual, “private” mutations that collectively characterized its genome. However, a recent large-scale screening of DNA mutations that occur in breast cancer identified more than 1700 different genic mutations, but only three of these genes were mutated at high frequencies: *PI3KCA* (43%), *TP53* (15%), and *MAP3K1* (9%) [37]. The stratification of these patients according to expression subtypes, showed that *TP53* mutation is more frequent in basal-like and HER2-enriched disease, while the *PI3KCA* mutation is more frequent among luminal A tumors [37]. The occurrence of *PIK3CA* mutations was explored both in in situ and in invasive breast cancers and the conclusion was reached that its frequency is similar in these two tumors, thus supporting the concept that it is more likely to play a role in breast tumor initiation than in invasive progression [38].

Another recent study confirmed these findings; in fact, Banerji and coworkers reported five genes to be frequently mutated in breast cancer: *TP53* and *PI3KCA*, both in 27% of cases; *AKT1* in 6% of cases; *MAPK1* in 6% of cases; and, *GATA3* in about 4% of patients (both *AKT1* and *PI3KCA* mutations activate the PI3K pathway and are mutually exclusive) [39]. Interestingly, in this study, additional recurrent abnormalities occurring in breast cancer have been discovered, including mutations of the transcription factor *CBFB* (core-binding-factor beta subunit), which is associated with hemizygous deletions of one allele of *RUNX1* (4% of cases) and homozygous deletions of *RUNX1* (about 2% of cases); a balanced translocation between the *MAGI3* and the *AKT3* genes leads to the formation of the *MAGI3-AKT3* fusion protein, exhibiting constitutive activation of the AKT kinase (about 3% of all cases and more frequent in triple-negative breast cancers) [39]. Ellis and colleagues have also explored the occurrence of mutations in estrogen receptor-positive breast cancers. They have included in their analysis the genomes of tumors that were derived from patients participating in pre-operative clinical evaluation of their response to aromatase inhibitors. This analysis confirmed the frequent mutation of genes, such as *PIK3CA, TP53, GATA3, MAP3K1, RB1,* and *MLL3*, which are known to be mutated in breast cancer and led also to the discovery of rare mutations involving genes, such as *RUNX1, MYH9, TBX3,* and *CBFB*. Particularly, *PIK3CA* mutations were observed in 16% of cases, *MAP3K1* mutations in 15.5% of cases, and *GATA3* mutations in about 9% of cases. These patients were subdivided into two groups according to their sensitivity to aromatase inhibitor treatment: tumors displaying a high frequency of cells expressing the protein Ki67 are aromatase-resistant. Several interesting findings emerged from this comparative analysis: (a) the *TP53* mutations were higher in the aromatase-inhibitor-resistant group (38%) than in the aromatase-inhibitor-sensitive group (16%); (b) *TP53* mutations were significantly enriched in luminal B tumors and higher histological grade tumors; (c) *MAP3K1* mutations were more frequent in luminal A tumors, in grade 1 tumors, and in tumors with lower Ki67 levels (premature inhibitor-sensitive tumors); (d) alterations in DNA replication and mismatch repair are more frequent in the aromatase-inhibitor-sensitive group; and, (e) the presence of mutant *GATA3* correlated with suppression of proliferation upon aromatase inhibitor treatment [40].

The integration of genomic and gene expression studies has recently led to the identification of more breast cancer molecular subtypes. This study was based on the analysis of 2000 breast cancers [41]. In this study, Curtis et al. defined 45 regions of sequence amplification or deletion that deregulate genes that are involved in the pathophysiology of cancer [41]. Among these subtypes, particularly interesting was the identification of the ER-positive subgroup that was composed of 11q13/14 cis-acting luminal tumors that harbor other common alterations. This subgroup is a high-risk subgroup. Various putative driver genes reside in the chromosome region, including *CCND1, EMSY, PAK1*, and *RSF1*. Additional subgroups identified using this analysis were represented by two subgroups characterized by paucity of copy number and cis-acting alterations. Both of these subgroups have good prognosis and one of them is represented by luminal A cases and is enriched in histiotypes corresponding to lobular and tubular carcinomas; the other subgroup included both ER-positive and ER-negative cases [41]. In addition, several intermediate prognosis groups were identified, including a 17q23/20q cis-acting luminal B subgroups, an 8p12 cis-acting luminal subgroups, and an 8q cis-acting/20q^-^ amplifies mixed subgroup [41]. An additional subgroup within intermediate prognosis group is characterized by the classical 1qgain/16q loss, representing a common translocation event [41]. Within these intermediate prognosis groups are included also basal-like tumors, characterized by high-genomic instability and typical cis-acting alterations, such as 5 loss/8q gain/10p gain/12p gain [41]. Finally, the *ERBB2*-amplified cancers represent a separate subgroup with negative prognosis and they are composed of HER2-enriched (ER-negative) cases and luminal (ER-positive) cases [41].

Stephens and coworkers have studied somatic mutations and copy-number variants in 100 breast cancers and observed that the number of somatic mutations varied markedly between individual tumors and discovered nine new cancer genes that were rarely mutated, but that can represent driver mutations. The new cancer genes so identified were *AKT2, ARID1B, CASP8, CDKN1B, MAP3K1, NCOR1, SMARCD1*, and *TBX3* [42]. Interestingly, many of these mutations predict the synthesis of truncated, non-functional proteins, thus suggesting that these genes could act as breast cancer tumor suppressors [42]. In this study, it was also observed a strong correlation between mutation number, age at which cancer was diagnosed and cancer histological grade: particularly, it was observed that certain DNA-base substitutions are clearly associated with the age of the patients in tumors not overexpressing the estrogen receptor (estrogen receptor-negative tumors), but not in tumors overexpressing (estrogen receptor-positive) the estrogen receptor [42].

Three recent studies have provided new important insights into our understanding of the life span of breast cancers and of the mutational processes that occur at the level of cancer genome [43,44,45]. Importantly, these studies have provided a mutation life span in the natural history of development of breast cancer. The whole-genome DNA sequencing, through the interpretation of the results that were obtained using sophisticated algorithms, allowed for proposing an archeological map for the accumulation of point mutations and chromosomal rearrangements occurring during the development of breast cancer. At early time points of cancer development, driver mutations (such as *TP53* or *PIK3CA* mutations or *ERBB2* amplifications) occur and frequently lead to subsequent large-scale chromosomal instability. As a consequence of this event of fundamental importance in breast cancer tumorigenesis, a clonal population of tumor cells is established, which is identified as the “most recent common ancestor”. This initial event is followed by a long period of time during which the tumor subclones acquire new mutations, which make tumor cells more malignant, through various genetic processes: (a) gradual accumulation of genetic alterations; (b) catastrophic genetic events known as chromothripsis (a genetic event of recent identification, called chromothripsis, to indicate shattering of chromosomes into pieces: the shattering event is followed by the stitching of genomic fragments into derivative chromosomes) or kataegis (a phenomenon that is responsible for the rapid development of point mutations that cause regional accumulation of alkylation-based damage of cytosines and guanines [43,44,45]. Nik-Zainal et al. observed that the regional hypermutation (kataegis) is common in breast cancer and described five different kataegis mutational signatures in these cancers, seemingly occurring through different mutational events: signature A was characterized by C > T mutations at XpCpG sites; signature B was represented by C > T, C > G and C > A mutations at TpCpX trinucleotides; signature C was characterized by C > T, C > G and also C > A mutations at XpCpG trinucleotides; signature D showed a uniform distribution of the different mutational classes; and, signature E was characterized by C > G mutations, but not C > T mutations al TpCpX trinucleotides [43]. Multiple mutation processes contribute to most of the breast cancers, although, in some cases, one process was dominant [43]. Finally, later, during tumor development, a late rate-limiting step is responsible for the emergence of one subclone that becomes dominant due to its capacity to expeditiously grow and represents a significant part of the tumor mass.

The Cancer Genome Atlas Network recently published a large survey of the mutational background occurring in human breast cancer [46] (Figure 1).

This analysis was, in large part, confirmatory of the results that were obtained in other studies and provided a very useful general overview of the genetics of breast cancers. The luminal A subtype harbored a considerable number of mutated genes, with the most frequent being *PI3KCA* (about 45%), followed by *MAP3K1, GATA3, TP53, CDH1,* and *MAP2K4*. Approximately 12% of these tumors contained inactivating mutations in *MAP3K1* and *MAP2K4*. Luminal B cancers, although, similar to luminal A cancers, exhibited a diversity of significantly mutated genes, with *PI3KCA* and *TP53* (29% of each) being the most frequent [46]. A feature of these tumors was the high mRNA and protein expression of the luminal expression signature, including *GATA3* and *FOXA1* (mutated in a mutually exclusive fashion), *EISR1, XBP1*, and *MYB* (highly expressed, but scarcely mutated). In addition to a higher frequency of *TP53* mutations, the luminal B cancers are also characterized for more frequent incidence of p53 pathway-inactivating events, such as *ATM* loss and *MDM2* amplifications. The ensemble of these observations indicates that the TP53 pathway is frequently inactivated in the clinical more aggressive luminal B cancers. The retinoblastoma *RB1* expression was detectable in the majority of luminal breast cancers. *Cyclin D1* amplification and high expression, common oncogenic events in luminal tumors, are more frequent in luminal B than in luminal A breast cancers.

The HER2 breast cancer subtype was characterized by the frequent amplifications of the HER2 gene (about 80% of cases), frequent *TP53* mutations (about 72%) and *PIK3CA* mutations (39%), and a lower frequency of *PIK3R1* (about 4%). *Cyclin D1* amplifications were also frequent (about 38%) in these tumors [46]. At the mRNA expression level, a *HER2* amplicon signature characterized this group and the expression of *EGFR2* and *HER2* was observed.

More recently, Nik-Zainal and coworkers reported the landscape of somatic mutations by whole-genome sequencing in 560 primary breast cancers. This analysis showed that at least 12 base substitution mutational signatures and six rearrangement signatures contribute to the somatic mutations found, and 93 mutated cancer genes (31 dominant, 60 recessive, and two uncertain) are involved in the genesis of breast cancer [48]. Three rearrangement signatures, which are characterized by tandem duplications or deletions, are associated with defective homologous-recombination-based DNA repair: one with deficient BRCA1 function, another with deficient BRCA1 or BRCA2 function, and the third related to unknown causes [48]. The global analysis of copy number alterations and somatic mutations generated a total of 1,628 presumptive driver mutations occurring at the level of 93 cancer genes; at least one driver mutation was identifiable in 95% cancers [48]. The ten most frequently mutated genes were *TP53, PIK3CA, MYC, CCND1, PTEN, ERBB2, ZNF703/FGFR1* locus, *GATA3, RB1*, and *MP3K1* [48]. *TP53, PTEN*, and *RB1* were more frequently mutated among ER^-^ tumors, while *GATA3, CCND1, PI3KCA, ZNF703/FGFR1, MAP3K1, MAP2K4*, and *CDH1* were more frequently mutated in ER^+^ tumors [48]. Kataegis (focal base-substitution hypermutation) was observed in 49% of breast cancers [48].

Pereira et al. have reported the analysis of the sequencing of 173 genes that were selected for their “involvement” in breast cancer according to previous studies, in a very large population of 2,433 primary tumors [49]. The mutational landscape was dominated by *PIK3CA* mutations that occur in about 40% of samples and *TP53* mutations occurring in about 35% of cases; the other five genes more recurrently mutated are represented by *MUC16* (16.8%), *AHNAK2* (16.2%), *SYNE1* (12.0%), *KMT2C* (11.4%), and *GATA3* (11.1%) [49]. Some genetic alterations, such as those occurring at the level of *PI3KCA, MAP3K1, CDH1,* and *GATA3* genes are more frequent among ER^+^ than ER^-^ breast cancers; in contrast, *TP53* alterations are markedly more frequent in ER^-^ than ER^+^ breast cancers [49]. The analysis of gene alterations in histological subtypes showed very remarkable differences; (a) in ductal/NST histotype (largely the most frequent), *TP53* and *PI3KCA* mutations are largely the most frequent; (b) in mixed histotype, *PI3KCA* mutations are predominant genetic alterations; (c) in mixed histotype, *CDH1* and *PI3KCA* are the most frequent genetic alterations; (d) in the medullary histotype, *TP53* is largely the predominant genetic alteration; and, (e) in the mucinous histotype, *GATA3* and *MAP3K1* are the predominant genetic alterations [49].

Griffith and coworkers have explored the genetic landscape in 1128 primary breast cancer samples [50]. The analysis of the mutation landscape showed that: (i) 17 genes were mutated at a rate greater than 5% and only six at a rate greater than 10%; (ii) the most recurrent mutations were *PIK3CA* (41.1%), *TP53* (15.5%), *MLL3* (13.4%), *MAP3K1* (12.0%), *CDH1* (10.5%), *MALAT1* (10%), *GATA3* (9.1%), *MLL2* (8.7%), *ARID1a* (7.2%), and *BRCA2* (6.6%); (iii) favorable prognostic associations for breast cancer-specific survival were detected for non-silent mutations in *MAP3K1, ERBB3, XBP1,* and *PIK3CA*, while adverse prognostic effects were observed for non-silent mutations in *DDSR1* and *TOP53*, as well as for frameshift and nonsense mutations in *NF1* [50]. Since *PIK3CA* and *MAP3K1M* mutations often co-associate, their combined effect was explored, showing that patients with tumors exhibiting both genes mutated have a more favorable prognosis than cases with either singly mutant gene or without either gene mutated [50].

In 2013, Ciriello and coworkers performed a comparative analysis of oncogenic signatures among the major human cancers [51]. This analysis showed that tumors are dominated by either mutations (M class) or copy number changes (C class); recurrent chromosomal gains and losses characterizes the C class and includes almost all breast cancers, as well as ovarian cancers [51]. Zack et al. confirmed these findings, who observed that breast cancers, as well as other tumors with frequent CNAs, such as lung cancer, bladder cancer, ovarian cancer, and colorectal cancer, display also with high frequency (45% of cases) whole genome duplication (WGD) [52]. In cancers with WGD, such as breast cancer, most other CNAs occurred after the event of WGD [52].

As previously reported, the acquired CNAs in breast cancers act in cis (a given variant at a locus affects its own expression) or in trans (a given variant affects genes at other sites in the genome) [41]. Approximately 20% of loci exhibit CNA-expression associations in cis and this abnormality includes genes, such as *PTEN, ZNF703, MYC, CCND1, MDM2, ERBB2, CCNE1, MDM1, MDM4, CDK3, CDK4, CAMK1D, PI4KB, NCOR1, PPP2RA, MTAP,* and *MAP2K4* [41]. Trans-acting aberration spots were detected at the level of loci on chromosomes 1q, 7p, 8, 11q, 14q, 16, 17q, and 20q [41].

Single cell genome sequencing methods have been applied to study mutational evolution in ER^+^ and TNBC patients. Wang et al. have combined this approach with targeted duplex single-molecule sequencing to profile thousands of cells and define the role of some genetic alterations in tumor evolution [53]. The study of two patients (one with ER^+^ and the other with TNBC) showed: (i) in both patients, a large number of subclonal and *de novo* mutations gradually evolved over long periods of time, thus generating extensive clonal diversity; and, (ii) in contrast, the single cell copy number profiles were highly comparable, suggesting that chromosome rearrangements occurred early during tumor development, according to punctuated bursts of tumor evolution, leading to stable clonal expansions required for tumor mass formation [53]. Mathematic modeling showed that the TNBC tumor cells displayed an increased mutation rate when compared to ER^+^ breast cancer cells [53].

Chromosomal instability represents a driving element generating multiple DNA copy-number events, selected during disease development, and resulting in ER^+^ breast cancers in selection of gene amplification of core regulators of proliferation: in these tumors, stable DNA copy-number amplifications of the core regulators TPX2 and UBE2C are associated with the expression of a whole gene module associated with cell proliferation [54].

Eighty-five percent of the variations in gene expression of breast cancers are due to somatic CNAs at gene loci [41]. Frequently, CNAs involve oncogenes and tumor suppressors that directly affect breast cancer development and progression. Recent studies have measured the existence of a possible association between CNA burden (defined as the percentage of the genome that is affected by CNAs) and tumor grade, recurrence, and metastasis. Thus, Zhang and coworkers have explored the possible role of CNA burden as a prognostic factor associating with the survival outcome of breast cancer patients [55]. In both METABRIC and TCGA data sets, there was an association between CNA burden and patient’s overall survival, in that patients with high CNA burden have a significantly shorter OS than those with low CNA burden [55]. CNA burdens on chromosomes 1, 8, and 16 were significantly higher than other chromosomes: some aberrations on chromosome 1 closely interact with the genes that are involved in the regulation of humoral immune response, while other aberrant genes located on this chromosome belong to *MAPK/MAPK3*-knockdown related genes involved in cell proliferation; genes that are affected by somatic CNAs on chromosome 8 mainly pertain to the MYC-MAX complex and also include members of the TP53 receptors and ligands gene set; and, genes affected by somatic CNAs on chromosome 16 involve gene sets that are related to cell-cell junction interactions [55]. Furthermore, a string association between CNA burden and age, as well as CNA burden and breast cancer PAM 50 subtypes was observed [55]. Hieronymous and coworkers confirmed these findings, providing evidence that CNA burden of primary and metastatic breast cancers is a prognostic factor, being associated with disease-free and overall survival [56].

Loibl et al. reported the results of a next generation sequencing analysis that was carried out in the neoadjuvant GepurSepto trial; in this study, 851 pretherapeutic formalin-fixed samples were sequenced using a breast cancer-specific hotspot panel of 24 genes (allowing to screen the most relevant mutation and CNA events occurring in this tumor), performing a stratified analysis according to the luminal, HER2-positive, and TNBC subtypes. In this trial, the patients were randomized to either weekly nab-paclitaxel or solvent-based paclitaxel for 12 weeks followed by standard epirubicin/cyclophosphamide; patients with HER2-positive breast cancers received trastuzumab and pertuzumab every three weeks, simultaneously to all chemotherapy cycles [57]. Point mutations and CNAs displayed high heterogeneity among subtypes: *TP53* mutations and *TP53* and *TP2A* amplifications were more frequently observed among TNBC; as expected, *ERBB2* amplifications are almost exclusively observed among HER2^+^ tumors; *PIK3CA* mutations, *ZNF703, CCND1, PAK1*, and *FGFR1* amplifications were more frequently observed among Lum/HER2^-^ breast cancers [57]. In the complete cohort, several genomic alterations were significantly linked to differences in the chemotherapy response in univariate analysis; however, in multivariate analysis, the response to neoadjuvant chemotherapy remained statistically significant for only three of the genomic alterations: *PIK3CA* mutation, *ERBB2* amplification, and *PAK1* amplification [57].

## 4. Molecular Classification of Breast Cancer

Studies that were carried out during the last two decades have strongly supported the prognostic significance and predictive capacity of the breast cancer classification of the four intrinsic subtypes of breast cancer (luminal A, luminal B, HER2-enriched, and basal-like), initially proposed by Perou et al. [58]. These studies started with a genome-wide based gene expression profiling from microarray datasets and then moved to a PCR-based test with a curated list of only 50 genes (the so-called “PM-50” gene signature) to classify breast into one of these four groups [59]. The diagnosis by intrinsic subtype improves prognostic and predictive informations to standard a histopathologic parameter for patients with breast cancer [59].

The intrinsic subtypes of breast cancer displayed consistent differences in incidence and response to treatment. The informations that are provided by the intrinsic subtypes complement and expand the information provided by classical clinical parameters and pathologic markers related to the hormonal receptor status. The intrinsic subtype provided prognostic information for patients with metastatic HR^+^ breast cancer that was treated with first line letrazole ± lapatinib [60]. A recent study explored the changes occurring in intrinsic subtypes at the level of breast cancers undergoing metastatic progression: the rate of subtype conversion was 0% for basal-like tumors, about 23% in basal-like tumors, 30% in luminal B tumors, and 55.3% in luminal A tumors (in large part, these tumors shifted to luminal B) [61].

More recently, the Nanostring nCounter Dx Analysis System provided a system ensuring the more accurate measurements of mRNA expression levels in formalin-fixed tissue when compared to PCR [62]. The system developed while using this system of analysis was called the Prosigna breast cancer signature assay and its prognostic significance was validated [63]. A complete transcript quantification agreement between RNA-Seq and digital multiplexed gene expression platform, and the subtype call after running the PAM assay was observed in a group of breast cancer patients with triple negative cancer [64].

The Prosigna algorithm provides an evaluation of a risk-of-recurrence (ROR) score, being represented as a value from 0 to 100, assessing the risk categories (low, intermediate, or high) and reflecting the 10-year risk of distant recurrence of patients with early stage HR-positive breast cancer. Many studies have clearly supported the utility of data obtaining the Prosigna assay, combined with standard prognostic criteria, to stratify the recurrence risk [65].

Oncotype DX is one of the earliest clinically validated molecular tests for evaluating the clinical risk in breast cancer patients with early stage disease. This assay was based on the evaluation of 21 genes, of which 16 are tumor-associated and five used as controls; the 16 cancer-related genes include five genes that are involved in proliferation (*Ki-67, STK15, Survivin, Cyclin B1, MYBL2*), invasion (*Stromelysin 3, Cathepsin L2*), estrogen (*ER, PR, Bcl2, SCUBE2*), HER2 (*GRB7* and *HER2*), and *GSM1, BAG1,* and *CD68* [58]. The results of this assay are expressed as a recurrence score (RS): <18RS corresponds to low-risk disease; RS 18–30 corresponds to intermediate-risk disease; ≥31 RS corresponds to high-risk disease [66]. In this assay, a higher expression of genes associated with a favorable outcome (*ER, GSTM1, BAG1*) is linked to a lower RS, whereas a higher expression of genes associated with an unfavorable outcome, such as Ki67 and cyclin-B1, contributes to higher RS [66]. The Oncotype DX assay was evaluated and validated in the context of the prediction of 10-year recurrence risk in patients with ER^+^ and LN^-^ breast cancer [66].

The clinical predictive validity of the Oncotype DX assay was evaluated in a large group of postmenopausal women with HR-positive breast cancer that were treated with adjuvant aromatase inhibitors: for patients with node-negative disease, the recurrence rates were 4%, 12%, and 25% for patients with a low, intermediate, and high RS, respectively; for node-positive disease, the recurrence rates were 17%, 28%, and 49% for patients with a low, intermediate, and high RS, respectively [67]. The prognostic value of RS was similar for patients undergoing tamoxifen or aromatase inhibitor treatment [67].

MammaPrint is a 70-gene assay that is based on DNA microarray technology for the assessment of gene expression and quantifies the expression of genes related to tumor progression and metastasis. The FDA approved this test in 2007 for the prediction of the risk of developing metastasis. MammaPrint is currently used in patients with stage II, ER-positive, or ER-negative breast cancers. MammaPrint classifies tumors into groups that are associated with a good prognosis or a poor prognosis on the basis of the risk of recurrence at five years and at 10 years [68]. A large phase III clinical trial explored the clinical utility of MammaPrint as an aid to treatment decisions in early-stage breast cancer. Women at an early-stage of breast cancer were evaluated for the genomic risk while using the MammaPrint assay and for the clinical risk according to standard criteria: at low genomic and clinical risk, these patients did not receive chemotherapy, whereas those with high genomic and clinical risk receive chemotherapy treatment; chemotherapy treatment was used in patients with discordant genomic and clinical risks [69]. The trial that was carried out by Cardoso and coworkers provided evidence that among women with an early-stage breast cancer who were at high clinical risk and low genomic risk for recurrence, the absence of chemotherapy administration on the basis of the MammaPrint assay led to a five-year survival rate without metastasis that was 1.5% points lower than the rate that was observed in patients treated with chemotherapy [69]. A recent study (WSG-PRIMe study) showed that the results of the MammaPrint assay strongly impacted physicians’s therapy decisions in the treatment of patients with luminal early breast cancer [70].

The balance between somatic mutations and alterations in copy number (CNAs) has been investigated in the context of the activities of TCGA, based on the pan-cancer characterization of 12 tumor types [51]. This analysis showed that some of the tumors were dominated by mutations and called M-class tumors, while other tumors are dominated by CNAs and are called C class, such as breast cancer and ovarian cancer [51]. This finding highlights the need for a classification scheme of breast cancers that is based on the pattern of CNAs. To meet this need, Ali and coworkers have performed, in 2014, a large analysis (integration of genomic and transcriptomic profiles) based on over 7500 breast cancer samples and developed a classification system, called IntClus, allowing for the classification of these tumors into 10 IntClust [71]. Integrative cluster 1 is composed by ER-positive tumors, being mainly classified into the luminal B intrinsic subtype; the molecular feature of this cluster is the amplification of the 17q23 locus and *GATA3* and *TP53* the predominant mutations. Integrative cluster 2 englobes ER-positive tumors and both luminal A and luminal B tumors and it is molecularly characterized by high genomic instability and the amplification of 11q13/14 (involving genes such as *CCND1, ERSY, PAK1*) and by frequent (about 50%) *PIK3CA* mutations. Integrative cluster 3 is mainly composed of luminal A subtype and it is enriched for histopathological subtypes, such as invasive lobular and tubular carcinomas, associated with good prognosis; at molecular level, this cluster is characterized by low genomic instability, frequent *PIK3CA, CDH1*, and *RUNX1* mutations, and very rare *TP53* mutations. Integrative cluster 4 englobes both ER^+^ and ER^-^ breast cancers, including 26% of TNBCs and basal-like tumors; at the molecular level, these tumors have low genomic instability and a low level of CNAs; at the mutational level, the most frequent mutations are at the level of *PIK3CA* (28%) and *TP53* (20%). Integrative cluster 5 is composed by both HER2-enriched, ER-negative (58%), and luminal ER-positive tumors, with high-grade tumors with regional lymph nodes involvement; at the molecular level, these tumors are characterized by intermediate levels of genomic instability and a high *TP53* mutation frequency (63%). Integrative cluster 6 is a distinct subgroup of ER^+^ breast cancers, comprising both luminal A and luminal B subgroups and, at the molecular level, is characterized by amplification of the 8p12 locus and high levels of genomic instability; interestingly, these tumors display the lowest PIK3CA mutation rates (about 14%). Integrative cluster 7 mainly comprises ER^+^ luminal A tumors and corresponds to a good prognostic subgroup; at the molecular level, this cluster is characterized by an intermediate level of genomic instability and specific 16p gain and 16q loss and at mutational level by a high *PIK3CA* mutation frequency (42%) and by the highest mutation frequency of *MP3K1* (32%) and *CTCF* (11%) among the various clusters. Integrative cluster 8 comprises tumors that are predominantly of luminal A intrinsic subtype, associated with a good prognosis; at the molecular levels, these tumors are characterized by the classical 1q gain/16q loss event corresponding to a common translocation event; these tumors display high levels of *PIK3CA, GATA3*, and *MAPK24* mutations. Integrative cluster 9 is predominantly composed by ER^+^ tumors of the luminal B subgroup, with an intermediate prognosis; high levels of genomic instability and high mutation frequency of *TP53* (58%) and *PI3KCA* (41%) characterize this cluster. The Integrative Cluster 10 mainly englobes TNBCs and it is molecularly characterized by 5q loss and gains at 8q, 10p, and 12, and by the very frequent TP53 mutations (82%) [72].

A recent study analyzed the relationship existing between the IntClust classification and traditional clinic-pathological features. This analysis showed that: IntClust 3 was enriched for tubular and lobular carcinomas, thus explaining the association with CDH1 mutations in this cluster; mucinous carcinomas were not present in IntClust 5 or 10, but they were scattered in the remaining IntClusts; medullary-like tumors were associated with IntClust 10; HR-positive tumors were scattered along all IntClusts; HER2-positive tumors were predominantly clustered in IntClust 5; and, triple-negative tumors are comprised predominantly in IntClust 10 and in part in IntClust 4 [73].

A statistical framework was recently developed to try to identify peculiar risk groups of breast cancers, while taking the immunohistochemical, intrinsic (PAM50), and integrative (IntClust) subtypes into account. Thus, while using this approach, four late-recurring IntClust subtypes were identified, comprising 26% of ER^+^/HER2^-^, each with characteristic genomic copy number driver alterations and with high (42–55%) risk of recurrence up to 20 years post-diagnosis [74]. Furthermore, a subgroup of triple-negative breast cancers that rarely recur after five years and a separate group that remains at risk were identified [74].

The heterogeneity of ER^+^ breast cancers is supported by many other studies. According to the positivity for PR, ER^+^ breast cancers can be subdivided into PR^+^ (more frequent) and PR^-^ (less frequent, 10–155 of all breast cancers, defined as luminal-like). Patients with ER^+^PR^-^ status exhibit a higher recurrence and worse prognosis, as compared to ER^+^PR^+^ tumors. Some studies have explored the molecular features of ER^+^PR^-^ breast cancers. These tumors have been associated with a significantly higher frequency of HER2 positivity than ER^+^PR^+^ tumors. The PR negativity in these tumors might be related to promoter hypermethylation or a loss of heterozygosity at the PR locus. A recent study provided a fundamental analysis of ER^+^PR^-^ breast cancers, based on the study of five large cohorts of patients. The main results of this study can be summarized, as follows: (i) ER^+^PR^-^HER2^-^ tumors displayed lower endocrine responsiveness that did ER^+^PR^+^HER2^-^ tumors; (ii) copy number loss or promoter methylation of PR genes occur in about 75% of ER^+^PR^-^HER2^-^ tumors, offering an explanation for loss of PR expression; (iii) ER^+^PR^-^HER2^-^ tumors displayed higher *TP53* (30% vs 17%) and lower *PIK3CA* mutation rates (25.8% vs 42.7%) and exhibited more *ZNF703* (21.5% vs 13.6%) and *RPS6KB1* (18.5% vs 7.8%) amplification events than ER^+^PR^+^HER2^-^ tumors; and, (iv) ER^+^PR^-^HER2^-^ tumors were classified according to the PM-50 gene expression assay as luminal A (46%), but, in part, also luminal B (29%) and basal (16) [75]. It was particularly important to determine the fraction of ER^+^PR^-^HER2^-^ tumors that are non-luminal-like in that this subgroup of tumors only showed limited benefit from endocrine therapy, when compared to ER^+^PR^-^HER2^-^ luminal-like tumors [75].

Ethier et al. characterized the ER^+^PR^-^HER2^-^ subtype, who reached the conclusion that these tumors mainly pertain to the luminal B subtype, and are characterized by higher proliferation and worse outcomes [76]. A systematic review and meta-analysis of the literature data showed that, among patients with hormone receptor-positive breast cancer, patients with either ER^+^PR^-^ or ER^-^PR^+^ tumors have a higher risk of recurrence and shorter survival time than those with ER^+^PR^+^ tumors [77]. Patients with both these types of breast cancers need additional or better treatments [77].

Villon-Christersson et al. have reported a cross comparison and prognostic assessment of breast cancer multigene signatures in a large population-based contemporary clinical series of Sweden breast cancer patients. Gene signature classification (the proportion of low- and high-risk) was well aligned with stratification based on current immunohistochemistry-based clinical practice. Most of the signatures did not provide any further risk stratification in TNBC and HER2^+^ER^-^ patients. Risk classifier agreement in the assessment of ER^+^ patients was around 50–60%, the disagreement mostly concerning the evaluation of intermediate-risk patients [78]. Most of the investigated gene signatures provided additional prognostic information beyond conventional clinicopathological factors in some specific clinical groups, mainly ER^+^/HER2^-^ breast cancers [79].

## 5. HER2 Positive Breast Cancer

Approximately 15–20% of breast cancers are associated with HER2-positivity, being defined as evidence of HER2 protein overexpression that is measured by immunohistochemistry or by fluorescence in-situ hybridization (FISH) measurement of a HER2 gene copy number of six or more [79].

Recent studies have extensively characterized the genetic heterogeneity of HER2-positive breast cancers. Thus, Ferrari and coworkers have performed a whole-genome sequence and transcriptome analysis of 99 HER2-positive breast cancers, showing that: (i) at gene expression level, four transcriptomic groups were delineated: A and B groups mostly composed of ER^+^ and PR^+^ and luminal B tumors, while C and D groups were mostly composed by ER^-^ and PR^-^ and HER2-enriched tumors; (ii) 52 genes were mutated in at least four of these tumors and eight of them are known cancer genes: *TP53* (more frequently mutated in ER^-^than ER^+^ tumors and particularly in those pertaining to group D), *PI3KCA* (more frequently mutated in groups A and B), *JAK2, ATRX, MAP2K4, ERBB2, KMT2C,* and *KTM2D*; and, (iii) copy number variation affected 59% of the genome and contributed to the molecular heterogeneity of these tumors: gains of 2p and 2q chromosomal arms are more frequent in group D, loss of 11q was more frequent, while the loss of 14q was less frequent in group A, the amplification of *CCND1* and *PPM1D* gene was more frequent in group A [80]. These observations support a consistent heterogeneity of HER2-positive breast cancers.

Daemen and Manning also supported these conclusions, who, through the analysis of published genomic data relative to 3155 breast cancers, reached the conclusion that HER2-positive breast cancers do not constitute a cancer subtype [81]. In fact, *HER2* amplification is observed in all breast cancer subtypes, with major characteristics restricted to amplification and the overexpression of HER2 and neighboring genes [81]. Interestingly, HER2-positive tumors are highly enriched in estrogen receptor-driven breast tumors, thus suggesting therapeutic opportunities [81].

Zhao and coworkers have characterized the molecular properties of HER2-positive breast cancers subdivided according to the expression of ER and PR, with particular emphasis on the triple-positive (ER^+^PR^+^HER2^+^) subgroup [82]. According to the hormone receptor expression, HER2-positive breast cancers can be subdivided into ER^+^PR^+^HER2^+^ (TPBC), ER^+^PR^-^HER2^+^, ER^-^PR^+^HER2^+^ (very rare), and ER^-^PR^-^HER2^+^. The triple-positive subgroup displays several peculiarities: (i) it had a significantly better prognosis than the group ER^-^PR^-^HER2^+^; (ii) TPBCs displayed a lower TP53 mutation rate than ER^-^PR^-^HER2^+^ breast cancers (30% vs 69%); and, (iii) TPBCs exhibited lower HER2 mRNA and protein expression than ER^-^PR^+^HER2^+^ tumors [82]. More than 40% of TPBCs can be classified as luminal A and these patients have a better prognosis than those with TPBC of other subtypes [82]. Interestingly, this study also showed that *MUC16, GATA3,* and *ERBB3* mutations are strongly associated with the ER^+^PR^-^HER2^+^ phenotype [82]. Finally, concerning CNVs, TPBCs display less frequent *MYC* amplification and *NCOR2* loss, but more frequent *CCND1* amplification and *FANCA* loss than ER^-^PR^-^HER2^+^ tumors [82].

The analysis of primary HER2-positive breast cancers that were obtained from premenopausal Asian women with recurrent breast cancers showed some differences in the rate of several genetic abnormalities, compared to corresponding non-Asian premenopausal breast cancers: particularly, *TP53, KTM2D, KMT2C*, and *SDK1* gene alterations were significantly more frequent in Asian than in non-Asian HER2-positive breast cancer patients [83].

Few studies have explored the genetic abnormalities of HER2-positive breast cancers resistant to trastuzumab therapy. A recent study showed that the mutational burden in heavily treated trastuzumab-resistant HER2-positive metastatic breast cancer is highly variable and not directly correlated with outcome [84]. The activation of the MAPK/ERK pathway through mutations in *EGFR, BRAF*, or *KIT* might mediate resistance to trastuzumab [84].

Importantly, the intratumoral heterogeneity of HER2-positive breast cancers with respect to tumor genomics significantly affected the probability of achieving a pathological complete remission following neoadjuvant therapy that is based on chemotherapy and HER2 targeting with specific monoclonal antibodies [85].

Recent studies indicate that a peculiar gene signature could predict the response to therapy of HER2^+^ breast cancers. This predictive gene signature was obtained while taking two important elements into account: (a) HER2 overexpression drives mammary carcinogenesis, via its effects on normal and malignant mammary stem cells; (b) in a mouse mammary model of HER2^+^ breast cancer, mammary stem cells (CD24^+^/JAG1^-^) have been purified and used to generate a 17-gene signature [86]. This HER2 Tumor Initiating Cells-Enriched Signature (HTICS) consists of eight up-regulated (*AURKB, CCNA2, SCRN1, NPY, ATP7B, CHAF1B, CCNB1, CLDN8*) and nine downregulated genes (*NRP1, CCR2, C1QB, CD74, VEAM1, CD180, ITGB2, CD72, ST8SIA4*): the up-regulated set includes genes that are associated with passage through the S/G2/M phase of the cell cycle, whereas the down-regulated genes include genes that are involved in cell adhesion, angiogenesis, and immune-response [86]. This signature was specific for HER2^+^/ER^-^ breast cancers and it identifies tumors that are resistant to chemotherapy, but sensitive to chemotherapy+trastuzumab [86].

Lesurf et al. provided clear evidence that HER2-positive breast cancers classified as HER2-enriched achieved significantly higher rates of complete remission when compared to those as luminal A, luminal B, or basal-like [87]. Furthermore, immune and inflammatory signatures correlated with response to neoadjuvant therapy based on chemotherapy and anti-HER2 antibody [87]. Several other studies supported these conclusions. HER2-positive breast cancer consists of four intrinsic molecular subtypes, luminal A, luminal B, basal-like, and HER2-enriched, with the last-one being the predominant subtype corresponding to about 60–70% of all HER2-positive breast cancers. HER2-enriched subtype is a predictor of complete response following neoadjuvant therapy with dual HER2 inhibitors (trasuzumab and lapatinib) without chemotherapy in early stage HER2-positive breast cancers [88]. Furthermore, in the randomized clinical trial NeoALTTO, the expression of HER2 and the HER2-enriched subtype were the most significant predictors of pathological response [89]. Finally, Pernas et al. reported the analysis of intrinsic tumor subtypes and residual tumors following neoadjuvant trastuzumab-based chemotherapy in a group of 150 patients with stage II-IIIC HER2-positive breast cancers [90]. This study was focused in order to evaluate the association of genomic variables with pathologic response [90]. In these patients, the complete pathological response after neoadjuvant chemotherapy was 53% with higher responders among HR-negative tumors when compared to HR-positive tumors (70% vs 39%); the HR-negative HER2 breast cancers were enriched in HER2-enriched tumors (75%) [90]. The study in pre- and post-treatment samples derived from patients not achieving a complete pathological response, showed a lower proportion of HER2-enriched and twice the number of luminal tumors were observed at baseline, and luminal A was the most frequent intrinsic subtype in residual tumors; interestingly, the majority of luminal A tumors maintained the same subtype in residual tumors, whereas HER2-enriched tumors changed to non-HER2-enriched tumors [90].

In a recent study, Prat and coworkers evaluated 305 breast cancer patients with early HER2-positive disease and 117 patients with advanced HER2-positive disease. These patients were evaluated in the context of five different clinical trials for the response to dual HER2 blockade therapy and HER2-enriched subtype with the PAM 50 assay and ERBB2 mRNA levels [91]. In early disease, the HER2-enriched subtype corresponded to about 84 and 45% of ERBB2-high and ERBB2-low tumors, respectively. After lapatibin and trastuzumab neodjuvant treatment, the HER2-E/ERBB2-high group achieved a rate of complete pathological responses that was higher than the rest of HER2-positive patients (44.5% vs 11.6%); similar findings were observed in early patients undergoing neoadjuvant treatment with trastuzumab and pertuzumab (66.7% complete responses among HER2-E/ERBB2-high patients, when compared to 14.7% in the rest of patients); finally, the HER2-E/ERBB2-high group exhibited longer PFS and OS in patients with advanced disease [91].

HER2-positive breast cancers exhibit aggressive clinical behavior, responding only moderately to chemotherapy, and have higher rates of recurrence and metastasis. The introduction of HER2-targetd therapies, including the monoclonal antibodies anti-HER2 trastuzumab and pertuzumumab and the HER2 tyrosine kinase inhibitor lapatinib, has revolutionized the therapy and substantially improved the outcomes of patients with HER2-positive breast cancers. However, the development of resistance to anti-HER2 treatment represents a consistent challenge, which indicates the clinical need for novel therapies.

Importantly, recent studies have reported the long-term effects of trastuzumab administered alone or in combination with chemotherapy to HER2-positive breast cancer patients, all showing a significant effect of this drug on disease-free survival and overall survival [92,93,94].

In this context, recently, two new drugs targeting HER2 were introduced in the clinical treatment of HER2-positive breast cancers: (i) trastuzumab emtansine, an antibody-drug conjugate of trastuzumab with the cytotoxic agent emtansine, a microtubule inhibitor; (ii) pyrotinib, an irreversible pan-ERB receptor tyrosine kinase inhibitor targeting HER1, HER2, and HER4. Thus, the risk of recurrence of invasive breast cancer or of death was 60% lower with adjuvant trastuzumab emtansine than with trastuzumab alone among patients with HER2-positive early breast cancer who had residual invasive disease after completion of neoadjuvant therapy [95].

Lapatinib combined with capecitabine is one of the recommended regimens for patients with HER2-positive metastatic breast cancer who have been pretreated with taxanes, anthracyclines, and trastuzumab. In this context, a recent study showed that, in women with HER2-positive metastatic breast cancer treated with taxanes, anthracyclines, and/or trastuzumab, pyrotinib plus capecitabine elicited significantly better overall survival than lapatinib plus capecitabine [96].

As repeatedly emphasized, HER2-positive breast cancers are a family of distinct diseases, particularly including ER^-^ and ER^+^ tumors. HER2-positive tumors display important biological differences, implying differences in drug sensitivity. Data that were derived from a consistent set of clinical trials indicate that, within the group of HER2-positive tumors, the ER^-^ subgroup is clearly more sensitive than the ER^+^ subgroup to anti-HER2 treatment [97]. More particularly, patients with triple-positive breast cancer are less responsive than patients with HER2-posive, ER-negative tumors to achieve a pathologic complete response in neoadjuvant trials, including HER2-targeted therapies, involving the dual blockade of the HER2 receptor with trastuzumab and pertuzumab or combination treatment with the tyrosine kinase inhibitor klepatinib [97]. In this context, an example of this study is given by the CALGB 40601 clinical trial, a randomized phase III trial evaluating HER2 targeting with paclitaxel plus trastuzumab with or without lapatinib in HER2-positive patients with operable HER2-positive breast cancer [85]. This trial evaluated the capacity of the neoadjuvant treatment to induce pathologic complete response while considering the phenotypic and molecular heterogeneity of HER2-positive breast cancers [85]. The results of this study clearly showed that the rate of responses to treatment were clearly influenced by HR status and the intrinsic subtype; thus, ER^+^/HER2^+^ tumors responded less to HER2-targeting treatments than ER^-^/HER^+^ tumors; tumors pertaining to the HER2-E subtype respond better than luminal A and luminal B to HER2-targeting therapy [85] (Figure 2). In a subsequent analysis of the pretreatment tumors of these patients by mRNA sequencing and DNA exome sequencing, it was shown that somatic DNA alterations (mutations and DNA copy number alterations), tumor molecular subtype and the microenvironment (immune cells) were independent predictors of response to neoadjuvant treatment [98]. 

HER2-positive breast cancer patients consist of four intrinsic molecular subtypes, luminal A, luminal B, basal-like, and HER2-enriched, the last one being the predominant subtype corresponding to approximately 60–70% of all HER2-positive breast cancers. HER2-enriched subtype is a predictor of complete response following neoadjuvant therapy with dual HER2 inhibitors (trastuzumab and lapatinib) without chemotherapy in early stage HER2-positive breast cancer [88]. The presence of stromal tumor infiltrating lymphocytes (TILs) is associated with complete response and improved outcomes in HER2^+^ breast cancer patients treated with trastuzumab plus chemotherapy. The analysis of patients enrolled in the PAMELA trial showed that the number opf TILs during treatment, but not at baseline, is an independent determinant of response to neoadjuvant anti-HER2 therapy [99].

In the randomized clinical trial NeoALTTO, the expression of HER2 and the HER2enriched subtype were the most significant predictors of pathological response [89]. The 41-gene classifier TRAR was able to identify patients sensitive to anti-HER2 enrolled in the context of the NeoALTTRO study [100].

Pernas et al. have reported the analysis of intrinsic tumor subtypes and residual tumors following neoadjuvant trastuzumab-based chemotherapy in a group of 150 patients with stage II-IIIC HER2-positive breast cancers [90]. This study was focused in order to evaluate the association of genomic variables with pathologic response [90]. In these patients, the complete pathological response after neoadjuvant chemotherapy was 53% with higher responders among HR-negative tumors when compared to HR-positive tumors (70% vs 39%); the HR-negative HER2 breast cancers were enriched in HER2-enriched tumors (75%) [90]. The study in pre- and post-treatment samples that were derived from patients not achieving a complete pathological response, showed a lower proportion of HER2-enriched and twice the number of luminal tumors were observed at baseline, and luminal A was the most frequent intrinsic subtype in residual tumors; interestingly, the majority of luminal A tumors maintained the same subtype in residual tumors, whereas the HER2-enriched tumors changed to non-HER2-enriched tumors [90]. A large meta-analysis of literature data showed that the HER2-enriched biomarker identifies breast cancer patients with a higher chance of achieving a complete pathological response following neoadjuvant anti-HER2-based therapy beyond hormonal receptor status and chemotherapy [101]. 

Preclinical studies have supported the use of CDK4/6 inhibitors in the treatment of HER2-positive breast cancers, this strategy being particularly promising in HER2^+^ER^+^ tumors for the presence of a crosstalk between signaling-linked to HER2 and ER. The phase II open label NA-PHER2 trial evaluated the activity of a combination regimen that was based on trastuzumab, pertuzumab, pabociclib, and fuvestrant in patients with triple-positive breast cancers [102]. This regimen elicited a marked reduction of Ki67 expression at week two post-treatment and at surgery after 16 weeks of treatment; importantly, 50% of treated patients achieved a complete clinical response and 27% achieved a pathological complete response [102]. The results of this study support the further exploration of CDK4/6 inhibitors in combination with anti-HER2 therapy and endocrine therapy in the therapy of patients with triple-positive breast cancers. Additionally, preliminary results of the SOLTI-1303 PATRICIA trial further supported the use of CDK 4/6 inhibitors in the treatment of triple-positive breast cancers. The results on the first 45 recruited patients showed that palbociclib in combination with tratuzumab is safe and active in HER2-positive breast cancers with advanced disease, pre-treated with trastuzumab, particularly at the level of HER2^+^/ER^+^ patients [103]. Patients HER2-positive pertaining to luminal intrinsic subtypes respond much better than those that correspond to non-luminal subtypes [103].

Tumors displaying HER2-mutated tumors represent a peculiar subgroup of HER2^+^/ER^+^ breast cancers. A recent study showed that ERBB2-activating mutations occur with increased frequency in metastatic breast cancers; in fact, 70% of the ERBB2 mutations are detectable in HER2^+^ER^+^ non-amplified breast cancers, with a higher frequency occurring in metastatic tumors (4.3%) compared to primary cancers (2.5%) [104]. Importantly, the inhibition of mutant HER2 function with neratinib restores the efficacy of antiestrogen therapy [104]. In line with these findings, it was proposed dual blockade of the HER2 and ER pathways as a strategy that is required for the treatment of HER2^+^/ER^+^ breast cancers [104]. Neratinib is a TKI, approved by FDA for extended adjuvant treatment of early-stage HER2-amplified breast cancer following completion one-year of trastuzumab-based therapy. The approval was based on the results of the ExteNET randomized study evaluating neratinib when compared with placebo as an extended adjuvant tharapy in patients with early-stage HER2-positive breast cancer who had completed treatment with trastuzumab: patients that were treated with neratinib had significantly fewer iDFS (invasive disease-free survival) events than those did in the placebo group [105]. Importantly, the benefit of adjuvant neratinib was more pronounced in patients with ER^+^ breast cancers [105]. Some authors suggested the capacity of neratinib to block CCND1 (cyclin D1) and through this mechanism to maintain tumor suppression [106].

The analysis of the long-term outcomes of HER2-positive patients that were enrolled in the GepartQuinto trial showed a similar survival benefit in patients HR^+^ receiving prolonged anti-HER2 treatment with neoadjuvant lapatinib, followed by adjuvant trastuzumab [107].

Sudhan explored the mechanisms through which extended adjuvant neratinib treatment achieved a better clinical outcome in patients with HER2^+^ER^+^ breast cancers; these authors have developed a human-in-mouse breast cancer model and, through the analysis of this model, reached the conclusion that HER2^+^/ER^+^ tumors rapidly evade ER blockade through ERBB pathway hyperactivation and incomplete suppression of cyclin D1 by estrogen inhibitors, requiring the complete suppression of the effect of nerotinib [108].

## 6. Genetic Abnormalities of Triple-Negative Breast Cancer

Nearly 10% to 20% of primary breast cancers are triple-negative breast cancers that lack expression of ER, PR and HER2, have usually a high-degree at presentation and display frequent *TP53* mutations.

Although often thought to be synonymous, TNBC and BLBC (basal-like breast cancer) represent different biologic phenomena. Rakha and coworkers have explored this important topic by studying two large cohorts of breast cancer patients with a large panel of biomarkers to explore the clinicopathologic differences that exist between TNBCs that express one or more of basal markers and defined as BLBC (such as cytokeratin, CK, CK5/6, CK17, CK14, and EGFR, about 70%) and TNBCs that do not express these markers, being defined as TNBCKE^-^ (about 30%) [109]. Although the morphologic features of BLBC and TNBCKE^-^ were similar, these tumor subtypes differed for several clinicopathologic features: BLBCs are associated with the expression of hypoxia-associated factor (CA9), neuroendocrine markers, and markers, such as p53; BLBC tumors display more frequently BRCA1 alterations when compared to TNBCKE^-^ (37% vs 4%); finally, the BLBC tumors show a unique pattern of tumor metastasis and respond better to chemotherapy, but they have a shorter survival when compared to TNBCKE^-^ tumors [109].

Claudin-low is another intrinsic subtype that is associated with the TNBC phenotype. Claudin-low and metaplastic breast cancers, triple negative breast cancers, form a group enriched in tumors displaying EMT features. The molecular characterization of these tumors showed several interesting and peculiar findings: the loss of genes that are involved in cell-cell adhesion; enrichment for stem cell-like and EMT markers; and, frequent genomic aberrations that activate the PI3K/AKT pathways [110]. These peculiar molecular features represent an important support to understand the poor response to therapy and the cellular origin of this cancer subtype. Interestingly, the claudin-low and metaplastic breast cancer subtypes exhibit a gene expression signature, called “EMT core signature”, which is similar to that displayed by human mammary epithelial cells induced to undergo an EMT by expressing Snail, Twist or by TGF-beta1 [111]. Claudin-low are present in about 7–14% of all breast cancers [112]. Approximately 70% of claudin-low breast cancers are TNBC, with high frequency of metaplastic and medullary breast carcinomas; these tumors are characterized by low levels of cell adhesion molecules and elevated expression of immune-related genes, such as CD4 and CD79a, and by mesenchymal features (high expression of CD44, vimentin, and N-cadherin) and low epithelial differentiation (low CD24 expression), resembling a mammary stem-like phenotype, which was acquired by EMT [112].

Some recent studies were dedicated to the characterization of the molecular abnormalities occurring in 104 TNBCs at the time of diagnosis and they showed a consistent degree of heterogeneity, with some samples displaying coding somatic aberrations that were limited to few pathways, while other samples contained hundreds of coding somatic mutations [110]. The overall pattern of CNAs in TNBCs resembled that generally observed in breast cancers, with the most frequent CNA events occurring at the level of *PARK2* (6%), *RB1* (5%), *PTEN* (3%), and *EGFR* (5%) [108]. The analysis of gene mutations showed that p53 is the most frequently mutated gene (62% in basal TNBC and 43% in non-basal TNBCs); frequent mutations in *PIK3CA* (10.2%), *USH2A* (9.2%), MYO3A (9.2%), *PTEN*, and *RB1* (7.7%) and *ATR*, *UBR5*, and *COL6A3* genes (6.2%) are also observed [110]. The analysis of clonal distribution of mutations further supported the consistent heterogeneity of TNBCs: the basal subtype of TNBC displays some more variation than the non-basal TNBC; *TP53*, *PIK3CA*, and *PTEN* somatic mutations are clonally dominant over other genes; in other tumors, their clonal frequencies are incompatible with a founder status [113]. Finally, mutations in cytoskeletal, cell shape, and motility proteins occurred at lower clonal frequencies, thus implying that these genetic events occurred during tumor progression [113].

A study by Lehmann et al., which was based on an aggregate analysis of publically available expression data set, confirmed the consistent genetic heterogeneity of TNBCs, with the identification of seven subtypes: LAR-positive displaying a luminal pattern of gene expression (high expression of *GATA3, FOXA1*), with elevated androgen receptor (AR) levels, corresponding to luminal A and luminal B intrinsic subtypes; claudin-low-enriched mesenchymal, characterized by low claudin expression and enrichment in angiogenesis and stem cell-associated genes; mesenchymal stem-like, being characterized by expression profiles related to cell motility, differentiation, and EMT; immune response (IM), characterized by expression of genes that are involved in antigen presentation and processing, immune cell, and cytokine signaling; and two-cell cycle-disrupted basal subtypes: BL-1 (characterized by high expression of genes involved in DNA-damage response and cell-cycle regulation) and BL-2 (characterized by high levels of growth factor signaling and metabolic pathway activity); an unstable (UNS) cluster, being characterized by genetic instability [114]. The seven-subtype classification predicted a pathological response to neoadjuvant chemotherapy, but not overall survival in retrospective studies [115]. Another study of targeted ultra-deep sequencing that was carried out on 104 TNBCs confirmed the above reported findings, with clonal *TP53* mutations being present in about 80% of samples and more subclonal mutations occurring at the level of the PI3K pathway (29.8%), MAPK signaling pathway (8.7%), and cell-cycle regulators (14.4%) [116].

A transcriptional study that was based on the analysis of 84 TNBC samples led to the identification of four TNBC subgroups that were associated with different clinical outcomes: luminal/AR, mesenchymal, basal-like/immune-suppressed, and basal-like/immune activated groups [117]. In this classification, a luminal androgen receptor (LAR) subtype is characterized by androgen receptor signaling; a basal-like subtype subtype is characterized by high immune cell signaling and cytokine signaling gene expression (BLIA); a basal-like and immune-suppressed (BLIS) subtype is characterized by an upregulation of cell cycle, the activation of DNA repair, and cytokine signaling gene expression; and, a mesenchymal-like (MES) subtype is characterized by enrichment in the expression of mammary stem cell pathways. In line with the previous study, TNBC tumors with tumors most expressing immune component features have the best outcome [117].

Subsequently, Lehmann et al. simplified their TNBC classification into seven groups and moved to a new classification in four groups: BL1, BL2, M, and LAR [118]. This change was justified by the observation that the histological analysis and laser microdissection before RNA isolation for gene expression studies provided evidence that the presence of stromal cells largely influenced the definition of the IM and MSL subtypes that were removed from the revised classification [118].

Jiang et al. have proposed a slightly different classification into four transcriptome-based subgroups: (i) a luminal androgen receptor (LAR) subtype (23%), being characterized by androgen receptor signaling; (ii) an immunomodulatory (IM) subtype (24%) with high immune cell signaling and cytokine signaling gene expression; (3) a basal-like and immune-suppressed (BLIS) (39%), characterized by the upregulation of cell cycle, activation of DNA repair, and downregulation of immune response genes; and, (iv) a mesenchymal-like (MES) subtype (15%) enriched in mammary stem cell pathways [119]. These authors also reported a very extensive characterization of molecular alterations that were observed in TNBC at the level of both somatic mutations and copy number alterations [119] (Figure 3). Echavarria and coworkers have explored the pathological response of TNBC patients subdivided according to the simplified Lehmann’s classification to standard neoadjuvant chemotherapy regimen that is based on carboplatin and docetaxel and observed the highest response rates among BL1 (65.6%) tumors, followed by BL2 (47.4%), M (36.4%), and LAR (21.4%) [120].

Bareche and coworkers analyzed the data that were related to TNBC patients on CNAs, somatic mutations, and gene expression contained in TCGA and Molecular Taxonomy of Breast Cancer International Consortium (METABRIC) and subdivided according to Lehmann’s classification, showing that: (i) BL1 subtype was identified as the most genomically unstable subtype (with the highest number of CNAs), with high TP53 mutation rates (92%) and copy number alterations in genes involved in DNA repair mechanisms (*BRCA2, MDM2*), AKT signaling (*PTEN*), and cell-cycle regulation (RB1) and high gain/amplifications of *MYC, CDK6,* and *CCNE1*; (ii) the LAR tumors were associated with higher mutational burden with enriched mutations at the level of the *PI3KCA* (55%), *KMT2C* (19%), *AKT1* (13%), *CDH1* (13%), *NF1* (13%), and *AKT1* (13%), and with higher frequency of gain/amplification of *EGFR* and *AKT1*; (iii) the M and MSL subtypes were associated with higher signature score for angiogenesis and low claudin expression; furthermore, the M subtype was associated with higher frequencies of gain/amplification levels of *DNMT3A* and *TP53* and enriched for EGFR and NOTCH signaling; (iv) IM showed a high expression of immune signatures and check inhibitor genes, such as *PD1, PDL1,* and *CTLA4* [121]. The presence of specific differences in mutational and CNA profiles between the various TNBC subtypes offers the way to potential trherapeutic approaches [121]. Thus, the presence in the BL1 subtype of high genomic instability, high copy number losses for *TP53, BRCA1/2,* and *RB1* genes, as well as copy number gains for PPAR1 gene, support the view that these tumors might be sensitive to PARP inhibitors. The pattern of genetic alterations of *RB1* and of expression of *CDK4* and *CDK6* in LAR and MSL subtypes suggests a potential sensitivity to CDK4/6 inhibitors. EGFR and NOTCH signaling pathways are activated in the M subtype, suggesting a possible targeting of these signaling pathways in these tumors. Finally, the pattern of expression of immune checkpoint inhibitor genes in IM subtype suggests a possible benefit deriving from therapy with checkpoint inhibitors [121].

Other studies have identified genes, whose expression might help to classify TNBCs and predict their response to therapy. Thus, Quist and coworkers identified a four-gene decision tree signature, which robustly classified TNBCs into six subtypes; all four genes, *EXO1, TP53BP2, FOXM1*, and *RSU1*, in the signature are associated with either genomic instability, malignant growth and tumor progression, or treatment response [122]. One of these six subtypes, MC6, represented the largest part of tumors (about 50% of primary TNBCs); in metastatic TNBC patients, only 25% of the tumors pertain to the MC6 subtype and they are associated with a higher response rate to platinum-based chemotherapy [122]. Hsu et al., using a gene co-expression network analysis, identified the immunoglobulin-related genes *IGHA1, IGHD, IGHG1, IGHG3*, and *IGLJ3* as the suppressor genes in the recurrence of TNBC patients; an immune score was established according to the expression of these six genes, being able to predict for the recurrence of TNBCs: as the score increases, the risk of recurrence decreases [123].

Jiang and coworkers performed a wide analysis of clinical, genomic, and transcriptomic data of a cohort of 465 of primary TNBC of Chinese patients [119]. These TNBCs have more frequent *PIK3CA* mutations and chromosome 22q11 copy-number gains than non-Asian TNBCs [119]. The LAR subtype showed more ERBB2 somatic mutations, infrequent mutational signature, and frequent *CDKN2A* loss [119]. LAR patients are candidates for combination therapy with the PI3K pathway and androgen inhibitors; furthermore, *CDKN2A* loss and *CCND1* amplification may render LAR tumors potentially sensitive to CDK4/6 and AR inhibitors [119].

Using multi-omics datasets of TNBC, the heterogeneity of tumor microenvironment it was explored, providing evidence about three different clusters: cluster 1 was characterized by an incapability to attract immune cells, and *MYC* amplification was correlated with a low immune response; cluster 2 was characterized by chemotaxis, but inactivation of innate immunity and low tumor antigen burden, thus contributing to immune escape, in association with mutations of the PI3K-AKT pathway; cluster 3 was characterized by a high expression of immune checkpoint molecules [124]. These observations have some obvious potential therapeutic implications.

Tumors that were present in the public datasets, related to 494 TNBCs (153 treated with neoadjuvant chemotherapy) were evaluated for the development of a novel classification of these tumors. The tumors were subdivided into four subgroups, LAR, basal, claudin-low, and claudin-high, while using the cancer stem cell hypothesis as reference [125]. Samples with high luminal metanode activity were classified as LAR; tumor with low luminal metanode activity and high basal metanode activity were classified as basal; tumors low in both the luminal and basal metanodes were screened for claudin expression and then subdivided into a low-claudin and high-claudin subgroups, according to the claudin levels [125]. 18% of TNBC tumors corresponded to LAR, 11% to claudin-low, 63% to basal, and 8% to claudin-high subgroups [125]. This classification did not show any prognostic impact. Taking the immune metanode into account, the tumors were split according to their immune activity into low/high; in the whole TNBC population, 52% of tumors were IM-high and 48% IM-low: IM-low and IM-high tumors were present in all the four subgroups [125]. The immunological subdivision of LAR and claudin-high subgroups introduced a prognostic tool: in fact, LAR and claudin-high immune-negative TNBCs were associated with a significantly lower survival than the corresponding subgroups immune-positive; the immunological subdivision did not impact the prognosis of basal and claudin-low tumors [125].

Other studies were focused on defining the changes in genetic alterations that occurred in TNBCs. Tumor cells remaining after neoadjuvant chemotherapy contain cell populations that are intrinsically resistant to chemotherapy. Balko and coworkers have explored the molecular profiling of the residual disease of TNBCs after neoadjuvant chemotherapy and they have raised some observations: (i) when compared with basal-like primary tumors in the TCGA database, a higher frequency of MCL1 amplifications and a tendency to higher frequency of *PTEN* deletions or mutations and *JAK2* amplifications in residual disease were observed; (ii) some of the molecular alterations in the residual disease after neoadjuvant chemotherapy, such as *JAK2* amplification, and *MYC* amplifications predicted poor overall survival, while *PTEN* alterations were a favorable prognostic factor [126].

In another study, Jiang and coworkers have identified gene abnormalities conferring sensitivity to neoadjuvant chemotherapy in TNBC [127]. This study was based on the analysis of 29 TNBCs, in part (18/29) achieving a complete pathological response following neoadjuvant chemotherapy; the results of this analysis showed that, although mutations in single genes were not individually predictive, TNBCs exhibiting mutations in genes that are involved in AR and FOXA1 pathways were much more sensitive to chemotherapy [127]. Furthermore, mutations that lowered BRCA1 or BRCA2 RNA are associated with a better overall survival; a BRCA deficiency signature was defined and used to define a subset of chemosensitive TNBCs [127]. BRCA deficient tumors were characterized for a higher number of mutations per clone, a higher level of immune activation, and a higher mutational burden [127].

Other studies have shown the existence of a hereditary genetic component in a part of TNBC patients. Thus, over 80% of hereditary *BRCA1*-mutated breast cancers are classified as TNBC and about 15% of TNBCs occur in carriers of a BRCA germline mutation (gBRCA) [128,129]. The *BRCA1* and *BRCA2* gene products play an essential role in the activation and transcriptional regulation of DNA damage and cell-cycle control. Particularly, the BRCA1 and BRCA2 proteins exert a key role in DNA double-strand break repair by homologous recombination (HRR) and the maintenance of DNA stability. The results of a meta-analysis, including a large set of literature data, provided evidence that breast cancer patients with *BRCA1*^Mut^ carriers were more likely to have TNBC than those of *BRCA2*^Mut^ carriers or non-carriers [130]. Other genes and genetic elements, beyond *BRCA1* and *BRCA2*, have been associated with an increased risk of TNBC [131].

A set of recent studies characterized the TNBC exhibiting deficiencies of homologous recombination repair (HRDs). HR is a biochemical pathway that coordinates the repair with the high-fidelity of double-stranded DNA breaks. HRD determines a condition of cellular dependency on alternative, error-prone DNA-repair pathways; the activation of this pathway determines the development of characteristic genomic alterations, higher mutational rates, and specific dependencies that can be exploited for therapeutic targeting of these tumors. Germline and acquired *BRCA1* and *BRCA2* alterations represent the most typical genetic alterations that determine a condition of HRD, leading to the inactivation of these tumor suppressors. Previous studies carried out in various tumors have shown that the loss of *BRCA1* or *BRCA2* determines a typical pattern of base-substitution mutations that are commonly called signature 3 [132]. In this context, Polak and coworkers have explored DNA signatures in a large cohort of 992 breast cancers and detected four recurrent signatures: (i) signature 1 (C > T at CpG sites); (ii) APOBEC-related signatures; (iii) signature 6, associated with microsatellite instability; and, (iv) a uniform signature, similar to signature 3, which is associated with *BRCA1* and *BRCA2* mutations [133]. In this study, 250/992 cases displayed a signature 3: 60 of these cases displayed 10 *BRCA1*, 29 *BRCA2*, 19 *RAD51C*, 2 *PALB2* gene alterations that can be directly related to HRD [133]. The most frequent alterations involving *BRCA1* imply *BRCA1* germline biallelic mutations, somatic null mutations, epigenetic silencing, and mRNA downregulation; for the *BRCA2* gene, the most frequent alterations involve germline biallelic mutations, gene deletion, and other undetermined germline alterations; *RAD1C* alterations are almost exclusively relatable to epigenetic silencing mechanisms [133]. Interestingly, a logistic regression model allowed for accurately detecting a sensitivity of about 99% *BRCA1/BRCA2*-deficient tumors, including both of those exhibiting the loss and functional deficiency of *BRCA1/BRCA2* genes [134]. Wide-genome sequencing of matched germline/tumor DNA, coupled with the somatic mutational signatures, represent a very sensitive tool for the definition of the etiology of familial breast cancer and the prediction of HRD and consequent sensitivity to PARP inhibitors [135].

Staaf et al. used the HRD detect mutational-signature-based algorithm to screen a large population of TNBCs, allowing for the classification of these tumors into threes different subgroups: HRD detect-high, HRD detect-intermediate and HRD detect-low [136]. Fifty-nine percent of these tumors were classified as HRD detect-high and 67% of them are explained by germline/somatic mutations of *BRCA1/BRCA2, BRCA1* promoter hypermethylation, *RAD51C* hypermethylation, or biallelic loss of *PALB2*; this group displayed better outcome on adjuvant chemotherapy as compared to HRD detect-low patients [136]. HRD-detect intermediate patients are minority (about 5%) and they have the poorest outcomes [136]. Finally, HRD detect-low TNBCs display poor outcomes, frequent PIK3CA/AKT1 pathway abnormalities, and in 4.7% of cases were mismatch-repair-deficient [136].

The identification of a condition of HRD in TNBC patients is important and clinically relevant, because it is associated with a clinical response to platinum compounds or PARP inhibitors. Thus, a recent study showed that the HRD score predicts the response to platinum-containing neoadjuvant chemotherapy in patients with triple-negative breast cancer [137].

Echavarria and coworkers evaluated the response to neoadjuvant chemotherapy (NACT) with carboplatin and docetaxel in a cohort of TNBC patients, which were classified in subtypes according to Lehmann’s refined classification [120]. The response to NACT was significantly associated with Lehmann’s subtype, even in multivariate analysis with the highest pCR rate in BL1 (65%), followed by BL2 (47%), M (36%), and LAR (21%) [120]. The LAR subtype was predominantly composed by non-basal intrinsic subtype, while BL1, BL2, and M tumors mainly correspond to basal-like PAM50 intrinsic subtype [120].

These observations have represented the background for clinical studies attempting to verify the therapeutic activity of PARP inhibition in TNBC. The subgroup of TNBC patients with germline BRCA mutant had an objective response rate of 55% and displayed a clinical benefit at the level of PFS in comparison with patients receiving the physician’s choice of treatment in the registration phase III trial of the PARP inhibitor olaparib (5.6 months vs 2.9 months, respectively) [138,139]. In the registration phase III trial of talazoparib tosylate, patients with germline BRCA mutation exhibited an objective response rate of 62% and a PFS of 5.8 months [140,141]. Monotherapy with PARP inhibitors had an efficacy that was limited to patients with *BRCA* mutations. A recent clinical trial evaluated the efficacy of a PARP inhibitor (niraparib) with an anti-programmed death receptor-1 in the treatment of advanced, metastatic TNBC [142]. A combination of niraparib with pembrolizumab (anti-PDL1) displayed anti-tumor activity in patients with advanced TNBC, particularly in patients with tumor *BRCA* mutations [142]. A recent study evaluated the efficacy of talazoparib in a neoadjuvant setting of 20 patients with operable breast cancer with germline BRCA-disease (15 patients were TNBC) [12]. The results of this study showed that neoadjuvant single-agent oral talazoparib once per day for six months without chemotherapy resulted in a high-rate (53%) of complete responses [143].

*BRCA 1–2* mutations predispose TNBC not only to an enhanced sensitivity to PARP inhibitors, but also to platinum-containing agents. The TNT trial evaluated carboplatin in *BRCA 1–2*-mutated TNBC BRCAness subgroups [144]. In the unselected patient population carboplatin was not more active than docetaxel; in contrast, in patients with germline BRCA-breast cancers, carboplatin had double objective responses than docetaxel (68% vs 33%); such a therapeutic benefit was not observed with *BRCA1* methylation, BRCA1 mRNA-low tumors, or a high score of a Myriad HRD assay [144]_._ In the neoadjuvant setting, the addition of a PARP inhibitor to carboplatin and paclitaxel did not improve the proportion of TNBC patients achieving pCR [145].

The analysis of changes that are induced by neoadjuvant chemotherapy in TNBC cells was of fundamental importance to understand the mechanisms involved in chemoresistance. Thus, several studies have shown that genomic instability, as measured through the acquisition of point mutations only modestly changed in patients displaying chemoresistance following neoadjuvant chemotherapy, whereas dynamic changes at the level of the copy number alterations, such as focal amplifications, were frequently acquired during chemotherapy [146]. This observation is largely in line with the fundamental observation that the bulk of the genomic instability of TNBCs is, in large part, related to the somatic copy number alterations that are driven by TP53 loss and emerge as rapid, punctuated bursts during disease development [147]. This conclusion was based on the analysis of individual cells in 10 TNBC patients, providing evidence that, in each of these tumors, one to three major clonal subpopulations were identified, sharing a common evolutionary lineage; furthermore, a minor subpopulation of non-clonal cells was also identified and classified as metastable, pseudo-diploid or chormazemic [147]. Philogenetic and mathematical modeling suggested that the development of tumor heterogeneity was unlikely to be developed through the gradual accumulation of copy number alterations over time, but seemingly through punctuated events, occurring early during tumor evolution: thus, each tumor evolved from founder CNAs that were concurrently acquired in the early stages of tumor evolution and stably maintained during initial tumor development; no evidence of intermediate branching was observed when the tumors progressed from diploid to aneuploid genomes; some TNBC tumors showed evidence of divergent subclones, only occurring during later stages of tumor evolution and being only characterized by the acquisition of few (one to three) CNAs [147].

Kim and coworkers performed the analysis of chemoresistance evolution by single-cell deep-exome sequencing at DNA and RNA level; using this technique, 10 patients achieving pCR were studied and 10 patients in which malignant clones persisted after treatment [148]. This analysis allowed for reaching the fundamental conclusion that resistant genotypes were pre-existing and adaptively selected by neoadjuvant chemotherapy, while the transcriptional profiles were acquired by reprogramming in response to chemotherapy in TNBC patients [148]. Particularly, this fundamental study showed two patterns of genomic and phenotypic evolution following NAC, showing two different classes of clonal dynamics: extinction, involving the elimination of tumor cells by NAC, leaving only normal diploid cells after treatment; clonal persistence, being characterized by a large number of residual tumor cells with genotypes and phenotypes that were altered by chemotherapy [148]. The genotypic changes occurring following chemotherapy are related to CNAs that emerge after chemotherapy and pre-existing to treatment; these adapting copy number alterations are responsible for transcriptional reprogramming of relapsing TNBCs [148].

In line with these observations, Hanckock et al. performed the analysis of 135 TNBC patients undergoing NAC and analyzed matched somatic genomes pre/post NAC of patients with residual disease post-chemotherapy [146]. Their main findings that were obtained through sequencing of tumors showed: (i) chaotic acquisition of copy gains and losses, including the amplification of prominent oncogenes (frequent events occurring in relapsing tumors were gains on 1q, 8q, and 10p with amplifications of *MCL1, MYC*, and *GATA3* and loss of *BRCA2*, commonly in conjunction with *FLT3* and *RB1*); (ii) absence of significant gains in deleterious mutations and insertion/deletions; (iii) gene transcriptome profiling analysis showed an enrichment of regulators of stem cell-like behavior and the depletion of immune signaling; and, (iv) analysis of the type of *TP53* alterations present in tumors showed that *TP53* loss (associated with low copy number), observed in 29% of cases, was associated with high oncogenic signaling and decreased overall survival: somatic gain in 18q were associated with poor prognosis, seemingly driving the upregulation of TGFβ signaling through SMAD2 [146].

## 7. Inflammatory Breast Cancer (IBC)

IBC is the most aggressive and lethal form of primary breast cancer, representing a clinically and pathologically unique subtype of breast cancer accounting for 2–4% of all breast cancers and it is responsible for 7–10% of breast cancer associated deaths [149]. IBC is a cancer in which cancer cells block lymph vessels in the skin of the breast. This type of cancer is called inflammatory, because the breast often looks swollen and red or inflamed. The median overall survival duration of patients with IBC at stage III is 4.75 years, as compared to 13.4 years for non-IBC breast cancers [150,151], whereas for stage IV patients is 2.27 years for IBC versus 3.4 years for non-IBC [150,151]. The diagnosis of IBC is made on the basis of a set of clinicopathologic criteria. In approximately 70% of IBCs, a dermal-lymphatic invasion is observed and the formation of tumor emboli and their tissutal invasion within the dermal-lymphatic vessels is seemingly a mechanism that contributes to the high metastatic activity of these tumors to lymph nodes and distant sites [149]. Increased cytokine-mediated infiltration of tumor-infiltrating lymphocytes or tumor-associated macrophages is another typical feature of IBC [149].

IBCs are heterogeneous for that concerns the expression of hormone receptors: HR is lower in IBC than in non-IBC (30% vs 60–80%), whereas the HER2 (40% vs 25%) and TNBC (30% vs 10–15%) subtypes are higher in IBC than in non-IBC [149]. Various studies have explored the genetic abnormalities that were observed in IBC. A first study by Ross and coworkers reported a genomic profiling analysis of 53 IBCs. These tumors display a median of five genomic aberrations/tumor; the most frequently altered genes in IBC were *TP53* (62%), *PIK3CA* (28%), *BRCA2* (15%), *PTEN* (15%) mutations, and *MYC* (32%), *HER2* (26%), and *FGFR1* (17%) amplifications [152]. Masud and coworkers studied a large set of metastatic breast cancers, including IBCs and reported that the most frequently mutated genes were *TP53* (75%), *PIK3CA* (41.7%), and *ERBB2* (16.7%) [153]. More recently, Liang and coworkers reported the analysis of genetic abnormalities on 156 IBCs by targeted next-generation sequencing [152]. The most frequently mutated genes were *TP53* (43%), *PIK3CA* (29.5%), *MYH9* (8.3%), *NOTCH2* (8.3%), *BRCA2* (7.7%), *ERBB4* (7.1%), *FGFR3* (6.4%) *POLE* (6.4%), *LAMA2* (5.8%), *ARID1A* (5.1%), *NOTCH4* (5.1%), and *ROS1* (5.1%) [154]. The analysis of the most frequently activated signaling pathways showed that DNA repair, RTK/RAS/MAPK, and NOTCH pathways for the HR^+^/HER2^-^ subgroup and DNA repair, epigenome, and diverse pathways for the HR^+^/HER2^+^ subgroup and these pathways are all significantly differently altered between IBC and non-IBC [154]. Importantly, the *PIK3CA* mutations in IBC were independently associated with negative prognosis, since the metastasis-free survival for the *PIK3CA* mutant type was 26 months and for the *PIK3CA* wild-type was 101 months: this association was true in TNBC and HR^-^/HER2^+^ subgroups, but not in the HR^+^/HER2^-^ subgroup of IBC [154].

Numerous studies have attempted to define specific features of the gene expression of IBC. However, these studies failed to define features that allow for distinguishing IBCs from other types of breast cancers and some peculiarities apparently related to IBC were just related to variations in the incidence of some subtypes, such as HER2-positive tumors, in IBCs [155].

Germline mutations were observed in 14.4% of IBC patients and they were mainly related in in 7.3% of cases to *BRCA1/BRCA2* mutations, whereas 6.3% of patients had mutations in other breast cancer-related genes, such as *PALB2, CHEK2, ATM,* and *BARD1*; finally, 1.6% had mutations in genes that were not related to breast cancer [156]. The highest frequency of germline mutations was observed among IBC cases with a TNBC phenotype [156]. Gutierrez Barrera and coworkers have explored the rate of BRCA variants among a very large group of breast cancer patients (1684 with non-IBC and 105 with IBC) and showed that BRCA pathogenic variants were found in 27.3% of patients with non-IBC and 18% of patients with IBC [157]. Interestingly, patients with IBC-BRCA-Positive tumors were diagnosed at significantly younger ages when compared to those with non-IBC BRCA-positive tumors [157].

A major limitation in the development of specific and effective treatments for IBC is related to the lack of molecular abnormalities specific for this type of breast cancer. Emerging evidences suggest that the aggressive nature of IBC could be related not to an intrinsic property of IBC tumor cells, but rather to the interaction between tumor cells displaying an autonomous cell signaling and the tumoral stroma mediating cytokine-related networks [158,159]. Particularly, the type I interferon (IFNα) pathway was shown to be upregulated in IBC and it seems to contribute to the development of tumor aggressiveness [158].

Other recent studies have shown that PDL1 is overexpressed in 38% of IBC and its expression is associated with ER^-^ status, basal, and ERBB2-enriched aggressive subtypes [160]. Interestingly, PDL1 overexpression was associated with better responses to chemotherapy [160]. In a recent study, Van Berckelaer et al. have analyzed PDL1 expression in two large cohorts of IBC patients. PDL1 expression predicted the complete pathological response to neoadjuvant chemotherapy was higher in IBC patients than on non-IBC patients, correlated with T-lymphocyte infiltration in the tumor [161]. An infiltration of T-lymphocytes of more than 10% of the stroma was a significant predictor of improved OS in multivariate analysis [161].

Hamm and coworkers have performed a parallel analysis of both genetic alterations and the infiltration of CD8^+^/PDL1^+^ lymphocytes in IBCs. Targeted NGS showed that the most frequent genetic alterations observed in IBC samples are represented by *TP53* (58%), *HER2* (amplified in 53%), *ATM* (53%), *APC* (37%), and *HER3* (26%); interestingly, *APC, ATM,* and *HER3* alterations only occur at low frequencies in non-IBC breast cancers [162]. Pathway analysis showed that IBC tumors are characterized by frequent genomic alterations in HER/PI3K/mTOR pathway [162]. A part of IBC tumors displayed a high T-lymphocyte infiltration, in association with a high mutation rate and a high immune score; a high PD-L1 staining was observed in a minority of IBCs [162].

## 8. Metastatic Breast Cancer: Genomic Alterations

In contrast to the abundance of genomic information on primary breast cancer, considerably less is known on the genomic alterations occurring in metastatic cancers.

The initial studies that were carried out on metastatic breast cancer have provided evidence that metastases are clonally related to the primary tumor, but showing, in some instances, acquired additional variants, apparently not detectable in the primary tumors. Yates and coworkers have performed a subclonal analysis of primary breast cancers through multiregion sequencing providing evidence that the extent of subclonal diversification varied among individual cases and followed spatial patterns [163]. The order of acquisition of key driver genetic alterations, such as those occurring at the level of *PIK3CA, TP53, PTEN, BRCA2*, and *MYC*, do not follow a regular temporal order, in that, in some tumors, these gene alterations are early events, while, in other ones are late events [163]. The acquisition of metastatic potential or development of drug resistance to chemotherapy arose from detectable subclones of antecedent lesions [163].

Ng et al. have explored in parallel primary tumors and metastatic lesions by high-depth whole-genome sequencing, showing that: genomic differences were observed between primary and metastatic lesions, with a median of 60% of shared somatic mutations; spatial heterogeneity was observed both at the level of primary and metastatic tumors; and, mutations involving genes affecting epithelial-to-mesenchymal transition-related genes, such as *SMAD4, TCF7L2*, and *TCF4* were restricted or enriched at the level of metastases [164].

The analysis of a group of ER^-^ breast cancer patients showed that 45% of non-synonymous somatic mutations and 55% of copy number alterations were shared between the primary tumors and the metastatic lesions; furthermore, synchronous metastases showed higher concordance with the paired primary tumor than metachronous metastases [165].

However, all of these studies were carried out on small sample sizes, thus impeding to identify general patterns of tumor evolution during relapse and metastasis. Yates and coworkers have analyzed by whole genome sequencing 17 breast cancer patients displaying three different patterns of tumor evolution to bypass these limitations: synchronous axillary lymph node metastasis; distant metastasis; and, local relapse after treatment [166]. Locoregional relapsed tumors and distant metastases are usually associated with acquisition of additional driver mutations compared with the primary tumors, while synchronous lymph nodes metastases usually display the same driver mutations that were observed in primary tumors [166].

A careful characterization of metastatic breast cancers would be of fundamental importance for several reasons, including: (i) the identification of genomic drivers responsible for metastatic disease progression; (ii) the evaluation of the impact of tumor heterogeneity at the clinical level; (iii) the identification of the main biologic determinants responsible for variability of response to therapy at the level of individual patients; and, (iv) the discovery of new therapeutic targets [167]. Ravazi et al. have evaluated the genomic landscape of endocrine-resistant advanced breast cancers in a group of patients enriched in hormone receptor-positive tumors [167]. Particularly, in a group of patients exposed to hormonal therapy, an increased number in alterations of the mitogen-activated protein kinase (MAPK) pathway and in the ER transcriptional machinery was observed [167]. Particularly, 22% of post-hormonal therapy HR^+^HER2^-^ breast cancers displayed non-overlapping alterations in one or multiple effectors of MAPK signaling or in MYC or other transcription factors; these mutations were mutually exclusive with hotspot mutations in *ESR1* [167]. The MAPK alterations are involved in the mechanism of resistance to hormonal therapy [167]. Activating *ERBB2* mutations and *NF1* loss-of-function mutations were more than twice more common in endocrine-resistant than in endocrine-responsive tumors [167]. Importantly, the global evaluation of endocrine-resistant breast cancers suggests a taxonomy categorizing these tumors into four groups: (i) tumors bearing *ESR1* mutations; (ii) tumors with molecular alterations at the level of the MAPK pathway; (iii) tumors with genetic alterations at the level of the molecular machinery involved in transcriptional regulation; and, (iv) pan-wild type tumors with an unknown mechanism of resistance [167].

Bertucci and coworkers have recently reported a detailed analysis of the genomic landscape of 617 metastatic breast cancers, being mainly represented by HR^+^/HER2^-^ (381) and TNBC (182) [168]. The most frequently altered genes were *TP53* (47%), *PIK3CA* (30%), *GATA3* (18%), *ESR1* (17%), *KMT2C* (10%), *CDH1* (9%), *PTEN* (7%), and *NF1* (7%); in the HR^+^/HER2^-^ breast cancers, nine genes were the most frequently mutated in the metastatic setting, including *TP53* (29%), *ESR1* (22%), *GATA3* (18%), *KMT2C* (12%), *NCOR1* (8%), *AKT1* (7%), *NF1* (7%), *RIC8A* (4%), and *RB1* (4%), with the *TP53* and *ESR1* genes being mutually exclusive [168]. *TP53, NF1*, and *RB1* mutations were associated with poor outcome in HR^+^/HER2^-^ patients [168]. HR^+^/HER2^-^ metastatic breast cancers also displayed an increase in mutational signatures S2, S3, S10, S13, and S17, some of which are associated with poor outcome [168]. Importantly, the metastatic tumors show a significant increase in mutational burden and clonal diversity when compared to primary breast cancers [168]. In a precedent study based on the analysis of a smaller number of metastatic breast cancer patients, Lefebvre and coworkers identified twelve genes (*TP53, PIK3CA, GATA3, ESR1, MAP3K1, CDH1, AKT1, MAP2K4, RB1, PTEN, CBFB,* and *CDKN2A*) as significantly more mutated in metastatic breast cancer than in primary tumors [169].

De Mattos-Arruda et al. profiled multiple biopsies from autopsies of 10 patients with therapy-resistant breast cancer [170]. The analysis of copy number alterations showed that the CNA profiles were remarkably similar in 9/10 cases; only in 1/10 cases, some metastasis-specific CNAs were observed in some metastatic sites, but all metastases derive from a common ancestor with 1q gain and 16q loss [170]. The analysis of the mutational profile of metastases by wide exome sequencing showed that these mutations can be classified as “metastatic stem” when present in all metastases from a single case, “metastatic clade” when present in at least two but not all metastases and “metastatic private” when present in a single metastasis [170]. Metastatic stem and metastatic clade were the most frequent types of mutations; metastasis stem mutations were present in all cases, with a number from 1 to 9 [170]. In conclusion, the studies on the metastatic mutations showed that metastases exhibit the tendency to accumulate mutations, with an apparent hierarchy of expression (stem-clade-private) and, as more mutations accumulate in metastases, these are increasingly passengers [170]. A part of mutations observed in metastases, including driver mutations shared across metastases, are not detectable in primary tumors, suggesting either their acquisition in metastatic cells or their derivation from very minority clone present in primary tumors [170]. The analysis of variant allelic fraction (VAF) of stem and clade mutations across all metastases indicates that metastases are initiated and maintained as groups of cellular clones [170].

A recent study showed that whole-genome doubling (WGD) is an event that frequently occurs in cancers at an advanced stage of development [171]. WGD occurs in >35% of advanced breast cancers; particularly, in ER^+^ breast cancers the presence of WGD predicts an increased morbidity and it is associated with negative outcome [171].

Angus and coworkers have recently reported a whole-sequencing study (WGS) on 625 patients with metastatic cancer, providing another fundamental contribution to the analysis of the molecular abnormalities related to the metastatic status of this tumor [47]. When compared to the WGS from 560 primary breast cancers [48], the median number of single nucleotide variants in metastatic cancers was significantly higher than in primary tumors; similarly, tumor mutational burden was higher in metastatic than in primary tumors [47]. The increase of TMB was related to both the progression to the metastatic condition and treatment exerting a pressure in tumor evolution [47]. The analysis of mutational signatures that are present in primary breast cancer contributes to the observed increased TMB in metastatic breast cancer; particularly, a shift from a more indolent age-related mutagenesis in primary cancers toward more APOBEC-driven processes in metastatic breast cancer was observed [47]. Using the ratio of non-synonymous to synonymous mutations, 21 potential driver genes were identified in metatstatic breast cancers, including the key drivers *TP53, PIK3CA, ESR1, GATA3,* and *KMT2C*; as expected, *TP53* was enriched in TNBC, whereas *ESR1, PIK3CA*, and *GATA3* were more frequently mutated in ER^+^ metastatic breast cancers [47]. The comparison of the mutational frequency of these driver genes between primary and metastatic tumors showed that, in ER^+^ metastatic breast cancers, the mutational frequency of *ESR1, TP53, NF1, AKT1, KMT2C*, and *PTEN* genes was higher than in primary breast cancers [47]. Finally, this study allowed for defining a number of molecular events in metastatic breast cancers that could be targeted by specific treatments: 11% of patients displayed a high TMB that represents a biomarker to select patients for immunotherapy; 9% of patients exhibit a homologous recombination deficiency and they are thus candidate for poly-ADP ribose polymerase inhibitors and/or double-stranded DNA break-inducing chemotherapy; specific genomic alterations, for which the FDA-approved drugs are already available, were observed in 24% of patients [47].

As above-mentioned, the ESR1 gene, the gene encoding the estrogen receptor α is frequently mutated in metastatic breast cancer. Aromatase inhibitors (AIs) block the conversion of androgens to estrogens and represent the first-line of treatment for post-menopausal women with ER^+^ breast cancer. The mutations in ESR1 are mainly observed in breast cancer patients who progress on AI therapy, but less frequently in patients who have not received AI for metastastic disease. Toy and coworkers reported *ESR1* mutations in 14/80; these mutations occur in the ligand binding domain and favoring receptor’s agonist conformation and spontaneous activity during ER-dependent transcription and proliferation in the absence of hormone, reducing the efficacy of ER antagonists [172]. The most common mutations have been functionally characterized as hormone-independent activating mutations (Y537, D538). In a more recent study, Toy et al. analyzed 929 breast cancer samples and observed a frequency of *ESR1* mutations of 3.5% in primary tumors and 13.6% in metastatic tumors; *ESR1* mutations were found in ER^+^ tumors, but not in TNBC [173].

Robinson and coworkers studied 11 patients with ER^+^ metastatic breast cancer and observed that six of them harbored mutations affecting the ESR1 ligand binding domain [173]. All of these *ESR1* mutants displayed constitutive activity [174].

In addition to coding mutations, Bailey and coworkers reported somatic mutations at the level of the set of regulatory elements (SRE) regulating *ESR1* expression; these mutations regulate *ESR1* expression by modulating the transcription factor binding to the DNA [175].

In a population of 21 patients with recurrent breast cancer under adjuvant hormone therapy, *ESR1* mutations were observed in 19% of patients, while using digital droplet PCR technique on plasma DNA samples [176].

Spoerke and coworkers have characterized ESR1 mutations in a group of ER^+^ metastatic breast cancer patients who have failed previous therapy with AIs and randomized to receive either the pan-PI3K inhibitor pictilisib plus fulvestrant (a selective estrogen receptor degrader) or placebo plus fulvestrant [177]. In this study, *ESR1* mutations were observed in 37% of baseline tumor samples: these mutations are clearly enriched in luminal A (72%) and *PIK3CA*-mutant (60%) patients [177]. Interestingly, 40% of *ESR1*-mutant patients contemporaneously displayed 2 or more *ESR1* mutations [177]. The presence of *ESR1* mutations was not associated with a clear difference in the risk of progression in fulvestrant therapy [177]. Fribbens and coworkers performed a similar study, providing evidence through analysis of ESR1 mutations in the context of two clinical studies, SOFEA (study of Faslodex versus Examestane with or without Arimidex) and PALOMA3 (Palbocilcib combined with Fulvestrant in hormone receptor-positive, HER2-negative metastatic breast cancer after endocrine failure) trials that *ESR1* mutations, present in 39% (SOFEA trial) or 25% (PALOMA3 trial) of patients, improved (SOFEA trial), or did not change the response to Fulvestrant [178].

Lopez-Knowles et al. have explored 48 ER^+^ breast cancer patients before treatment and at progression on AI therapy. Some patients displayed detectable *ERS1* mutations, which were only present in the secondary sample. The analysis of gene expression showed that, in *ESR1*-mutated AI-resistant tumors, the expression of four classical estrogen-regulated genes was sevenfold higher than in ESR1 wild-type tumors [179].

The majority of the recurrent ESR1 mutations were found to be located in the ER binding domain (LBD); these mutations stimulate constitutive activity in the absence of estrogen and decreased sensitivity to ER antagonists, thus suggesting that these gain-of-function mutations are drivers of endocrine resistance [172]. A recent study showed that *ESR1*-LDB mutations, in addition to promoting ligand-independent growth, also promote a more aggressive phenotype, leading to changes in ER transcriptional network that mediate cancer progression [180]. Importantly, the CDK7 inhibitor THZ1 blocks mutant ER phosphorylation at S118 and inhibits mutant ER cell growth [180]. In line with the idea that *ESR1*-LBD mutants confer increased the malignancy to breast cancer cells, Zinger et al. showed that these mutations rewire cellular metabolism of breast cancer cells in that induce a shift of energetic metabolism from a glucose dependency to a metabolic condition where the cancer cells do not strictly depend on glucose and develop the capacity to utilize glutamine as an alternative carbon source [181].

The selection of *ESR1* mutations can occur not only in metastatic disease, but also in primary disease, as supported by a recent study investigating *ESR1* mutations in primary breast cancers that were treated with AIs for more than six months [182]. *ESR1*-mutant tumors exhibited an increased expression of ESR1 transcript and limited suppression of estrogen-regulated genes and proliferation-associated genes in response to AI treatment [182].

A recent study showed that copy-number alterations of *ESR1* and key CDK pathway genes are frequent (for *ESR1* 13%) in metastatic breast cancer and their clinical significance must be carefully evaluated in future studies [183].

The treatment might affect the genomic landscape of relapsing/metastatic breast cancer. Various studies support this conclusion. Thus, Juric and coworkers discovered loss of PTEN in a patient with a PI3KCA-mutant breast cancer undergoing treatment with the PI3K inhibitor BYL719 and developing resistance to this drug [179]. The examination of six additional patients undergoing treatment with BYL719 showed biallelic *PTEN* loss in one of these patients and the loss of *PI3KCA* mutation in two of these patients [184]. *PTEN* loss seems to also be involved in the mechanism of resistance to CDK4/6 inhbitors. Thus, Costa and coworkers explored a group of ER^+^ breast cancer patients undergoing treatment with the combination of CK4/6 inhibitors with the anti-estrogen letrozole and observed that, in patients developing resistance to this treatment, some tumors acquired *RB1* loss, whereas other tumors lost *PTEN* expression [184]. Interestingly, *PTEN* loss induced the exclusion of p27 from the nucleus, with consequent increased activation of CDK4 and CDK2 [185].

FGFR aberrant signaling that was observed in ER+ breast cancer patients undergoing treatment with CDK4/6 inhibitors plus estrogen inhibitors represents another mechanism of tumor resistance. Thus, Formisano et al. have performed a next generation sequencing of circulating tumor DNA in 34 patients after the progression on CDK4/6 inhibitors and have identified *FGFR1/2* amplification or activating mutations in 41% post-progression specimens [186]. Drago and coworkers have explored the mechanisms of resistance to endocrine and targeted therapies in a cohort of ER^+^/HER2^-^ breast cancer patients and have shown that patients with *FGFR1*-amplified tumors: (i) have a tendency to a more frequent PR^-^ disease (47% vs 20%) and coexisting *TP53* mutations (41% vs 21%); and, (ii) display a shorter time to progression disease following endocrine therapy alone or in combination with a mTOR inhibitor [187]. Lucitinib, an inhibitor of FGFR1-3, VEGFR1-3, and PDGFRα/β, was recently tested in a group of metastatic ER^+^/HER2^-^ breast cancer patients, showing only modest antitumor activity and significant clinical toxicity; exploratory biomarker analyses suggested that patients with *FGFR1* amplification or expression may derive a greater benefit from treatment [188].

Other recent studies focused on the identification of *NF1* mutations in advanced breast cancers and in the definition of their potential role in the mechanisms of resistance to endocrine therapy. Thus, Pearson et al. have sequenced 210 samples of advanced breast cancers (including 63% of HR^+^/HER2^-^ tumors, 12% HER2^+^ and 17% TNBC), and observed enrichment when compared to primary tumors, for mutations of *HER2* (6,2%), *AKT1* (7,1%), and *NF1* (8,1%). Interestingly, *NF1* mutations were frequently acquired in the disease progression, not being present in the original primary disease [189]. In ER^+^ breast cancer models, *NF1* loss induced resistance to endocrine therapy through ER-dependent and ER-independent mechanisms [189]. *NF1* loss induced cyclin D1 expression and patients with *NF1* mutations at baseline had good response to treatment with CDK4/6 inhibitor palbociclib [189].

The monitoring of plasmatic tumor DNA in endocrine-resistant breast cancer patients showed that 8.4% of these patients developed novel *HER2* mutations; interestingly, the treatment of one *HER2*-mutant patient with the HER2 irreversible inhibitor Neritinib resulted in a remarkable clinical response, with the disappearance of two tumor clones bearing *HER2* mutations, but the persistence of other passenger subclones [190].

Bone metastases are a significant cause of morbidity in patients with ER^+^ breast cancer. The analysis of these metastases is difficult for obvious technical problems. A recent study reported a comparative analysis of gene abnormalities and the gene expression profile of 11 matched samples of primary tumors and matched bone metastases. This analysis showed, at the level of bone metastases, some relevant changes in the CDK/Rb/E2F and FGFR pathways [191]. Brain metastases occur in 10–15% of patients with metastatic breast cancer and they represent a great clinical problem for the short survival of patients due to the lack of specific therapies. Only HER2-positive brain metastases can be treated with HER2 inhibitors. The analysis of 20 brain metastases and their matched primary tumors showed changes of the expression in actionable genes, such as gains of *FGFR4, FLT1, AURKA*, and the loss of *ESR1*; the most recurrent expression gain was at the level of *HER2*: in some instances, HER2-negative primary tumors acquired an elevated *HER2* expression in brain metastases [192]. This last observation is of potential clinical interest, because it might offer some therapeutic opportunities to these patients [192].

## 9. Invasive Lobular Breast Carcinoma (ILC)

ILC is the second most frequent histologic subtype of breast cancer, representing 10–15% of all breast cancers. ILC is a type of breast cancer that begins in the milk producing glands (lobules) of the breast. ILCs are typically ER and/or PR-positive. Several variants of ILC have been identified, including *classic ILC*, *solid ILC*, *alveolar ILC*, *tubule-lobular ILC*, and *pleomorphic ILC*. The common molecular feature of all ILC variants is the loss of the epithelial cell-cell adhesion molecule E-Cadherin encoded by CDH1 gene [193].

The dysregulation of cell-cell adhesion that is driven by lack of E-Cadherin protein is responsible for the histologic feature of ILC, consisting in small neoplastic discoadhesive neoplastic cells invading the stroma in a single-file pattern. Genetic mutations causing a loss-of-function have been identified in all variants of ILC, with a frequency that ranges from 50% to 60% [194]. Mutations of the *CDH1* gene are usually associated with the loss of 16q, the chromosome region where this gene is located [195]. The lack of E-cadherin protein expression is observed in approximately 90% of ILCs, confirming the highly non-cohesive morphological characteristics of this type of tumor. A recent study explored, in detail, the frequency of absent E-cadherin protein expression in tumors that can be classified as ILCs: about 4.5% of these tumors show E-cadherin expression, whereas 95.5% of these tumors display absent E-cadherin expression [196]. At variance, in breast cancers that were classified as invasive mammary with mixed ductal/lobular features, E-cadherin expression was absent in 40% of cases, whereas the remaining cases display E-cadherin expression with a heterogeneous pattern of positivity at the histological level [196]. Some rare breast cancers have CDH1 alterations and exhibit IDC morphology [196].

Recent studies that are based on next generation sequencing have elucidated the molecular abnormalities of ILCs. Ciriello and coworkers have analyzed 127 pure ILCs and 88 mixed ILCs/IDCs and then compared them to 490 ductal breast cancers (IDSc) and showed mutations targeting *PTEN/FOXA1* and *TBX3* as ILC-enriched features [197]. *PTEN* loss that is associated with increased AKT phosphorylation, which was highest in all ILC variants; interestingly, alterations acting upstream of AKT were identified in 40% of ILC cases and they were associated with increased AKT phosphorylation [197]. FOXA1, which is a key modulator of transcriptional activity of ER, controlling ER DNA binding through a modification of chromatin accessibility, is mutated in approximately 3% of IDCs and 7% of LICs [197]. Analysis of mRNA expression data allowed for classifying ILCs into three ILC subtypes, termed reactive-like, immune-related, and proliferative [197].

Desmedt et al. performed the genomic characterization of a large set of ILCs providing evidence that: (i) *CDH1* was mutated in 65% of tumors; (ii) alterations in one of the key genes of PI3K pathway, *PIK3CA, PTEN*, and *AKT1* were observed in more than 50% of cases; (iii) *HER2* and *HER3* were mutated in 5.1% and 3.6% of the tumors, respectively (these mutations activate EGFR/ERBB pathway); and, (iv) mutations in *FOXA1* and *ERS1* copy number gains were detected in 9% and 25% of cases, respectively [197]. All these alterations were more frequent in ILCs than in IDCs [198]. The histologic diversity of ILCs was associated with specific genetic alterations: (i) enrichment of *HER2* mutations in the mixed, non-classic ILC variant; and, (ii) *ERS1* gains in the solid ILC variant [198]. The *AKT1* mutations were associated with an increased risk of early relapse [198].

Michaut et al. have performed a comprehensive genomic, transcriptomic, and proteomic analysis on a large cohort of ILC patients [198]. This study confirmed that mutations in the *CDH1* and PI3K pathway are the most frequent molecular alterations in ILC and identified two main subtypes of ILCs: (i) an immune-related subtype with elevated PD-L1 and CTLA-4 expression and greater sensitivity to DNA-damaging agents; a hormone subtype, which is associated with higher ER and PR expression, upregulation of cell-cycle genes, epithelial to mesenchymal transition, and gain of chromosomes 1q and 8q and the loss of chromosome 11 [199].

*ESR1* mutations may occur in breast cancer patients as above discussed. A recent study explored the occurrence of *ESR1* mutations in matched primary and metastatic ILCs, providing evidence that the mutations of this gene were observed in 6% of primary tumors and 15% of metastatic samples, with 9% of patients harboring *ESR1* mutations in metastatic, but not in the corresponding primary tumors [200]. The comparison of ILCs and IDCs did not show any significant difference in the frequency of these mutations and in the type of mutations, with the D538G mutation being the most frequent [200]. Cao and coworkers recently reported frequent *ESR1* amplifications [201]. In fact, these authors have explored copy number alterations occurring in ILCs and observed frequent amplification of *CCND1* (33%) and *MYC* (17%) and *ESR1* and *ERBB2* amplifications in 19% of patients; tumors with copy number alterations are more likely to recur when compared to those with normal copy number [201]. Finally, the *MDM4* gene was amplified in 17% of ILC cases [201].

A recent study showed that *FGFR4* is hyperexpressed in metastatic breast cancer, when compared to primary tumor, both in IDC (2.4 fold) and ILC (4.8 fold). *FGFR4* hot spot mutations are enriched in ILCs, as compared to IDCs: in metastatic IDCs, *FGFR4* mutations range from 0 to 0.5%; in metastatic ILCs, the *FGFR4* mutations range from 1.5 to 10% [202]. These observations suggest a potential utility of FGFR4 blockade with small molecule inhibitors in ILCs.

ILCs (ER^+^/HER2^-^) showed immune cells infiltration lower than ER^+^/HER2^-^ IDCs [203]. Furthermore, in ILCs, high TIL levels were associated with young age, lymph node involvement, and negative prognosis.

Mixed ductal-lobular (MDL) carcinomas show both ductal and lobular morphology and they constitute an example of intratumoral morphological heterogeneity. A recent study provided an analysis of 82 MDLs, showing data that support a model in which the two different components of these tumors arise from a common ancestor and lobular morphology can arise from a ductal pathway of tumor progression. In MDLs that present in association with lobular carcinoma in situ (LCIS) and ductal carcinoma in situ (DCIS), the clonal divergence between the two lineages occurs early and it is associated with the complete loss of E-Cadherin expression, whereas, in most of MDLs, which present in association with DICs and not LCIS, clonal divergence of the lobular phenotype occurs late during tumor evolution [204]. It was hypothesized that the mechanism driving the phenotypic change might involve aberrant E-Cadherin expression (in fact, in the lobular components of MDLs, E-Cadherin is aberrantly located in the cytoplasm) [204].

Recent studies have investigated the molecular alterations of LCIS, a preinvasive lesion of the breast often multifocal and bilateral; however, growing evidence suggests that LCIS is a non-obligate precursor of invasive breast cancer [205]. A study of Begg and coworkers that was based on the analysis of LCIS lesions derived from 30 patients suggested a clonal relationship between LCIS and invasive lobular carcinoma [206]. More recently, Lee and coworkers have explored 43 LCIS and 27 synchronous more advanced lesions from 24 patients [207]. Whole-exome sequencing showed that LCIs display genetic abnormalities that are similar to those that were reported for ILCs, with *CDH1* mutations present in 81% of the lesions; 42% of LCIS were found to be clonally related to synchronous DCIS and/or ILCs, with clonal evolutionary patterns supporting clonal selection and/or parallel/branched progression [207]. Interestingly, intralesional genetic heterogeneity was higher in LCIS clonally that was related to DCIS/ILC [207]. Tumor progression was accompanied by a shift from an aging signature to an APOBEC-related mutational signature [207].

Few studies have extensively explored the genetic alterations that were observed in metastatic ILCs. Sokol and coworkers have reported the analysis of 180 metastatic ILCs and showed that the most common genetic alterations in these samples were mutations of *CDH1* (77%), *PIK3CA* (53%), *TP53* (24%), the co-amplified 11q13 locus genes *CCND1* (22%), *FGF19* (21%), *FGF4* (19%), *FGF3* (19%), and *ESR1* (17%) [208]. Interestingly, other than the *CDH1* gene (77% vs 6.8%), three genes were enriched in metastatic ILCs versus metastatic IDCs: *NF1* (12.2% vs 3.1%), *TBX3* (12.8% vs 3.7%), and *PIK3CA* (53% vs 39.8%) [208]. NF1 alterations are predominantly the loss of heterozygosity, are mutually exclusive with *ESR1* mutations and they are frequently polyclonal [208]. Studies of paired specimens indicate that NF1 alterations arise in patients developing acquired resistance to endocrine therapy [208].

Recent studies have explored, in experimental models, the role of E-Cadherin mutation on the pathogenesis of ILCs. A first study by An and coworkers explored the role of *CDH1* mutations on the genesis of immune-related ILC [209]. As discussed above, the immune-related subtype of ILC was identified and defined by the overexpression of transcripts coding for interleukins, chemokines, and cytokines, as well as by gene expression that is linked to lymphocyte and macrophage function [197,199]; genomic studies showed a very high frequency of *CDH1* loss-of-function and *PIK3CA* gain-of-function mutations [197,199]. The deletion of *CDH1*, together with the activation of *PIK3CA* in mammary epithelium of genetically modified mice, leads to the formation of immune-related-ILC-like tumors with immune infiltration [209]. The study of this model suggests a possible therapeutic intervention with immune checkpoint inhibitors [209]. Another study provided evidence that E-Cadherin loss induces a targeatable autocrine activation of growth factor signaling in ILC: this autocrine mechanism, also observed in primary ILC samples, leads to PI3K/AKT activation and inhibitors of these two signaling pathways exert a pronounced inhibition of cell growth and survival of ILC cells [210].

The prognostication of ILCs is currently performed according to clinic-pathologic criteria, including tumor size, grade, and lymphnode status and immunohistochemistry detection of ER, PR, and HER2. The current molecular signatures do not offer any prognostication evaluation of ILCs. A recent study used an integration analysis of gene expression and copy number data as a tool to identify genes that influence ILC behavior and prognosis [211]. Using this combination approach, it a 194-metagene signature was developed, which was termed LobSig, and it possesses a prognostic value in ILC patients [211]. Particularly, LobSig status predicts outcome in ILC patients with an accuracy of about 95% [211]. Interestingly, a high LobSig status was associated with TCGA proliferative subtype [211]. Finally, ILCs that were predicted as having a negative outcome according to LobSig-high phenotype were associated with mutations in *ERBB2, ERBB3, TP53, AKT1,* and *ROS1* [211].

As above-indicated, ILC has unique clinical, pathologic, and biologic features, which suggests that it can be considered to be a distinct clinical entity; however, ILCs continue to be treated as well as IDCs and ILC patients are currently included in studies involving IDC patients and treated as well as HR^+^/HER2^-^ IDC patients [212]. Few clinical studies have selectively investigated the response of ILC to standard treatment anti-breast cancer. Neoadjuvant treatment could be considered to be a preferential therapeutic approach in view of the frequently locally advanced stage of these tumors [212]. Several studies have explored the potential therapeutic impact of neoadjuvant chemotherapy in ILC patients; however, these studies have shown that the response rates to neoadjuvant chemotherapy are lower for ILC than for IDC, as supported by the reduced rates of pathologic response and breast conservation [213].

A very limited number of studies have specifically evaluated neoadjuvant hormonal therapy for lobular carcinoma. Neodjuvant endocrine therapy has the advantage when compared to neoadjuvant chemotherapy consisting in a lower toxicity, but the former has the inconveniency with respect to the latter one to require longer time for the development of a clinical benefit [214]. Dixon et al. have explored the response of 61 patients with ILC to neoadjuvant letrozole, showing a mean tumor volume reduction of 66%, with a high rate of breast conservation (81%); however, in this study, there was no comparison between neoadjuvant endocrine therapy and neoadjuvant chemotherapy [214]. Interestingly, a recent study by Thornton and coworkers reported the analysis carried out on 5942 ILC patients (with node-positive disease) undergoing either neoadjuvant endocrine therapy or neoadjuvant chemotherapy [215]. A similar overall survival was observed in patients undergoing neoadjuvant endocrine therapy or neoadjuvant chemotherapy after adjustment for clinicopathologic variables present in the two groups of patients [215]. Other studies have explored hormonal therapy in an adjuvant setting. In this context, particularly interesting was a study by Metzger Filho et al., who compared the relative efficacies of tamoxifen and letrozole for lobular and ductal carcinomas in the context of the BIG 1-98 clinical trial [216]. Patients with ILCs displayed more benefits from letrozole than tamoxifen, regardless of whether patients pertain to luminal A or luminal B-like subtype [216]. The eight-year disease-free survival was 66% for tamoxifen when compared to 82% for letrozole in the ILC group and it was 75% for tamoxifen and 82% for letrozole in the IDC group [216]. The clinical study MA-27 compared five years of adjuvant anastrozole or exemestane in postmenopausal patients with hormone receptor positive early breast cancer [217]. In this study were enrolled 5709 patients, 5021 with IDC and 688 with ILC: ILC patients had a better overall survival when treated with anastrozole versus exemestane, whereas no difference was noted for patients with IDC [217]. Finally, a recent study retrospectively investigated 379 patients with early stage ILC undergoing adjuvant treatment with endocrine therapy or chemotherapy; no differences in disease-free survival between endocrine monotherapy and chemotherapy were observed [218].

A large part of ILCs are treated with endocrine therapy, but about 30% of these tumors are *de novo* resistant to therapy. Walsh and coworkers have explored the potential mechanisms of resistance of these tumors to endocrine therapy and discovered that the high expression of the epigenetic reader, bromodomain protein 3 (BRD3) was associated with poor recurrence-free survival [219]. The ILC cell lines resulted in being sensitive to JQ1, and inhibitor of BET family proteins; ILC cells resistant to JQ1 overexpress the anti-apoptotic protein Bcl-X_L_ or the FGFR 1–4 [219]. These observations suggest some therapeutic opportunities for ILCs.

It was identified the cortactin gene as a candidate ILC driver, exhibiting higher expression in tumors from patients with recurrent disease versus non-concurrent disease, while using the nanostring platform to measure the expression of copy number variation-associated genes in ILC tumors with available data on their long-term outcome [220]. Another study showed that the interaction of the surface protein ROR1 expressed in breast cancer cells interacts with cortactin and plays an important role in breast cancer cell migration and metastasis [221].

## 10. Mucinous Breast Cancer

Mucinous carcinoma of the breast is a rare subtype of breast cancer, which corresponds to approximately 2% of all breast carcinomas. This is a rare form of invasive ductal carcinoma, which is also called colloid carcinoma. According to the WHO classification of breast tumors, mucinous breast cancer is a special type of breast cancer. Mucinous breast cancers are subdivided into two subtypes, according to histo-morphological features: the pure type of mucinous carcinoma (the tumor tissue is exclusively of mucin-producing cells) and the mixed type of mucinous carcinoma (the tumor tissue consists of both mucin-producing and ductal epithelial without mucin regions). At the histological level, mucinous breast carcinoma is characterized by nests of cells that are floating in lakes of mucin that is portioned by fibrous septae containing capillary blood vessels. These tumors usually occur in postmenopausal women, have a good prognosis, and are positive for ERs and PRs. Several studies have characterized the molecular features of mucinous breast cancer. Mucinous breast carcinomas can be subdivided into two subgroups of tumors: mucinous A (hypocellular variant) and mucinous B (hypercellular variant), according to criteria that were initially described by Capella and coworkers [222]. Mucinous B carcinomas display histological features of neuroendocrine differentiation and exhibit transcriptional profiles different from those observed in mucinous A tumors: in fact, although both mucinous A and B tumors correspond to luminal A subtype, the transcriptional profile of mucinous B carcinomas is similar to that of neuroendocrine breast cancers, while mucinous A and neuroendocrine cancers have different transcriptomic profiles [223]. It was suggested that mucinous B and neuroendocrine carcinomas are part of a spectrum of different lesions, while mucinous A carcinomas represent a distinct, discrete entity, according to these findings [223].

Many studies support the view that mucinous carcinoma of the breast is genomically distinct from invasive ductal carcinomas. Thus, Lacroix-Triki in an initial study showed that pure mucinous breast cancers express in 100% of cases ER, have a low level of genetic instability and less frequently harbor gains of 1q and 16p and losses of 16q and 22q than grade and ER-matched invasive ductal carcinomas, while no pure mucinous breast cancers display concurrent 1q and 16q loss [224]. In line with these initial observations, Kehr and coworkers reported the absence of *PIK3CA* and *AKT1* mutations in mucinous breast cancer [225]. Ross and coworkers have analyzed the most frequent copy number alterations that were observed in mucinous breast cancers and showed that: *FGFR1* was more frequently amplified in mucinous than non-mucinous ER^+^ breast cancer (36% vs 11%); the *FGF3/FGF4/FGF19* amplicon was frequently amplified (27%) [226].

Recently, the genomic landscape was investigated in 32 cases of pure mucinous breast carcinomas, confirming and extending the findings of previous studies: *GATA3* (about 24%), *KMT2C* (19%), and *MAP3K1* (about 14%) were the most frequently mutated genes; *OA21-CSNK12G* and *RFC4-LPP* fusions were observed in 9.7% and 6.5% of cases; *TP53* and *PIK3CA* mutations were less frequent than in ER^+^/HER2^-^ intraductal carcinomas [227] (Figure 4). Furthermore, remarkable mutational differences were observed between the pure and mixed mucinous breast cancers [227] (Figure 4). A second recent study analyzed the genomic, transcriptomic, epigenetic, and immune profiling of mucinous breast cancer, showing that mucinous breast cancers were characterized by low genomic instability, decreased prevalence of *PIK3CA* mutations and low tumor infiltrating lymphocyte levels (lymphocyte infiltration in these tumors could be hindered by extracellular mucin acting as a barrier to immune recognition) [228]. Finally, this study identified aberrant DNA methylation of *MUC2* as a possible cause of extracellular production of mucin in mucinous breast cancer [228].

Interestingly, a recent study provided evidence for significant familial clustering of mucinous breast carcinomas, but the genetic variants responsible for the increased risk are unknown [229].

The micropapillary variant of mucinous carcinoma of the breast (MPMC) is a rare form of ER-positive invasive carcinoma. MPMCs show hybrid mucinous and micropapillary features, consisting of micropapillary clusters of tumor cells embedded in a lake of extracellular mucin. In contrast to pure mucinous tumors that usually have an indolent course, MPMCs are associated with a high rate of nodal metastases and outcomes intermediate between those that were observed in patients with pure mucinous carcinoma and invasive micropapillary breast cancers [230]. Whole genome sequencing studies showed that: (a) no mutations in genes significantly mutated in breast cancer, including *TP53, PIK3CA, GATA3*, and *MAP3K1* were detected; (b) copy number alterations that have been reported in micropapillary carcinomas, such as recurrent gains of 1q, 6p, 8q, and 10q, recurrent losses in 16q, 11q, and 13q, as well as a recurrent 8p12–8p 11.2 amplification encompassing *FGFR1* were detected [231]. It is important to point out that MPMCs are heterogeneous for that concerns their copy number alterations and the ensemble of the genetic characterization study suggests that these tumors do not harbor pathognomic or highly recurrent genetic alterations [231]. MPMCs might represent in some instances mucinous carcinomas that acquired micropapillary features, but in other instances could correspond to micropapillary carcinomas that acquired mucin production capacity [231].

## 11. Micropapillary Carcinoma of the Breast

Micropapillary carcinomas (MPCs) are a group of breast carcinomas, which are characterized by a unique histological pattern that consists of clusters of tumor cells with an inverted polarity, immersed in a spongy stroma; these tumors represent about 5–7% of all invasive breast cancers. High-resolution micro-array comparative genomic hybridization analysis provided evidence that MPCs have distinct molecular genetic profiles compared to invasive ductal carcinomas of no-special type [232]; furthermore, pure and mixed MPCs display similar genetic alterations [233]. The characterization of the genomic features of 13 MPCs showed that the gains of 1q, 8q, 12q, and 20q and the losses of 1p, 8p, 13q, 16q, and 20q were more prevalent in MPCs than they were in ICD-NOS; recurrent mutations affected genes of the mitogen-activated protein kinase family (*MAP3K1, MAP2K6*, and *MAP3K4*); a constellation of the mutations found in MPCs largely corresponds to those typically observed in luminal B cancers (*PIK3CA, TP53, ATRX,* and *CSMD2*) [234]. RNA sequencing studies revealed the existence of fusion genes in MPCs: of these fusions, two were predicted to be in frame (*SLC2A1-FAF1* and *BCAS4-AURKA*) and they were present in a single tumor [222]. Whole exome sequencing studies confirmed that MPCs exhibit a mutational profile comparable to common breast cancers with *PIK3CA, TP53, GATA3*, and *MAP2K4* being the genes most frequently mutated [235].

Recently, Lewis and coworkers reported the analysis of the molecular status of a very large group (865 patients) of MPCs showing that 75.3% displayed HR^+^/HER2^-^ disease; 14.8% HR^+^/HER2^+^ disease; 4.7% HR^-^/HER2^+^ disease; and, 5.2% had TNBC disease [236].

The outcome of MPCs was compared to that of classical IDCs and, in spite some early studies, suggested a negative outcome of MPCs, a recent study provided clear evidence that MPCs displayed no statistically different survival when compared to IDCs [237].

## 12. Pleomorphic Lobular Carcinoma

Pleomorphic ILC (PILC) is a recently recognized variant of ILC that represents approximately 10% of lobular tumors. PILCs share many histological and biologic properties with classic ILC, particularly for that concerning its infiltrative growth pattern, lack of E-Cadherin expression, and common chromosomal alterations. However, some features allow for distinguishing PILC from ILC, such as a greater degree of cellular atypia and nuclear pleomorphism.

Several recent studies have characterized PILCs at molecular level, showing some features distinguishing PILC from ILC. In this context, Chen et al. reported the analysis of 31 cases of PILC (13 apocrine and 18 non-apocrine subtypes); at histological level, these tumors displayed dishesive cellular elements, with pleomorphic nuclei; at immunohistochemical levels, when compared with ILCs, PILCs showed a significantly higher Ki67 index, lower ER and PR expression, and higher incidence of HER2 amplification; at the molecular level, PILCs and ILCs both displayed frequent loss of 16q and gain of 1q [238]. Interestingly, apocrine (characterized by cells with abundant eosinophilic cytoplasm, cytoplasmic granules, and prominent nucleoli) PILCs exhibited significantly more genomic alterations than non-apocrine PILCs and classical ILCs [238]. Rosa-Rosa and coworkers reported the molecular analysis of 39 tumor samples of PILC showing that: all cases showed the absence or aberrant expression of E-Cadherin and abnormal expression of β-catenin and p120; *CDH1* (89%), *PIK3CA* (33%), and *HER2* (26%) were the most recurrently mutated genes; a higher frequency of mutations in *ARID1B, KMT2C, MAP3K1, TP53* and *ARID1A* was observed in PILC than previously reported in classic ILC [239]. The *HER2* mutations preferentially affected the tyrosine-kinase activity domain and are targetable with available drugs [238].

Zhu and coworkers recently reported the characterization of molecular alterations that could provide an understanding of the mechanisms that are responsible for the aggressive behavior of PILC. Targeted sequencing analysis identified the genes that were most frequently mutated in PILC and showed that these tumors are more similar to classic ILCs than to IDCs [240]. The frequency of molecular alterations in genes discriminating between lobular and ductal breast cancers, such as *TP53, MYC, GATA3, FOXA1, CDH1,* and *TBX3*, was more similar to that reported for ILCs than for IDCs [240]. Interestingly, some recurrent molecular alterations distinguish PILCs from classic ILCs: this is the case of the *IRS2* gene, encoding the IRS2 adaptor protein mediating signaling downstream of the insulin and IGF-1 receptors and frequently mutated in PILCs (29% of cases); functional studies supported an important role for Insulin Receptor and IGF-1-R signaling pathways in promoting PILC development and aggressive behavior [240].

## 13. Metaplastic Breast Cancer

Metaplastic breast cancer (MpBC) is a rare and peculiar malignancy that accounts for approximately 0.2–5% of all breast cancers. The typical feature of MpBC is represented by the differentiation of the neoplastic mammary epithelium to a non-glandular component, usually squamous or mesenchymal. According to their differentiation properties, these tumors have been classified as squamous cell carcinoma, spindle cell carcinoma, mixed squamous and spindle, mesenchymal, or spindle cell and mesenchymal. Hormone receptor markers are negative in these tumors that can be classified as TNBCs. Gene expression profiling studies have shown that MpBCs pertain to the claudin-low subtype.

Recent studies have clarified the peculiarities of the genetic alterations that were observed in MpBCs. Initial studies of genomic analysis using array comparative genomic hybridization showed that MpBCs display greater genomic instability than other invasive breast cancers and they also show a high frequency of *PIK3CA* mutations, a finding suggesting that these tumors are different from other TNBCs, showing a low frequency of PI3KCA mutations [111]. Whole-exome sequencing studies provided evidence that MpBCs harbored frequent *TP53* (69%) *PIK3CA* (29%), *PIK3R1* (11%), *ARID1A* (11%), *FAT1* (11%), and *PTEN* (11%) mutations; when compared with standard TNBCs, MpBCs significantly more frequently displayed mutations in PI3K/AKT/mTOR pathway (57% vs 22%) and canonical Wnt pathway (51% vs 28%) [241]. These findings were confirmed in several studies reporting the molecular alterations observed in MpBCs. Thus, in these studies, *TP53* mutations were observed in 56–75% of cases, *PIK3CA* in 23–48% of cases, *PTEN* mutations in 11–25% of cases [242,243,244,245].

Krings and Chen have performed a next generation sequencing study in 28 metaplastic carcinomas, which were subdivided into the main histologic subtypes subdivided into chondroid-matrix-producing carcinomas, spindle cell carcinomas, carcinomas with squamous, mixed spindle/squamous, and mixed metaplastic differentiation [246]. In the whole tumor cell population recurrent mutations were observed at the level of *TP53* (64%), *PIK3CA/PIK3R1* (61%), *RAS/MAP* kinase (25%), and *TERT* (25%) mutations [246]. A great mutational heterogeneity was observed in the various histological subtypes: (i) Chondroid-matrix producing carcinomas lacked PI3K and RAS/MAPK aberrations and *TERT* promoter mutations, when compared to 100%, 39%, and 39% of non-matrix-producing tumors, respectively; in contrast, *TP53* mutations were observed in 90% of these tumors; (ii) *TERT* promoter mutations were particularly frequent (47%) in the spindle cell carcinomas and in tumors with squamous differentiation; (iii) *PIK3CA* mutations were particularly frequent in squamous and spindle/squamous tumors, whereas *PIK3R1* mutations were particularly frequent (80%) in spindle carcinoma; and, (iv) spindle cell carcinomas lacked *TP53* mutations [246]. Ng et al. obtained similar findings [241].

Interestingly, Dave and coworkers reported a very high frequency (97.5%) of mutations at the level of the ribosomal protein L39: the mutations of this protein increased the inducible nitric oxide synthase; this finding suggests a possible use of nitric oxide synthase inhibitors in these tumors [247].

Afkhami and coworkers recently explored the mutational and immune profiling of 21 MpBC patients and have evaluated possible correlations with survival: the most commonly altered genes were *TP53* (68%), *PIK3CA* (42%), and *PTEN* (16%); for patients with *PIK3CA* mutations, relapse-free survival and overall survival were significantly worse than for those without; PD-L1 expression was associated with worse survival [248]. In line with these observations, Joneja et al. observed PD-L1 expression in a high proportion (46%) of metaplastic tumors [243].

McCart and Reed have performed the analysis of a large set of MpBC patients (347 patients) and have analyzed the mutation profile of a subset of these patients; this study showed that the most significant indicators of poor prognosis were large tumor size, loss of cytokeratin expression, EGFR overexpression and, for mixed MpBCs, the presence of more than three different morphologic entities within the tumor [249]. Exome sequencing studies confirmed the enrichment in these tumors of *TP53* and *PTEN* mutations; intriguingly, these authors also observed concurrent mutations of *TP53, PTEN*, and *PIK3CA* [249].

Among the types of MpBCs, there is a subgroup of spindle cell tumors that resemble mesenchymal lesions, but exhibit an epithelial/myoepithelial immunophenotype; a molecular characterization of these tumors showed some peculiar features when compared to other metaplastic tumors: in 82% of cases a distinct chromosomal loss in the 9p21.3 region, including *CDKN2A* and *CDKN2B* was observed; the biallelic loss of the *CDKN2A/CDKN2B* region was observed in 50% of deleted cases; the expression of CDKN2A was absent in all cases with 9p21.3 loss; other genetic alterations frequently observed in other metaplastic breast cancer subtypes, such as *TP53* mutations, were virtually absent in myoepithelial carcinomas, whereas *PIK3CA* mutations were present in 53% of myoepithelial breast cancers [250].

## 14. Neuroendocrine Breast Cancer

Neuroendocrine tumors of the breast are defined according to the expression of neuroendocrine markers, in association with the presence of morphologic features of neuroendocrine cell differentiation. The WHO classification of tumors of the breast describes three main histologic types: neuroendocrine tumor, well-differentiated neuroendocrine carcinoma, and poorly differentiated/small cell and invasive breast carcinoma with neuroendocrine differentiation. The well-differentiated neuroendocrine tumors represent approximately 0.5% of all breast cancers and they are characterized at immune histochemical level by the expression of synaptophysin and chromogranin in >50% of the cells and ER and PR-positivity [251].

Few studies have characterized neuroendocrine breast cancers at the molecular level. Weigelt and coworkers showed that these tumors are luminal breast cancers with transcriptomic profiles that are similar to type B mucinous carcinomas [223]. Two studies have explored the mutational profiling of small numbers of neuroendocrine breast cancers by targeted sequencing panels and identified recurrent gene mutations that occur at the level of PIK3CA, the FGFR family, and chromatin remodeling genes [252,253]. Thus, these studies provided evidence that neuroencrine breast cancers display a profile of somatic mutations that appears to be different from common types of ER^+^/HER2^-^ breast carcinomas and, to some extent, intermediate between luminal A and luminal B breast carcinomas from TCGA [252,253]. These studies also underlined the heterogeneous nature of neuroendocrine breast cancers.

A recent study assessed the clinical, phenotypic, and molecular features of 47 neuroendocrine breast carcinomas providing evidence that the stratification of the tumors into the three different WHO groups (well-differentiated neuroendocrine tumors, poorly differentiated neuroendocrine carcinomas, and invasive breast carcinomas with neuroendocrine differentiation) did not reveal statistically significant differences in terms of progression-free survival or overall survival; in 7% of cases, *PIK3CA* or *TP53* mutations were detected [254]. According to these findings, it was concluded that neuroendocrine breast carcinoma is a distinct subtype of luminal carcinoma with a low rate of *PIK3CA* mutations and aggressive clinical behavior [254].

Pareja and coworkers recently reported a careful analysis of somatic mutations observed in 15 cases of neuroendocrine breast cancers, showing that the most frequently mutated genes in these tumors were *FOXA1, TBX3,* and *KMT2C* (all, mutated in 20% of cases) [255]. Importantly, when compared with HR^+^HER2^-^ tumors reported in TCGA, neuroendocrine breast cancers harbor a higher frequency of mutations targeting *FOXA1* (20% vs 2.9%) and *TBX3* (20% vs 2.9%) and a lower frequency of *TP53* mutations (0% vs 23.95) [255]. Another important feature of neuroendocrine breast cancer is a lower frequency of *PIK3CA* (13.3% vs 40%) and concurrent 1q gains and 16 losses (20% vs 47%), in comparison with HR^+^/HER2^-^ tumors that were reported in TCGA [255].

## 15. The Genetic Abnormalities of Male Breast Cancers

Male breast cancer is a rare disease and it accounts for less than 1% of all breast malignancies. Its treatment is currently guided by studies performed on women breast cancer. Male breast cancers are usually diagnosed at later stage and older age. Estrogen levels, *BRCA2* or *PALB2* germline mutations or hereditary syndromes, such as Klinefelter syndrome, mainly represents risk factors for male breast cancer. The large majority of male breast cancers are represented by ER-positive ductal carcinomas; only rarely, male breast cancers display HER2 gene amplification of a triple-negative phenotype [256].

Given the rarity of the disease, only few studies have characterized at molecular level male breast cancers. Male breast cancer exhibits a heterogeneity of molecular alterations, basically reflecting those observed inER^+^/HER2^-^ female breast cancers, including frequent mutations of *PIK3CA* (20–36%) and of *GAT3* (15–16%), but less frequent *TP53* mutations (3–7%) than in women breast cancers (Figure 5) [257,258]. Furthermore, somatic mutations in genes regulating chromatin function and homologous recombination deficiency-related signatures were more frequent among male cancers [257,258]. *MDM2* amplifications were frequent (11%) and correlated with protein overexpression and they were associated with poor outcome [240]. In conclusion, male breast cancers constitute a heterogeneous disease, with a limited number of recurrently mutated genes (*PIK3CA, GATA3*, and *MAP3K1*) and numerous genes with pathogenic mutations at lower frequencies [257,258] (Figure 5).

## 16. Intra-Tumor Heterogeneity

In addition to inter-tumor heterogeneity, breast cancers also display also a consistent degree of intra-tumor heterogeneity. This intra-tumor heterogeneity is largely expected if we take into account that tumors are composed by clonal subpopulations of tumor cells. It is important to understand that intra-tumor heterogeneity is driven by genetic and epigenetic mechanisms, both contributing to tumor variability that, when combined with selection, drives tumor progression.

Breast cancers, like other solid tumors, are mosaics of cancer cells that follow a branched Darwinian evolutionary pattern with the existence, in addition to the initiating or founder clone, of a number of subclones that coexist and evolve in parallel. A clear example is given by primary TNBCs that have been shown to be composed by numerous and, in part, genetically distinct subclones, reflecting the high degree of genetic instability observed in these tumors [113]. The analysis of the various patients showed that clonal heterogeneity of TNBC is a continuum, with some tumors displaying a low clonality, while other cancers showing more extensive clonal evolution at diagnosis [113]. This variability in clonal heterogeneity of TNBC is also reflected by a consistent mutational variability, with some patients only showing few somatic mutations and pathways altered, whereas other patients displaying a more extensive mutational profile. Furthermore, in these tumors, although mutations in known driver genes, such as *TP53, PIK3CA*, and *PTEN*, are found to be truncal and therefore occur in the majority of cells of TNBCs, in some tumors their clonal frequencies are lower suggesting that they are not always the first genetic events [113].

Few studies have directly explored the genetic intra-tumor heterogeneity of breast cancers. In some studies, intra-tumor genetic heterogeneity was inferred from sequencing data. Thus, Ding et al. have analyzed the metastatic progression of a basal-like breast cancer to the brain: they have detected basically the same set of 50 coding mutations in the primary tumor and in metastases; although, few de novo mutations were detected at the level of metastases, the same set of 50 mutations was detected in primary tumor ad metastases, but important changes in allelic frequencies of mutations were observed, thus suggesting the existence of minority subpopulations of cells with metastatic potential at the level of the primary tumor [259]. Further evidence in favor of intra-tumor heterogeneity related to tumor evolution/progression came from a study carried out by Shah et al. [260]. They have sequenced the genomes and transcriptosomes of a lobular breast cancer at the level of the primary tumor and of metastases and showed that among 32 somatic mutations present in the metastasis, five were prevalent in the primary tumor, six were present at lower frequencies and 19 were not detected in the primary tumor [260]. This observation shows that single nucleotide mutational heterogeneity can be a property of breast cancers and significant clonal evolution occurs with disease progression [260].

Studies of multiregion sequencing have contributed to the definition of intratumor heterogeneity of breast cancers. In a fundamental study that was carried out by Yates and coworkers it was shown that the extent of subclonal heterogeneity in breast cancer is highly variable in the various tumors, with some tumors (about 45%) showing no marked differences in point mutations across the different tumor sub-regions (however, a minority of these tumors displayed significant heterogeneity in copy-number changes), a minority (about 6%) displaying pronounced heterogeneity and the majority showing intermediate levels of tumor heterogeneity [163]. Comparative analysis of tumor samples at diagnosis and on residual tumor cells after neoadjuvant chemotherapy suggests that clones only detected in residual mass after neoadjuvant chemotherapy are likely to represent subclones in which most of the mutations were present before treatment [163]. This analysis also supported the important conclusion that many of the breast cancer driver genes, such as TP53, PIK3CA, PTEN, and BRCA2 can be mutated early or late during breast cancer development: in fact, mutations in *TP53, PIK3CA, PTEN, BRCA2,* and *CDKN2A* are fully clonal in some tumors, but subclonal in others [163].

Particularly interesting is the analysis of multifocal primary breast cancers, being defined as multiple synchronous unilateral lesions of breast cancers, a condition that is frequently observed and associated with a greater aggressive behavior than unifocal tumors. Desmedt and coworkers have investigated by targeted sequencing 171 samples derived from 36 patients with multifocal breast cancers, being selected for their homogeneity at the level of tumor grade, histology, ER, PR and HER2 receptor status [261]. The presence of common point mutations in different tumor foci was observed in 67% of cases, indicating that all foci arose from a single tumor origin; somatic point mutations and particularly driver alterations were discordant in 69% and 44% of cases, respectively [261]. Interestingly, this discordance was more pronounced in spatially distant tumor foci [261]. According to the proportion of shared mutations the multifocal breast cancer patients were subdivided into homogenous, intermediate, and highly heterogeneous [261]. Yates et al. have reported a detailed analysis of four multifocal breast cancers, showing that in each case, separate foci of disease were clonally related; within individual foci, many private mutations reached high variant allele fractions, which suggested that during the growth of some foci complete “clonal sweeps” have occurred causing the complete replacement of all tumor cells by a single clone [163]. Interestingly, in one of these multifocal breast cancers, the focus with high-level of *CDK6* amplification responded poorly to neo-adjuvant chemotherapy, while the other focus without this amplification showed a complete pathological response [163].

Other evidences in favor of genomic heterogeneity of breast cancer derive from studies carried out at single-cell resolution. A study of this type was recently reported in two breast cancer patients, where 100 single cells were profiled from a triple-negative heterogeneous breast tumor, in addition to 100 single cells from a homogeneous breast tumor, and its paired liver metastasis; for each tumor the genomic copy number profile of single cells was obtained. This analysis showed that one tumor was monogenomic, consisting of cells with highly conserved copy number profiles and seemingly represents the result of a single major clonal expansion. In contrast, the second tumor was polygenomic, displaying three major clonal subpopulations exhibiting a common genetic lineage; interestingly, these three tumor subpopulations were organized into different regions of the tumor mass [262].

Ductal carcinoma in situ (DCIS) is the most common form of early-stage breast cancer; a relatively small percentage (10–30%) of these early lesions progress to invasive ductal carcinoma (IDC). The studies of cases of DCIS to IDC progression offer the unique opportunity the genetic basis at clonal level of this tumor progression. Recently, Casasent and coworkers have reported the development of a spatially resolved single-cell sequencing method to explore, at the single cell level, the mechanism of progression from DCIS to IDC [263]. A direct genomic lineage was observed between DCIS and IDC, and most CNAs and somatic mutations evolved in DCIS before invasion and development [263]. Importantly, invasion involved the co-migration of multiple clones of breast cancer cells from DCIS into the adjacent tissues [263]. The co-migration of multiple clones of tumor cells into the invasive regions raises a number of basic questions, suggesting that invasion might occur basically through two distinct mechanisms: a) the complete breakdown of the basement membrane and random migration of tumor clones into adjacent tissues; b) the cooperation of tumor clones that collectively break down the basement membrane [263].

In addition to genetically related intra-tumor heterogeneity, also heterogeneity related to epigenetic mechanisms plays a key role in breast cancer development and response to therapy. Thus, some studies have explored a tumor heterogeneity of ER expression/signaling. Abnormal ER signaling drives the majority of breast cancers and is targeted by endocrine therapies. The characterization of ER function in normal breast and in breast cancers indicates differential patterns of ER signaling, supporting the view that normal ER signaling is lost and tumorigenic ER signaling is gained during breast cancer formation [264]. Lindstrom and coworkers have explored the spatial tissutal distribution of ER in tumor biopsies of 1780 postmenopausal breast cancer patients randomly assigned to receive or not adjuvant tamoxifen and have classified the samples into two groups: low intratumor heterogeneity and high intratumor heterogeneity; patients with high intratumor heterogeneity of ER had a two-fold increased long-term risk as compared with patients with low intratumor heterogeneity [265]. A similar conclusion was reached by Saha et al. showing that the presence of heterogeneity in ER percentage staining was prognostic of reduced distant recurrence-free survival with a hard ratio of 4.26 [266]. In line with these observations, Patten et al. have evidenced a phenotypic heterogeneity in luminal breast cancer patients, which were related to the epigenetically-regulated YY1 transcription factor that was shown to be a critical determinant of ERα transcriptional activity promoting tumor growth in most luminal patients and also contributing to the expression of genes mediating resistance to endocrine treatment [267]. These observations support the view that epigenetic mechanisms significantly contribute to phenotypic heterogeneity and evolution in breast cancer patients.

The cause of inter- and intra-tumor heterogeneity of breast cancer is unknown and largely debated and it was related either to a multiplicity of molecular initiating events or to different cells of origin. Probably, a combinatorial model coupling both different cells of origin and different initial events can more plausibly explain the diversity of the different breast cancer molecular subtypes [268]. In this context, mutational catastrophic events, such as kataegis, may play a relevant role in the development of tumor heterogeneity.

## 17. PIK3CA Mutations in Breast Cancer

In spite of the dramatic progress in our understanding of the genetic mutations occurring in breast cancer, the role in cancer development and response to therapy of the most frequently mutated genes remains still unclear. In this context, the role of *PI3KCA*, the gene most frequently mutated in breast cancer, was intensively investigated to evaluate its role in tumor development and evaluate its possible therapeutic targeting. Activating mutations of *PIK3CA*, encoding the catalytic subunit p110α subunit of PI3K, occurs in about 30% of breast cancers and more frequently in ER^+^ breast cancers. *PI3KCA* mutations usually occur at the level of helical (hotspots E545K and E542K) and kinase (hotspot H1047R) domains and result in mutant constitutively active able to transform cells in culture and to promote tumorigenicity in various xenograft models [269]. *PIKR1* encodes the p85α regulatory subunit, which inhibits the catalytic activity of p110α.

Chen et al. have recently reported a detailed analysis of the spectrum of mutations affecting the PI3K pathway in a group of 126 Chinese breast cancer patients [270]. Sixty percent of these patients possessed one or more mutations in genes involved in the PI3K pathway. The most frequently mutated genes were *PIK3CA* (44%), *PIK3R1* (17%), *AKT3* (15%), and *PTEN* (12%); a high proportion of tumors harbors multiple mutations, particularly *PK3CA* plus *PIK3R1* mutations [270]. In line with this finding, Pereira et al. reported the *PIK3CA* or *PIK3R1* mutations frequently co-occurred with *PTEN* mutations in breast cancer patients [49]. The rate of PIK3Ca mutations differed in the four breast cancer subtypes with 50% of luminal A, 46,7% of luminal B, 41,2% of HER2-enriched, and 28% of TNBC patients [270]. It is important to note that in this study on Chinese breast cancer patients the frequency of *PIK3R1, AKT1, AKT2, AKT3,* and *PDK1* mutations was higher than that observed in the TCGA data set [270]. Zhang and coworkers, studying another large cohort of Chinese breast cancer patients, confirmed a high frequency of *AKT1* mutations (8%) in these patients, all pertaining to the HR^+^HER2^-^ subgroup [271].

It is important to note that PI3K is also activated in breast cancers through mechanisms different from *PI3KCA* mutations, such as *ERBB2* amplifications and *PTEN* loss of expression. It is also important to underline that multiple abnormalities of the PI3K pathway all cooperate to induce the activation along the PI3K-AKT signaling root: in fact, the majority of breast cancers overexpress *PDK1* (3-Phosphoinositide-dependent kinase 1), a serine threonine that is directly activated by PI3K and required for AKT activation [272]. PDK1 is the first enzyme of the PI3K signal output and it is required for the activation of AKT; PIP_3_ recruits PDK1 and AKT to the cell membrane through their pleckstrin homology domains, allowing for PDK1 to activate AKT by phosphorylating it at residue threonine-308. PDK1 mRNA and protein are overexpressed in the majority of breast cancers and 21% of these tumors have five or more copies of the gene encoding *PDK1, PDPK1* [273]. Increased *PDPK1* copy number is associated with upstream pathway alterations, such as *PTEN* loss, *PIK3CA* mutations or *ERBB2* amplification, and with patient survival [273]. Increased PDK1 levels are involved in the mechanism of acquired resistance to CDK4/6 inhibition in ER-positive breast cancers [273].

Clinical studies with PI3K inhibitor Alpelisib have shown that a population of treated patients displayed deep and prolonged clinical benefit [274]. A recent study explored the possible determinants of this pronounced sensitivity to PI3K inhibitors [275]. Exploring PIK3CA cancer genomes, Vasan et al. showed that 12–15% of breast cancers display multiple *PIK3CA* mutations, located in cis, thus ascribed to the same mutant allele [275]. In most double-*PIK3CA*-mutant breast cancers, one of the mutations was either a helical or kinase domain major-hotspot mutation [275]. Double *PIK3CA* mutant hyperactivate PI3K and enhance cell proliferation of breast cancer cells [275]. Importantly, double *PI3KCA* mutants are hypersensitive to PI3K inhibition in cells and double-mutant *PIK3CA* mutant tumors exhibit an increased sensitivity to PI3K inhibition in patients [275]. Thus, the analysis of data from the SANDPIPER phase III clinical trial of the PI3Kα/δ/γ inhibitor taselisib with fulvestrant in patients with ER^+^ metastatic breast cancer revealed that the overall response rate was better in patients with multiple mutations in PI3KCA rather than one [275].

Genomic studies have shown inactivating mutations of *MAP3K1* (13–20%) and *MAP2K4* (8%), two upstream kinases of the JNK apoptotic pathway in luminal A breast samples; interestingly, simultaneous mutations in *PIK3CA* and *MAP3K1* are observed in about 11% of *PIK3CA*-mutant tumors [276]. Interestingly, a recent correlative molecular characterization of primary and metastatic breast cancer patients enrolled in a phase Ib study combining the pan PI3K inhibitor Buparlisib with Letrozole in ER^+^, HER2^-^ metastatic breast cancer, showed that: (a) activating mutations in PIK3CA and inactivating mutations in *MAP3K1* marked tumors with evidence of clinical benefit related to the treatment; and, (b) double mutations in both *PIK3CA* and *MAP3K1* in the same patients displayed the greatest clinical benefit and the longest overall survival [277]. Almost all patients responding to the treatment were classified as subtype Luminal A, according to PAM50 analysis [277].

Attempts to analyze the detailed effects of PI3K pathway driven transformation on both cell proliferation and tissue morphology have used three-dimensional (3D) models of epithelial cell cultures that more closely recapitulate breast physiopathology than simple adherent cell cultures. While using this experimental tool, it provided evidence that cells lacking PTEN [278] or expressing oncogenic mutants of p110α [279] have been found to display aberrant tissue architecture and fail to form a simple hollow lumen. Since one of the main activated PI3K targets is AKT, it directly explored a role for AKT activation in the PI3K-mediated control of mammary gland architecture. Using 3D cultures of murine mammary epithelial cells, it was explored in these cells the role of either expression of mutant *PI3KCA* or mutant *AKT1*, or the overexpression of *HER2* or the induction of *PTEN* loss in modifying mammary gland architecture, particularly in terms of lumen formation [280]. The analysis of the transduced cells showed that only the expression of mutant *PI3KCA* or the induction of *PTEN* loss elicited increased proliferation and defective lumen formation; in contrast, *HER2* overexpression or *PI3KCA* mutant expression both elicited increased proliferation, but not disturbed lumen formation [280]. These observations indicate that PI3K pathway controls lumen formation through a mechanism that does not correlate with its ability to control AKT.

Various studies in mouse experimental models clearly support a strong tumorigenic role of *PIK3CA* mutants. In fact, it was shown that the expression of the *PIK3CA-H1047R* mutant in mammary epithelial cells is sufficient to induce tumor formation in transgenic mice: importantly, the mammary tumors induced by this *PIK3CA* mutant are of various histologic subtypes and co-express markers of both luminal and basal epithelial lineages [281]. According to these interesting observations, it was postulated that *PIK3CA* mutants may either (a) transform bipotent progenitor cells to allow for both luminal and basal cell differentiation; (b) induce the de-differentiation of luminal progenitors to bipotent progenitors, which are able to generate both types of cells [281]. In a second study, Liu and coworkers used a mouse model that was similar to that used in the previous study, with the exception that the transgene *PIK3CA-H1047R* was under the control of a doxycycline-inducible system. After the induction of tumor formation, silencing of *PIK3CA-H1047R* by the withdrawal of deoxycycline induced complete tumor regression in 1/3 mice, partial regression in 2/3 mice, followed by tumor relapse [281]. A part of the recurrent tumors maintained high levels of p-AKT and were sensitive to the inhibitory effects of PI3K inhibitors, while the other part exhibited low p-AKT levels and was resistant to the PI3K inhibitors [282]. These last tumors showed a dependence on MYC and MET for their growth [282]. In a third study, it was provided evidence that genetic ablation of p110α blocks tumor formation in HER2 transgenic models of breast cancer [283]. Surprisingly, p110β silencing caused an opposite effect, which resulted in increased tumorigenesis [283]. These last finding were interpreted while assuming that p110β competes with the more active p110α for receptor binding sites, negatively modulating the level of PI3K activity [283]. A fourth study was based on the analysis of an organoid model of mammary neoplasia that was based on the analysis of the effects of the expression of mutant H-RAS^G12V^ into either basal or luminal mammary cells. The expression of H-RAS mutant in both basal and luminal mammary cells resulted in enhanced cell proliferation and, finally, in neoplastic transformation [284]. Although basal and luminal Ras activation produced similar overgrowth phenotypes, studies with PIK3 inhibitors showed a block of Ras-driven tumorigenesis into basal, but not into luminal cells [284].

Other studies have explored the oncogenic role of *PIK3CA* mutants at the level of mammary stem/progenitor cells. The induction of the expression of *PIK3CA(H1047R)* mutant in lineage-committed basal LGR5-positive and luminal Keratin-8-positive cells of the adult mouse mammary gland elicits cell dedifferentiation into a multipotent stem-like condition, thus favoring the formation of heterogeneous, multi-lineage mammary tumors; particularly, oncogenic mutant *PIK3CA* in basal cells induced the formation of luminal ER+/PR- tumors, while its expression in luminal cells induces the formation of luminal ER+PR+ or basal-like ER-PR- tumors; concomitant deletion of *TP53* and expression of mutant *PI3KCA* accelerated tumor development and induced the formation of more aggressive tumors [285,286]. Zhang et al. developed an efficient method for the generation of somatic genetically-engineered mouse models for breast cancer through ex vivo expansion and genome editing of mammary stem cells; using this system, they showed that thee suppression of the tyrosine phosphatase PTPN22 promotes PIK3-driven tumorigenesis, inducing metastatic activity in these tumors [287]. Interestingly, *PTPN22* deletion conferred resistance to PIK3 inhibitors to these tumors [287].

Other studies support a role for mutated oncogenic *PI3KCA* in breast cancer initiation, as supported by a mouse model that was generated with a Cre-recombinase regulated allele of p110a: tumor generated by two copies of p110a resulted in tumors with a carcinoma phenotype [288].

A recent study explored the oncogenic activity induced by loss of the tumor suppressor PI3K-p85α (*PIK3R1*): knockdown of this gene transforms human mammary epithelial cells, and its genetic ablation accelerates a mouse model of HER2/neu-driven breast cancer [289]. The partial loss of p85α allowed for defining its mechanism of action, mediated by the increase of p110α-p85α heterodimers bound to active receptors, thus increasing PIK3 signaling and oncogenic transformation [289]. PIK3 inhibitors block the oncogenic activity that is induced by *PIK3R1* loss [289].

Several studies have provided evidence that oncogenic *PI3KCA* mutations may play a role in breast cancer tumor progression. In fact, the frequency of *PI3KCA* mutations was explored in primary tumors, locally recurrent tumors, and distant organ metastases in a group of patients with invasive lobular breast cancer. Histological criteria identified this subtype of breast cancer, accounting for 10–15% of all breast cancers, forms part of the luminal-like breast cancer cluster and is characterized by ER-positivity and negativity beta-catenin and CK5/14 expression. Interestingly, the frequency of *PI3KCA* mutations in primary tumors was 33% and 26% in metastatic tumors; however, the frequency of *PI3KCA* mutants was particularly high in locally recurrent tumors (69%) [290] According to these findings, these authors concluded that the *PIK3CA* mutations are positively selected for during infiltrating lobular breast cancer progression to local recurrence, but not distant metastasis [290]. However, Shah and coworkers have challenged this role of *PIK3CA* mutations in lobular breast cancer progression [291]. In fact, exome sequencing studies revealed that *PIK3CA* mutations were as frequent as *CDH1* mutations in lobular carcinoma in situ, but they were not a useful biomarker of progression, as they were as frequent in pure in situ lobular cancer as in lobular cancer in situ associated with invasive lobular carcinoma [291].

The idea that PI3K pathway abnormalities play a role in tumor progression is also supported by the analysis of HER2 overexpressing breast cancer. In fact, the analysis of a group of breast cancers with amplified HER2 with disease recurrence after trastuzumab treatment provided evidence regarding a high frequency of abnormalities of the PI3K pathway; 71% of patients exhibiting *PI3KCA* mutations and/or loss of PTEN expression; this value has to be compared to 44% in an unexposed cohort of *HER2*-amplified tumors [292]. These observations suggest that PI3K pathway abnormalities play a role in the progression of HER2-amplified breast cancers and that a double targeting of both HER2 and PI3K could represent a potentially useful strategy for the treatment of these tumors [292]. STAR-FISH in situ single-cell analysis in HER2-positive breast cancer patients undergoing neo-adjuvant chemotherapy treatment showed that: i) this treatment increased the proportion of cells displaying *PIK3CA* mutations; ii) *PIK3CA* mutations and *HER2* amplification do not seem to be expressed in the same tumor cells; and, iii) treatment-associated changes in the spatial distribution of cellular genetic diversity correlated with long-term outcome following adjuvant therapy with trastuzumab [293]. It is unclear whether the presence of *PIK3CA* mutations might affect the clinical response of HER2-positive breast cancer patients undergoing treatment with HER2-targeting agents.

Thus, Loibl and coworkers reached the conclusion that PIK3CA mutations are associated with reduced pathological complete response rates in primary HER2-positive breast cancer through the pooled analysis of 967 patients from five prospective trials investigating lapatinib and trastuzumab [294]. Kim and coworkers have reported the results of BioPATH, a non-interventional study evaluating the relationship of molecular biomarkers, such as *PIK3CA* mutations, *PTEN* deletion/downregulation, truncated HER2 receptor, and tumor HER2 mRNA levels, to treatment responses in Asian patients with HER2-positive advanced breast cancer that was treated with lapatinib and other HER2-targeted agents [295]. The results of this study have shown that no difference was observed in the clinical outcome based on the status of these biomarkers, including *PIK3CA* mutations [295].

PI3KCA mutations seem to occur early during breast cancer tumorigenesis. This conclusion was based on the identification of identical *PI3KCA* mutations at the level of invasive breast carcinoma and in ductal carcinoma in situ (DCIS) [296]. Several other reports have confirmed the high frequency of *PIK3CA* mutations and of other members of the PI3K pathway in DCIS, but it is unclear whether these genetic alterations play a role in the progression of DCIS to IDC [297,298,299].

It is of interest to note that the PI3K pathway might be activated in a subset of breast cancers by a peculiar mechanism implying the formation of a complex with a cytoplasmic form of the ERα. In fact, it was shown that methylated ERα (this receptor is hypemethylated in some breast cancers) forms a cytoplasmic complex with Src and PI3K [300]. The screening of 175 breast cancer patients showed that 55% of them were found to display the cytoplasmic ERα complex: these tumors were associated with AKT activation, either positivity or negativity of the expression of nuclear ERα and reduced survival following standard treatment, when compared to patients with no cytoplasmic ERα complex [301].

The prognostic impact of *PIK3CA* genotype on breast cancer outcome is highly debated. In this context, highly significant was a recent meta-analysis that was carried out on the basis of the data of 10,319 patients that were derived from 19 studies [302]. *PIK3CA* mutations occurred in 32% of patients, displayed a significant association with ER positivity, increasing age, lower grade and smaller tumor size [302]. In univariate analysis, *PIK3CA* mutations were associated with invasive disease-free survival (IDFS) and overall survival; in multivariate analysis, *PIK3CA* genotype remained significant for improved IDFS, but not for overall survival [302]. Takeshita and coworkers reached a similar conclusion, who explored a group of early and late, metastatic ER^+^ breast cancer patients and showed that *PIK3CA* mutations, detected on circulating plasma DNA, were not associated with clinical outcome [303].

The high frequency of abnormalities and their role in tumor initiation and progression have strongly supported the targeting of the PI3K pathway as a potential therapeutic strategy for breast cancer. This choice was also supported by experimental studies showing that PIK3 hyperactivation might be involved in the escape from hormone dependence in ER-positive breast cancer. The inhibition of PIK3 and mTOR induces an apoptosis of estrogen-deprived cells and prevents the emergence of hormone-independent cells [304]. Thus, the activation of the PIK3 pathway was considered to be a critical step in the development of treatment resistance for both ER-positive and HER2-positive breast cancers and, consequently, consistent efforts have been made to develop various inhibitors [305]. PIK3 inhibitors are being developed for the treatment of ER-positive breast cancer in combination with antiestrogens. Understanding the temporal and pharmacodynamic effects of PIK3 inhibition in ER^+^ breast cancer is fundamental to develop a rationale for treatment scheduling to improve at maximum therapeutic index. Studies in experimental models have shown that short-term, complete PIK3 inhibition blocks cell growth in vitro more efficiently than chronic incomplete inhibition [306]. Longer-term PIK3 inhibition hypersensitive cells to growth factor signaling upon drug withdrawal [306].

The inhibition of PI3K can induce both decreased cellular proliferation and increased cell death of breast cancer cells. A variety large number of small molecule PIK3 inhibitors have been developed: dual PIK3/mTOR inhibitors, pan-PIK3 inhibitors, and isoform-specific inhibitors [307]. The safety and efficacy of these inhibitors has been tested in a wide range of preclinical and clinical trials [306]. Particularly, numerous clinical studies have explored the potential therapeutic impact of several PI3K inhibitors in breast cancer.

Buparlisib is an orally, pan-class I, reversible inhibitor of PIK3. Various clinical studies have explored the therapeutic effect of this PIK3 inhibitor in ER-positive breast cancer. Particularly relevant were the clinical studies BELLE-2 and BELLE-3. The BELLE-2 was a randomized phase III, double-blind, placebo-controlled, study, in which breast cancer patients ER^+^/HER2^-^ with advanced or metastatic disease and under disease progression after chemo-endocrine therapies were randomized to receive Buparlisib plus fulvestrant versus placebo plus fulvestrant [308]. In the Buparlisib-treated group, there was a slight improvement of progression-free survival from 5 to 6.9 months [307]. The subsequent analysis of the overall survival showed a trend to a slight, but not significant improvement, in the Buparlisib-treated group versus the placebo (33.2 versus 30.4 months) [309]. BELLE-3 was a randomized, double-blind, placebo-controlled, phase III study comparing Buparlisib plus fulvestrant to placebo plus fulvestrant in postmenopausal women with ER-Positive, HER2-negative advanced breast cancer progressing on or after mTOR inhibition [310]. Median progression-free survival was significantly longer in the Buparlisib group (3.8 months) as compared to the placebo group (1.8 months). However, the safety profile of Buparlisib plus fulvestrant does not support its further development in this setting of patients [310].

Taselisib is a potent and selective inhibitor of p100α, p110δ, and p110γ isoforms of class IA PI3K, with lower inhibitory potency against p110b isoform. Preclinical studies supported the use of the PIK3 inhibitor Taselisib in breast cancer patients bearing tumors with *PIK3CA* mutations, either ER^+^ or HER2^+^ [311]. Taselisib is being evaluated in breast cancer patients. A randomized, double-blind, placebo-controlled phase II trial (LORELEI) evaluated neoadjuvant letrozole plus Taselisib versus letrozole plus placebo in postmenopausal women with ER-positive, HER2-negative, early-stage breast cancer [312]. The results of this study showed that the addition of Taselisib to letrozole was associated with a higher proportion of patients achieving an objective response as compared to the placebo group (50% vs 39% in all patients and 56% vs 38% in PIK3CA-mutated patients) [312]. The preliminary results of the POSEIDON trial, testing tamoxifen plus Taselisib or placebo plus tamoxifen, support antitumor activity in both *PIK3CA* mutant and wild-type cancers [313]. As discussed above, a part of TNBCs (20–40%) express AR; these patients have a lower chance of achieving pathological complete response to neoadjuvant chemotherapy, thus highlighting the need for additional non-chemotherapy-based therapeutic strategies in these patients [117,118,119,120]. Interestingly, TNBC AR-positive are enriched in activating PIK3CA mutations and, based on the results that were obtained in preclinical models, there was a rationale to investigate in these patients, at a clinical level, the effect of combined AR and PI3K inhibition [308]. Thus, Lehmann and coworkers have initiated a randomized phase Ib/II clinical trial evaluating orally administered enzalutamide (AR inhibitor) with the PI3K inhibitor taselisib in patients with AR-positive TNBC, providing preliminary evidence that the combination of enzalutamide and taselisib increased the clinical benefit rate [314].

Alpelisib is an oral selective PIK3α isoform inhibitor, exhibiting dose-dependent antitumor activity in preclinical models, including tumor xenograft models, with more pronounced activity against *PIK3CA*-mutated tumors [315]. The randomized, phase III clinical trial SOLAR-1 provided evidence that treatment with Alpelisib-fluvestrant prolonged progression-free survival among patients with *PIK3CA*-mutated, HR-positive, HER2-negative advanced breast cancer who had received endocrine therapy previously (in the Alpelisib-fulvestrant group the PFS was 11 months as compared to 5.7 months in the placebo group) [315].

Ipasertib is an AKT inhibitor under clinical evaluation in TNBC patients. The LOTUS, randomized, placebo-controlled, double-blind, phase II clinical trial explored the potential therapeutic effect of Ipataserrtib in addition to Paclitaxel in women with inoperable, locally advanced, or metastatic TNBC [316]. The results of this study showed a longer progression-free survival among patients who received Ipatasertib when compared to those who received placebo [316]. Another study explored the effect of Ipatasertib to paclitaxel for early TNBC patients; however, this drug association did not significantly increase the pathological complete response rate as compared to the placebo group that was treated with paclitaxel alone [317]. Finally, a recent study provided preliminary evidence about the potential therapeutic efficacy of a triplet regimen based on the combination of Ipasertib, atezolizumab (anti-PD-L1), and paclitaxel [318]. The responses were observed irrespective of the biomarker status (PIKCA/AKT1/PTEN alterations or PD-L1 expression) [318].

Capivasertib is an AKT inhibitor that is being explored in breast cancer. Several studies are evaluating the therapeutic impact of this inhibitor in breast cancer. The BEECH study showed that adding Capivarsetib to paclitaxel did not prolong progression-free survival in a population of patients with ER-positive advanced or metastatic breast cancer [319]. In the context of the BEECH trial, dynamic changes in circulating tumor DNA were used as a surrogate to predict long-term outcome: median progression-free survival was 11.1 months in patients with suppressed ctDNA at four weeks and 6.4 months in patients with high ctDNA; there was no difference in the level of ctDNA suppression between patients randomized to Capivasertib or placebo overall or in the *PIK3CA* mutant subpopulation [320]. The FAKTION trial investigated the addition of Capivasertib to fulvestrant for postmenopausal women with ER^+^ and HER2-negative breast cancer after relapse or disease progression on an aromatase inhibitor [321]. The addition of Capivasertib to fulvestrant for patients with endocrine resistant advanced breast cancer resulted in a significantly longer progression-free survival and improvement in overall survival [320]. Particularly interesting were the results of a recent study assessing capivasertib plus paclitaxel versus placebo plus paclitaxel as a first-line tharapy for metastatic TNBC; addition of the AKT inhibitor capivasertib to the first line paclitaxel therapy resulted in significantly longer progression-free survival (9.3 months vs 3.7 months) and overall survival (19.1 months vs 12.6 months) and benefits were more pronounced in patients with tumors bearing alterations of the PI3K pathway [322].

## 18. TP53 Mutations in Breast Cancer

It was estimated that about 30% of breast cancers display *TP53* mutations and the frequency, spectrum, and timing of these mutations varied according to the molecular subtype of the disease. Thus, *TP53* mutations are less common in luminal than in basal-like tumors, occurring in about 17% of luminal A tumors, 41% of luminal B, 50% of HER2-positive, and 88% of basal-like [323]. The analysis of large genetic datasets indicates the existence of an association between TP53 mutational status in breast cancer and the presence of infiltrating immune cells; particularly, the mutation of *TP53* R175H was associated with increased immune infiltration [324].

Donehower et al. recently performed an analysis of five data platforms in 10,225 samples from 32 different types of cancer that were reported by TCGA, enabling the comprehensive assessment of p53 pathway involvement in these cancers [325]. Interestingly, more than 91% of *TP53*-mutant cancers exhibit second allele loss by various molecular mechanisms (mutation, chromosomal deletion, or copy-neutral loss of heterozygosity); furthermore, most of *TP53*-mutated cancers display enhanced chromosomal instability [325]. Concerning breast cancer, these authors observed a virtually complete mutual exclusivity between TP53 mutations and CDH1 alterations and a partial exclusivity between TP53 mutations and MAP3K1 and also, in part, PIK3CA mutations [325]. Meric–Bernstam and coworkers explored, by high-depth next generation sequencing, 257 patients with metastatic breast cancer, including 165 patients with HR^+^/HER2^-^ breast cancer; in this last group of patients, PIK3CA (32%) and TP53 (29%) were the most frequently mutated genes and TP53 mutations were associated with shorter overall survival and progression-free survival [326]. The TP53 mutations were also prognostic in the group of HR^+^ patients with PIK3CA mutations, corresponding to <1/3 of total TP53-mutated HR^+^ patients [326]. Ungerleider et al. analyzed the large METABRIC dataset and showed that, in all patients and in HR^+^ patients hormone therapy-treated, TP53-mutant status conferred inferior five-year overall survival, but survival curves crossed at later times (10 or more years) [327]. However, in patients that were treated with chemotherapy alone, without endocrine therapy, TP53-WT status conferred a remarkably poor overall survival; the addition of hormone therapy to chemotherapy improved survival, notably in patients with TP53-WT status [327].

The presence of *TP53* mutations in ER^+^ breast cancer seems to play an important prognostic role. Lopez and coworkers have performed an analysis on a total of 3589 ER^+^ breast cancers from the publicly available datasets TCGA, MSK, and METABRIC; 27% of these tumors were PR^-^ and 73% were PR^+^ [323]. The most frequently mutated gene in PR^-^ tumors was PIK3CA, with a lower frequency in PR^-^ (37%) than in PR^+^ (47%) tumors [326]. The prevalence of tumors showing mutations in *TP53* was higher in the PR^-^ (33%) than in PR^+^ (19%) samples, with enrichment in the missense mutations P728S, I195T, and H179R mutations in *TP53* among PR^-^ tumors [323]. The tumor samples were subdivided into four clusters according to the *TP53* and *PIK3CA* mutational status: cluster 1 (*PIK3CA*-mutant/*TP53*-mutant 11%); cluster 2 (*PIK3CA*-mutant/*TP53*-WT 26%); cluster 3 (*PIK3CA*-WT/*TP53*-mutant 21%); and, cluster 4 (*PIK3CA*-WT/*TP53*-WT 43%) [323]. *TP53* mutational status had a negative prognostic impact on PR^-^/ER^+^ breast cancers, also evident at the level of cluster analysis [323]. In line with these observations, Ahn and coworkers have explored a large group of 272 patients that were surgically treated for ER^+^ breast cancer: 10.3% of these tumors was *TP53*-mutated; *TP53* mutation rate was significantly higher in low-PR tumors than in high-PR tumors (17.1% vs 7.9%); and, low-PR tumors displayed a dysregulated glucose metabolism (as evidenced by 18F-FDG PET) and an adverse impact on recurrence-free survival [328].

Park et al. performed an analysis on young ER^+^ Korean breast cancer patients; these patients were subdivided into three subgroups: A, chromosomal-stable, mainly enriched in luminal A (91%); B, a mixture of luminal A and B (89%); and, C, including HER2-enriched and luminal B types (64%) [329]. *TP53* mutations largely prevailed in the group C [329]. Abubakar and coworkers have screened a very large (7226 cases) cohort of Chinese women with invasive breast cancer according to p53 immunohistochemical expression, a TP53 mutation surrogate [330]. Interestingly, in luminal A-like and B-like/HER2-negative subtypes, p53 positivity was associated with early-onset tumors, high-grade, high proliferative index, and basal marker expression [330].

*ESR2* (estrogen Receptor β) displays a consistent homology with ESR1 (Estrogen Receptor α); in spite of this structural homology, *ESR2* displays a pattern of expression and biologic functions different from *ESR1*. However, the exact function of ESR2 in breast cancer remains unclear, since some studies suggested a proliferative and other one an anti-proliferative effect on these tumor cells [331]. A recent study by Mukhopadhyay et al. provided a mechanistic explanation for this dual action, mainly related to the interaction of ESR2 with p53 [331]. *ESR2*-mutant TP53 interaction mediates the sequestration of mutant TP53, leading to TP73 activation and anti-proliferative effects [331]. Tamoxifen treatment increases ESR2 expression and reactivates *TP73* in mutant *TP53* cells [331]. TP53-expressing basal-like TNBCs with high *ESR2* expression display better survival [331].

Almost all *BRCA1*-related breast cancers contain deleterious *TP53* mutations (nonsense, frameshift, and splice mutations), leading to a loss of p53 expression [332]. The presence of deleterious *TP53* mutations in most BRCA1-related breast cancers suggests that p53 loss of function is essential for BRCA1-associated tumorigenesis [332]. Truncating *TP53* mutations have also been observed in sporadic basal-like breast cancers, displaying hallmarks of BRCAness [333]. Therefore, *BRCA1*-related tumors can be treated not only with drugs that target *BRCA1* deficiency, but also with drugs that target p53-deficient cells. In line with this assumption, a recent report showed that zinc metallochaperones (ZMCs), a new class of anti-cancer drugs that specifically reactivate zinc-deficient mutant p53 by restoring zinc binding, significantly improve the survival of mice bearing tumors harboring a zinc-deficient missense mutant *TP53* allele [334]. Furthermore, ZMC1 synergizes with olaparib in inducing the inhibition of the growth of *BRCA1*-deficient/*TP53*-mutated breast cancer [334].

Germline *TP53* pathogenic variants are rare and they represent a condition of predisposition to early-onset breast cancer, which, in association with other tumor types, such as sarcomas and malignant brain tumors, is clinically identified as a manifestation of Li-Fraumeni syndrome. In a recent study, Sheng et al. have explored in a large number of 10,053 Chinese breast cancer patients the incidence of germline *TP53* mutations: in the overall population 0.5% of patients carried a pathogenic *TP53* germline mutation and 3.8% in very early onset (<30 years) breast cancer [335]. The presence of germline *TP53* mutation was an independent unfavorable factor for recurrence-free survival, distant recurrence-free survival, and overall survival [336]. Histopathologic studies indicate that aggressive HER2-positive breast cancers with densely sclerotic stroma are common in germline *TP53* carriers [336]. Germline *TP53* mutations were associated with a moderate risk of developing TNBC, while the germline pathogenic variants of *BRCA1, BRCA2,* and *PALB2* were associated with high risk [129].

As above reported, TP53 mutations are particularly frequent in TNBC. Hancock and coworkers have recently explored in detail a possible contribution of different alterations of TP53 present in TNBC in response to neoadjuvant chemotherapy and in disease outcome [146]. Various abnormalities of TP53 are observed in TNBCs: some tumors harbored extensive involvement, showing features such ads loss-of-heterozigosity or copy loss of TP53, while other tumors displayed only subclonal point mutations [146]. 29% of TNBC patients showed abnormally low TP53 copy number, frequently associated with TP53 mutations: these patients are particularly chemoresistant and they have a reduced disease-free survival and overall survival, thus indicating that particularly low TP53 levels are associated with poor outcome [146]. The fact that only TNBCs with absent/very low residual TP53 activity have a poor prognosis helps to understand the lack of prognostic impact of TP53 mutations in TNBCs that were observed in some studies [337].

Recent experimental studies have explored the mechanisms through which *TP53* loss might have an oncogenic effect in breast tumor development. Cicalese and coworkers, while using a ERBB2 trasngenic model of breast cancer, showed that the self-renewing of cancer stem cells is higher that that observed at the level of normal mammary stem cells [331]. Interestingly, normal mammary stem cells with mutations of *TP53* possess the same increased self-renewing properties observed in cancer stem cells and promote the formation of mammary preneoplastic lesions; furthermore, pharmacologic reactivation of p53 in cancer stem cells correlated with restored asymmetric divisions and the inhibition of tumor growth [331]. In a more recent study, the same authors explored the molecular mechanisms through which *p53* loss promotes breast tumorigenesis [338]. Thus, it was shown that *c-myc* is a transcriptional target of *TP53* in mammary stem cells and it is activated in breast cancer as a consequence of *p53* loss [339]. Constitutive *myc* expression in mammary stem cells increased the rate of symmetric divisions and, in breast cancer, was necessary to maintain cancer stem cells [339]. Concomitant *myc* expression and *TP53* loss promotes the expression of 189 mitotic genes, corresponding to a gene expression signature allowing to identify breast cancer patients at a high-risk of relapse and of mortality [339]. Another experimental study provided evidence that the type of TP53 mutant affects the evolution of induced breast cancer: thus, p53R245W-driven breast cancers are aggressive and generate metastases at the level of lung and liver, whereas p53R172H-induced tumors are less aggressive and they require additional hits to develop metastases [340].

## 19. CDK4/CDK6 Inhibitors in Breast Cancer

The CDK4/CDK6-cyclin D1 complex phosphorylates Retinoblastoma 1 (Rb1), leading to a loss of repression on the E2F transcription factors, resulting in cell cycle progression from G1 to S phase and then in cell proliferation. CDK4/6 inhibitors, by blocking the CDK4/6-cyclin D1 complex, prevent the phosphorylation of Rb1, and block the cell cycle progression from G1 to S phase.

Cyclin D1 protein is very frequently (50–70%) overexpressed in breast cancer and its gene *CCND1* is also frequently (9–30%) amplified in these tumors. These findings suggest that molecular mechanisms other than gene amplification are also responsible for the overexpression of the cyclin D1 protein [341]. The *CCND1* gene maps to the 11q13 locus and the majority of tumors bearing *CCND1* amplification are ER-positive and overexpress cyclin D1 protein [342]. Although the majority of studies have suggested that *CCND1* amplification and/or cyclinD1 protein overexpression are associated with worse prognosis [342,343,344,345], other studies have suggested a better prognosis for these patients [346].

Two recent studies have contributed to clarifying this controversial topic. Thus, Lundberg and coworkers have explored the long-term prognostic and predictive capacity of cyclin D1 gene amplification in 2305 breast cancers, showing that a worse survival was observed for patients with luminal A and luminal B, ER^+^, lymph node-negative, HER2^-^ breast cancers [341]. The *CCND1*-amplified, ER^+^, luminal A tumors displayed an increased proliferation index and decreased progesterone levels and gene expression changes that were consistent with a more aggressive phenotype [341].

In the second study, Ahlin and coworkers have analyzed immunohistochemical nuclear expression of cyclin D1 in 364 breast cancer patients, showing that cyclin D1 expression over the median in ER^+^ breast cancers was associated with an increased risk for breast cancer-related death, even when adjusted for tumor size and grade [346]. No prognostic impact of cyclin D1 expression was found among the ER^+^ breast cancers [346]. In ER^+^ tumors, cyclin D1 overexpression was associated with an increased expression of the proliferation markers cyclin A and B [346].

ER and PR promote persistent *CCND1* gene activation through a transcriptional mechanism, which involves their interaction at the level of a specific gene regulatory element present at the level of the promoter of *CCND1* gene [347,348]. On the other hand, cyclin D1 stimulates ER transcriptional activity independent of cyclin-dependent kinase 4 activation [349]. Furthermore, other studies identified an estrogen-independent role for ER and CDK4/Rb/E2F transcriptional axis in the hormone-independent growth of breast cancer cells [349]. The ensemble of these observations strongly supported the development of CDK4/6 inhibitors in breast cancer treatment.

Preclinical studies showed also that combined inhibition of CDK4/6, mTORC 1–2, and ER induces marked and durable regression of ER^+^ breast cancer models [350].

Several clinical studies that were carried out during the last years led to the approval by the Federal Drug Administration and European Medicines Agency of three selective CDK4/6 inhibitors, Palbociclib, Ribociclib and Abemaciclib, for the treatment of advanced metastatic HR^+^/HER2^-^ breast cancer.

The endocrine therapy represents an effective and well-tolerated therapeutic strategy for patients with ER^+^ breast cancer. In the postmenopausal women, treatment with tamoxifen or aromatase inhibitors for five years was associated with approximately 30% or 40% reduction, respectively, of mortality from breast cancer [351]. However, women with early-stage ER^+^ breast cancer who receive standard endocrine therapy for five years remain at risk of distant recurrence for at least 15 years after treatment end [351]. Other studies have shown that the extension of the duration of adjuvant endocrine therapy to 10 years has reduced the risk of recurrence only in a subset of patients [352].

Preclinical studies have supported the marked anti-tumor activity of the CDK4/6 inhibitor palbociclib in ER-positive breast cancer cells. In an initial phase II clinical study (PALOMA-1), the administration of palbociclib resulted in an improvement of progression-free survival in previously untreated patients with ER-positive/HER2-negative breast cancer. Subsequently, a phase III clinical study (PALOMA-2) confirmed a benefit in progression-free survival in patients that were treated in first-line with palbociclib plus letrozole for ER-positive/HER2-negative advanced breast cancer [353]. In the phase III clinical trial PALOMA-3, the capacity of plabociclib was evaluated, in association with fulvestrant, to improve the outcome of breast cancer patients with ER-positive/HER2-negative disease under progression after previous endocrine therapy [354]. At a median follow-up of 44.8 months, breast cancer patients receiving palbociclib plus fulvestrant had a median overall survival of 34.9 months compared to 28 months in the placebo plus fulvestrant group; these results only show a positive trend, but are not statistically significant [354]. The lack of significant improvement in overall survival was attributed, at least in part, to the commercial availability of palbociclib, administered to 18% of patients in progression in the placebo arm [354]. In 2015, palbociclib was approved for use in combination with letrozole from the treatment of ER-positive/HER2-negative advanced breast cancer as initial endocrine-based therapy in postmenopausal women and, subsequently, in 2016 in combination with fulvestrant in HER-positive/HER2-negative advanced breast cancers in progression following endocrine therapy [355]. In 2019, FDA approved a supplemental drug application extending the use of palbociclib plus fulvestrant or an aromatese inhibitor to male patients with advanced/metastatic breast cancer [355]. The analysis of a battery of relevant biomarkers in patients undergoing the PALOMA-3 trial showed no significant interaction between the treatment response and expression levels of CDK4, CDK6, cyclin D1, and RB1; however, palbociclib efficacy was lower in patients with high versus low cyclin E expression [356].

Ribociclib is another high-selective, reversible, small-molecule inhibitor of CDK4/6. In a phase III randomized, double-blind, placebo-controlled MONALEESA-2 study, 668 HR-positive/HER2-negative breast cancer patients with advanced breast cancer were randomized to receive either ribociclib plus letrozole or placebo plus letrozole [357]. Ribocilib improved progression-free survival in these patients from 16 months to 25.3 months [357]. On 2017, the FDA approved ribociclib for the treatment of postmenopausal women with advanced or metastatic HR-positive/HER2-negative breast cancer. In a phase III trial, the MONALEESA-3 trial, postmenopausal women with advanced HR-positive/HER2-negative breast cancer were randomized to receive ribociclib plus fulvestrant or placebo plus fulvestrant: the median progression-free survival was 20.5 months in the ribociclib arm, as compared to 12.8 months in the placebo arm [358]. The FDA approved riboclib in combination with fulvestrant for the treatment of advanced/metastatic HR-positive/HER2-negative breast cancer as first-line or second-line therapy, according to the results of this trial.

The MONALEESA-7 trial involved women with HR-positive/HER2-negative, advanced breast cancer treated with ovarian function suppression and oral endocrine therapy with ribociclib or placebo: median progression-free survival of 23.8 months in the ribociclin arm and 13 months in the placebo arm [359]. Recently, the results of overall survival in the MONALEESA-7 trial were published, showing that the addition of ribociclib to endocrine therapy resulted in a significant improvement in overall survival when compared to endocrine therapy alone: at 42 months, the estimated overall survival was 70.2% in the rebociclib group and 46% in the placebo group [360].

The MONARCH-1 trial was the first phase II clinical study to report single-agent activity of abemaciclib for heavily pre-treated ER-positive, HER2-negative metastatic breast cancer patients, with a median progression-free survival of six months [361]. MONARCH-2 was a phase III study that involved the enrollment of 669 patients with HR-positive, HER2-negative breast cancer who had progressed during adjuvant or neoadjuvant endocrine therapy; the patients were randomized to receive abemaciclib plus fuvestrant or placebo plus fulvestrant: median progression-free survival was longer in abmeciclib arm (16.4 months), when compared to the placebo arm (9.3 months) [362]. According to the results of the MONARCH-2 trial, the FDA approved abemaciclin in association with fulvestrant for the treatment of HR-positive, HER2-negative breast cancer patients with advanced/metastatic disease; furthermore, according to the results of the MONARCH-1 study, FDA approved abemaciclib as monotherapy in breast cancer patients HR-positive, HER2-negative with advanced/metastatic disease. Recently, the final analysis of overall survival data of this trial showed that treatment with abemaciclib plus fulvestrant resulted in a statistically significant and clinically meaningful improvement of median overall survival of 9.4 months for patients with HR-positive, HER2-negative advanced breast cancer after prior endocrine therapy, regardless of menopausal status [363]. MONARCH-3 was a phase III randomized clinical trial that involved the enrollment of 493 HR-positive, HER2-negative breast cancer patients with advanced disease who have not received previous treatment and randomized to receive either abemaciclib plus anastrozole or letrozole or placebo plus anstrozole or letrozole [364]. Abemaciclib treatment elicited a significantly longer median progression-free survival than placebo treatment (28.2 months versus 14.8 months) [358]. The overall response rate was 61% in the abemaciclib arm versus 45% in the placebo arm; the median duration of response was longer in the abemaciclib arm (27.4 months) as compared to the placebo arm (17.5 months) [365]. A recent phase II neoadjuvant study (neoMONARCH) that was based on the administration of abemaciclib and anstrazole to postmenopausal women with stage I-III HR-positive, HER2-negative breast cancer showed that the combined administration of the two drugs when compared to anastrozole alone achieved complete cell cycle arrest and enhanced immune activation (enhanced antigen presentation and activated T-cell phenotypes) [366].

Recently, Giuliano et al. performed a very large meta-analysis on all of the studies that were carried out between 200 and 2017 on HR-positive/HER2-negative breast cancer patients with metastatic disease. The results of this large analysis showed that, in the first-line or second-line setting, CDK4/6 inhibitors plus hormone therapies provide a better benefit than endocrine therapies in terms of progression-free survival; furthermore, no chemotherapy regimen with or without targeted therapy was significantly better than CDK4/6 inhibitors plus hormone therapies in terms of progression-free survival [367]. These data support the actual treatment guidelines supporting the new combinations of endocrine therapies plus targeted therapies as first-line or second-line treatments in breast cancer patients with HR-positive, HER2-negative metastatic disease [367].

Currently, four large international clinical trials (PALLAS (NCT02513394), MONARCHE (NCT03155997), NATALEE (NCT03701334), and ADAPTcycle (EudraCT 2018-003749-40)) are evaluating the combined treatment of a CDK4/6 inhibitor with endocrine therapy for the duration of 2–3 years in patients with intermediate to high-risk HR-positive/HER2-negative early breast cancers.

## 20. Activation of Cell Signaling Pathways and Breast Cancer Progression

The results of the genetic analyses performed on various types of breast cancers allowed to define the pattern of somatic mutations, gene expression levels, DNA copy-number alterations, and DNA methylation and, thus, to define these alterations in the context of the main oncogenic signaling pathways [368]. These studies showed that: (i) in luminal A breast cancers, PI3K pathway is frequently (62%) altered, followed by frequency by cell cycle (31%), RTK/RAS (28%) and p53 (25%) pathway; (ii) in luminal B breast cancers the most frequently altered pathways are p53 (49%), PI3K and cell cycle (48%), RTK/RAS (44%), Wnt (31%), Myc (26%), and Notch (25%); (iii) in HER2-enriched breast cancers the most frequently altered pathways are RTK/RAS (82%), p53 (78%), PI3K (60%), cell-cycle (40%), and Myc (29%); (iv) in basal breast cancers the most frequent alterations are p53 (91%), PI3K (53%), cell-cycle (51%), RTK/RAS (46%), Myc (39%), and Notch (38%); and, (v) in normal-type breast cancers, the most frequent alterations occur at the level of RTK/RAS and cell-cycle (36), PI3K (33%), and p53 (31%) [368].

As above discussed, some signaling pathways, such as ER signaling and HER2 signaling, play an essential role in the development and progression of ER-positive and HER2-positive breast cancers, respectively.

As above discussed, in ER-positive breast cancers, ER is the driving transcription factor whose target genes control proliferative activity and endocrine response of tumor cells. The progression of ER^+^ luminal breast cancers is associated with development of endocrine resistance, a phenomenon only in part related to dysregulation of ER and its pathway and, particularly, to the acquisition of activating mutations in *ESR1* gene observed in about 18% of tumors with acquired resistance to aromatase inhibitors [172]. Recent studies have shown that mutations in genes encoding the subunits of the *SWI/SNF* complexes, including *ARID1A*, are particularly enriched in the endocrine-resistant metastatic patients, thus supporting the view that these mutations may play an important role in resistance to endocrine therapy [167]. This view is further supported by a very recent study, showing that *ARID1A* gene mutations promote a switch from a luminal to a basal lineage, a phenomenon that is associated with development of endocrine resistance [369]. These observations support the existence of a link between lineage switching and endocrine resistance [369]. The lineage switching caused by loss of *ARID1A*-dependent loss SWI-SNF complex targeting the genomic sites of the basal-like cells [369].

Additional mechanisms of ER inhibitors resistance imply the increase of *HER2* mutations observed in about 7% of metastatic ER^+^ breast cancers who had received previous endocrine therapy [370]. These observations support the view that *HER2* mutations are acquired under the selective pressure of ER-directed treatment [370]. In these patients, the *HER2* and *ER* mutations are mutually exclusive [370]. The *HER2* mutations conferred resistance to tamoxifen, fulvestrant, and CDK4/CDK6 inhibitors [370]. The treatment resistance of these tumor cells was overcome by the combination of an irreversible HER2 inhibitor (neratinib) with ER-directed therapy [370]. Another mechanism of ER inhibitor resistance is related to ERalpha downregulation. In fact, the transcriptional levels of *ESR1* are clearly decreased in relapsed tumor lesions when compared to primary tumors in tamoxifen-treated ER+ breast cancers, which suggests a possible transcriptional inhibition of ESR1 during acquired resistance to tamoxifen [371]. Recent studies have, in part, clarified the mechanisms responsible for ERalpha downregulation through epigenetic mechanisms in part mediated by: (i) the downregulation of the transcription factor SALL2, whose reduced expression was responsible for reduced expression of ERalpha and PTEN: restoration of SALL2 levels in tamoxifen-resistant cells with DNA methyltransferase inhibitors resynthesized the cells to tamoxifen therapy [372]; (ii) endocrine therapy resistance was associated with hypermethylation and the loss of ER binding, which was associated with the loss of three-dimensional chromatin interactions [373].

Current standard first-line endocrine treatment for post-menopausal HR^+^/HER2^-^ metastatic breast cancer patients implies the administration of an aromatase inhibitor with a CDK4/6 inhibitor, abemaciclib, palbociclib or ribociclib, based on the results of three recent phase III clinical trials showing superior PFS as compared with aromatase inhibitor clone. There is some evidence regarding a crosstalk between the CDK4/6 and the PI3K pathways [374]. The impact of previous CDK 4/6 inhibition on alpelisib efficacy is at the moment still unknown. In this context, preliminary results of an ongoing clinical study (ByLieve), recruiting patients with advanced HR^+^/HER2^-^, *PIK3CA*-mutant breast cancers who have had disease progression during or after a CDK4/6 inhibitor, have shown a rate of response that was comparable to that observed in the context of SOLAR-1 study [375].

As above outlined, activated HER2 affects many cellular functions through various signaling pathways, such as MAPK and PI3K, associated with breast tumorigenesis. However, HER2 breast cancer disease is heterogeneous, as supported by: (a) the presence of a consistent heterogeneity of genomic profile, with the presence of HR^+^/HER2^+^tumors exhibiting more mutations in genes involved in homologous recombination, TGF-beta, and WNT signaling pathways than HR^-^/HER2^+^ tumors [376]; (b) the gene expression profiling with about 75% of tumors exhibiting a HER2-enriched transcriptional landscape, but with some tumors displaying a luminal B or luminal A gene expression profile [377]. In addition to the PI3K and MAPK pathways, MED1, which is an estrogen receptor binding protein, co-amplified with HER2, plays an important role as a critical mediator of HER2-driven breast tumorigenesis; this protein is a key crosstalk point for the HER2 and ER pathways in mediating the anti-estrogen resistance of HER2^+^/ER^+^ human breast cancer cells [378]. Furthermore, HER2 also induces proinflammatory pathways that are essential for breast tumorigenesis. In fact, Liu et al. showed that IL-1alpha signaling driven by HER2 promotes chronic inflammation required for sustaining cancer stem-like maintenance in HER2-positive breast cancers [379].

HER2-positive breast tumors are frequently aggressive and acquire early during tumor development the capacity to mestastasize. Murine mammary epithelial cells engineered to express high level of HER2 acquire early during tumor development the capacity to metastasize through a pathway involving the autocrine production of Wnt4 and RANKL: Wnt4 supports cell proliferation, survival, and tumorigenesis of breast cancer cells; RANKL supports the survival of metastatic cells [380,381].

As discussed above, HER2-positive patients benefit from specific HER2 targeted therapies, but not all of the patients respond to these therapies. This offers the unique opportunity to explore the molecular changes occurring in the tumor cells responding to therapy with Trastuzumab and those resistant to this treatment. Thus, Sathparthy et al. analyzed core needle biopsies from HER2-positive breast cancers before and 48–72 h after the start of therapy with Trastuzumab: greater suppression of HER2 protein and of both phosphorylated HER2 and mTOR levels in tumor cases associated with complete pathological response was observed; androgen receptor signaling, low HER2 activity, and the absence of HER2 amplification were identified as the mechanisms of drug resistance [382].

The analysis of the various studies carried out by TCGA for the characterization of the molecular abnormalities of breast cancers provided evidence that the NOTCH pathway exhibited frequent alterations in breast cancer basal (38%), followed by luminal B (25%), HER2-enriched (18%), and luminal A (14%) and BRCA normal (3%) [368].

In addition to the ER and HER2 signaling pathways, various other signaling pathways are altered in breast cancer and play a relevant role in the development and in the progression of specific subtypes of breast cancer. Feng et al. [383] and Nicolini et al. [384] recently reviewed this topic. Here, we will focus just on some of these pathways, including NOTCH, Hippo, and WNT.

NOTCH signaling is an important signaling pathway having multiple roles in the development and differentiation of cellular elements of multicellular organisms. The NOTCH signaling pathway plays an important role in normal breast development and during breast cancer development and progression [385,386,387].

Wang et al. have performed an analysis of the data contained in the TCGA datasets concerning the mutational profiling of *NOTCH 1, NOTCH2*, and *NOTCH3* in breast cancer patients [383]. The analysis of the 956 tumor samples that were contained in these data sets showed 42 mutations in *NOTCH1, NOTCH2,* or *NOTCH3* clustered in the heterodimerization or in the PEST domain (a negative regulatory domain that is responsible for the degradation of the active Notch Intracellular Domain, NICD); furthermore, five additional patients displayed other alterations, such as large deletions, disrupting the PEST domain [383]. Some tumors displayed focal amplification of *NOTCH2* or *NOTCH3*; 37% of NOTCH mutations co-occurred with copy number gains [383]. *NOTCH* alterations were much more frequent in TNBC (16.2%) than in other breast cancers (2.7%): furthermore, in the large majority of TNBCs, the *NOTCH* mutations were associated with NOTCH pathway activation [388].

In line with these observations, various studies, as reviewed by Giuli et al. [389], support the view that NOTCH receptors expression and activation correlates with the aggressive pathological and clinical phenotype of TNBC and is required for mammary cancer stem cell survival and proliferation in these cells [389].

Recently, Bertucci and coworkers reported the molecular analysis of 101 patients with IBC and showed frequent alterations of the NOTCH pathway in these tumors: *NOTCH1* was mutated in 12% of cases, *NOTCH2* in 9%, and *NOTCH4* in 3% [390]. Furthermore, other genes that are involved in the NOTCH pathway, including *MALM1* (11%), *MED12* (9%), and *FBXW7* (8%), were frequently altered in IBCs [390]. Interestingly, the NOTCH activation score was clearly higher in IBCs than among non-IBCs [390]. Concerning the non-IBCs, it was confirmed the existence of a growing NOTCH activation score from HR^+^/HER2^-^ to HER2^+^ and to TNBCs [390].

Interestingly, a recent study reported the occurrence of frequent *NOTCH* mutations in a rare form of breast cancer, adenoid cystic carcinoma (ACC) of the breast (<0.1% of invasive breast carcinomas), a tumor that is characterized by the absence of ER, PR, and HER2, which is considered a subtype of TNBC [391]. This tumor can be subdivided into a common conventional-type ACC and a solid-type ACC: *NOTCH1* (47% vs 7%), *NOTCH2* (13% vs 0%), and *NOTCH3* (7% vs 0%) were clearly more frequent in solid-type ACC than in conventional-type ACC [391].

Recent studies support a role for NOTCH signaling in the development and progression of ER^+^ luminal breast cancers. Thus, Kumar and coworkers showed that DLL1, but not other NOTCH ligands (DLL3, JAGGED1, and JAGGED 3), is overexpressed in ER^+^ luminal breast cancers: in these tumors, DLL1 overexpression correlates with poor prognosis [392]. In experimental models, the tumor-promoting function of DLL1 is exclusive to ER^+^ luminal breast cancers; in these tumors, estrogen signaling stabilizes the DLL1 protein, thus preventing its proteosomal degradation [392]. In breast cancer cells, NOTCH3 acts as a positive regulator of ER expression, an effect that is mediated by the binding of NOTCH3 at the level of CSL (CBF1, suppressor of hairless, Lag-2) binding region of the ERalpha gene promoter, with the consequent stimulation of the rate of transcription of this gene [393]. In ER^+^ primary breast cancer, NOTCH3 expression was associated with prolonged relapse-free survival [393]. The depletion of NOTCH3 in luminal breast cancer cells promoted metastasis in vivo [393].

Death domain-associated protein 6 (DAXX) was recently identified as a NOTCH target, whose levels were significantly increased in human breast cancers that were treated with endocrine therapy after a short treatment with a gamma secretase inhibitor [394]. DAXX RNA expression inversely correlates with NOTCH in human ER^+^ breast cancer samples [394]. Estradiol-mediated ER activation induces a stabilization of the DAXX protein, which represses the expression of genes associated with stemness, such as *NOTCH4, SOX2, OCT4, NANOG,* and *ALDH1A*; this finding suggests that the combination of endocrine therapy and DAXX-stabilizing agents might represent a potential therapy to inhibit tumor recurrence, through the inhibition of the cancer stem cell activity [394].

Interestingly, in a phase I clinical study that was based on the administration of PF-06650808, an anti-NOTCH3 antibody-drug conjugate, the majority of responder patients had ER^+^/PR^+^/HER2^-^ breast cancers [395].

The Hippo signaling pathway is an evolutionary conserved pathway that is involved in the physiological control of organ size and cell differentiation through mechanisms that involve the regulation of proliferation, survival, and apoptosis. This signaling pathway consists of a kinase cascade ultimately leading to the phosphorylation of the main Hippo pathway effectors, which are represented by yes-associated protein (YAP) and the transcriptional co-activator with PDZ-binding motif (TAZ) that induces the inhibition of the transcription of their target genes [396]. YAP and TAZ are transcriptional co-activators and need to bind transcription factors, such as TEAD-1 to -4, to stimulate gene expression at the level of target genes. The Hippo pathway was involved in the development and progression of many tumors, including breast cancers. Somatic alterations of the Hippo pathway genes are not frequent in breast cancer, but the Hippo signaling pathway is frequently activated in these tumors [396].

Several recent reports support a role for the Hippo pathway in breast cancer development. Thus, Zhu and coworkers, through an in vivo proximity-dependent labeling technique, provided evidence that Hippo and ER signaling pathways crosstalk at the chromatin level and converge at the level of specific enhancers to control gene expression, thus supporting a non-canonical function for YAP/TEAD at the level of the control of gene expression of ER-target genes [397]. YAP/TEAD control the enhancer activation by modulating the recruitment of MED1 [397]. These findings support the potential of YAP/TEAD as possible actionable targets of ER^+^ breast cancers [397]. The evidence of a crosstalk between the Hippo and ER signaling pathways is also supported by the studies on the G protein coupled estrogen receptor (GPER), which is a mediator of the genomic and non-genomic effects of estrogens, highly upregulated in breast cancer tissues [398]. TAZ and GPER expression positively correlated in human IDC specimens [398]. GPER stimulation activates YAP and TAZ, a process that is required for GPER activity [398].

As stated above, alterations in the Hippo signaling pathway in breast cancer might be caused by mutations of genes that are not directly involved in this pathway but representing external essential components of this pathway. An example is given by Disc large homolog 5 (DLG5), a protein that plays an essential role in the maintenance of epithelial cell polarity. A recent study showed the existence of a connection between DLG5 and YAP in breast cancer development and progression. DLG5 expression progressively decreased in primary breast cancer samples from stage I to stage III [399]. Loss of DLG5 in breast cancer cells inhibited the Hippo pathway by decreasing the phosphorylation of MST1/2, LATS1 (two activators of YAP/TAZ) and by increasing the nuclear localization of YAP, the loss of DLG5 induced the transcription of TEAD-target genes and, through this mechanism, promoted tumor cell proliferation [399]. DLG5 downregulation increases the frequency of breast cancer cells with stem-like properties, through an effect mediated by TAZ induction [400]. These last findings are in line with some observations supporting a key role of the Hippo transducer TAZ in the induction of stem cell-related traits on breast cancer cells [401] and for the metastatic and chemoresistance properties of these cells [402].

Recent studies support an important role of Hippo signaling at the level of the molecular mechanisms underlying the chemoresistance of breast cancer cells. Thus, two different studies showed the induction of ROR1 (type 1 receptor tyrosine kinase-like orphan receptor) following treatment with trastuzumab emtansine [403] or chemotherapy [404]: ROR1 enhanced the expression of YAP/TAZ and the self-renewal of cancer stem cells, inducing therapeutic resistance. The silencing of ROR1 and YAP in these cells exerted a clear anti-tumor effect [403,404]. Other studies support a role for Hippo pathway in the mechanism of resistance to CDK4/6 inhibitors [405]. The loss of FAT1 (Protocadherin Fat1) seems to be involved in Hippo activation in chemo-resistant breast cancer cells [405]. In fact, FAT1 physiologically assembles a multimeric Hippo signaling complex, which results in the activation of the Hippo kinases and consequent YAP1 inactivation [406]. Thus, Li and coworkers have analyzed 348 ER^+^ breast cancers that were treated with CDK4/6 inhibitors and identified loss-of-function mutations at the level of *FAT1* and *RB1* genes in association with drug resistance [405]. *FAT1* loss induced a pronounced increase of CDK6 levels and restoration of FAT1 levels blocked the CDK6 hyperexpression [405]. Importantly, the induction of CDK6 expression was mediated by the Hippo signaling pathway through the accumulation of YAP and TAZ co-activators at the level of the CDK6 gene promoter [405]. Interestingly, in these chemoresistant tumors, other genomic alterations in other components of the Hippo signaling pathway, such as YAP1 amplification, *LATS1* mutations, and *LATS2* deletion, are also found to promote CDK4/6 inhibitors resistance [405].

Wnt signaling pathway is a highly conserved signaling network that plays an essential role in tissue morphogenesis during development and adult tissue homeostasis. The mammalian genome contains 19 WNT genes, which encode 19 WNT proteins that are able to bind to 10 different Frizzled (FZD) receptors and several co-receptors [407]. In the canonical Wnt activation pathway, Wnt ligands interact with a FZD receptor and low density lipoprotein receptor-related protein (LRP5/6) receptors, downregulating the activity of a destruction complex (Dishevelled (Dve)/Axin/Glycogen synthase kinase 3β (GSK3β)), with consequent accumulation of β-catenin (CTNNB1), and its localization at the level of the nucleus [407]. The non-canonical Wnt signaling pathways involves Dvl, but not CTNNB1, and it implies other signaling molecules [407].

In breast cancers, the members of the Wnt signaling family are rarely mutated: *APC* (1.3%), *CTNNB1* (0.2%), *AXIN1* (0.4%), *LRP6* (0.7%), *LGR5* (0.6%), *TCF7L2* (0.3%), and *FBXW7* (1.5%) [407]. In spite of this low mutation rate, Wnt signaling is activated in over 50% of breast cancer and is associated with enhanced cyclin D1 expression and tumor progression and reduced overall survival [408]. Several studies have shown the frequent activation of Wnt pathway in TNBC development and prognosis: β-catenin activation in TNBC is not associated with CTNNB1 mutations and predicts poor outcome [409,410]. Other studies show an important role for Wnt signaling activation in HER2-induced breast tumorigenesis and, particularly in the progression of these tumors [411,412]. Wnt activation in chemoresistant HER2-positive breast cancer is induced by CDK12 overexpression and it plays a significant role in mediating resistance to trastuzumab-based therapy [413].

However, Wnt signaling activation, as supported by the presence of high levels of nuclear beta-catenin, is also observed in other breast cancer subtypes [414]. This conclusion is also supported through a network correlation analysis of gene expression results contained in the data sets of studies originated by TCGA [415].

Since the large majority of breast cancers do not harbor the mutations of genes that are involved in the Wnt signaling pathway, the frequent activation of this pathway in these tumors implies the epigenetic modulation of several key regulators of the Wnt pathway [407]: in fact, canonical Wnt ligands and receptors are frequently overexpressed in breast cancers, whereas secreted antagonists are repressed. The LRP6 co-receptor is up-regulated in a subpopulation of breast cancers: LRP6 silencing in these cells reduces Wnt signaling and tumor cell proliferation [416]. The LRP6 co-receptor is also overexpressed in other tumors (colorectal, liver, and pancreatic cancers) and reducing LRP6 expression in these tumors, as well as in breast cancers, was proposed as a strategy for inhibiting tumor cell growth [417].

The G-protein receptor 5 (LGR5) is overexpressed in a part of breast cancers and activates beta-catenin signaling in these tumor cells via protein kinase A [418]. LGR5 is overexpressed in various breast cancer subtypes and particularly in TNBCs; in these last tumors, a correlation between LGR5 expression and nuclear beta-catenin staining was observed [419]. The concomitant LGR5 overexpression and beta-catenin activation was associated with poor prognosis [419]. In LGR5-overexpressing breast cancers, Wnt signaling induces cancer progression and drug resistance [420].

The activation of Wnt signaling pathway in mouse models induces breast cancer formation. Cleary et al. have explored a mouse tumor model induced by aberrant expression of the secreted signaling ligand Wnt1: these tumors are composed by mixed lineages and comprise both basal and luminal tumor cell types, that derive from a bipotent malignant progenitor cell [421]. The luminal subclone was characterized by the secretion of canonical Wnt ligands required for sustaining the proliferation of the basal-like component of the tumor [421].

Wnt signaling in TNBC is involved in metastatic activity of these tumors [422] and its inhibition reduces the cancer stem cell properties and metastatic potential [423].

Recent studies support a key role of Wnt signaling in the mechanisms that contribute to breast cancer progression. Thus, Wallenstein and coworkers, through the analysis of 16 different genetically mouse models for breast cancer, provided evidence that the loss of p53 in cancer cells induced the secretion of WNT ligands that stimulate tumor-associated macrophages to produce interleukin-1β, thus inducing a condition of systemic inflammation [424]. Pharmacological inhibition of WNT secretion in p53-null breast cancer cells blocks macrophage-mediated IL-1β release, neutrophilic inflammation, and decrease of metastasis formation [424]. On the other hand, Evre et al. provided evidence that microenvironmental IL-1β that is produced by bone marrow cells stimulates breast cancer colonization through autocrine WNT signaling [425].

## 21. Animal Models of Breast Cancer

The animal models of breast cancer represent an essential tool for better understanding the pathogenetic mechanisms and developing suitable preclinical models of breast cancer.

Various animal models have been developed to study breast cancer, being mainly represented by genetically engineered mouse models (GEMMs), xenograft, or organoid model systems.

### 21.1. In Vivo Models

GEMMs of breast cancer have greatly contributed to cancer research, allowing for better defining the role of cancer genes in tumor initiation and development and of cancer genetic pathways, and to develop preclinical models. GEMMs are the most evaluated models of breast cancer, since they offer the unique opportunity to explore the stepwise progression of normal mammary epithelial cells to hyperplasia first and then to invasive tumors in the context of a native microenvironmental compartment and in the context of a functional immune system [426]. GEMMs of breast cancer have been developed while using various models: the introduction of the oncogene by direct injection of DNA into fertilized egg; introduction of a targeting vector-modified encoding-modified version of the tumor suppressor gene into mouse embryonic stem cells; and, conditional knock-in and knock-out in specific organs, such as mammary gland using the Cre/lox B system [427].

Many of the GEMMs are represented by simple oncogene-driven models, in which gene promoters, such as mouse mammary tumor virus-long terminal repeat (MMTV-LTR), are used to drive mammary expression of oncogenes. Mammary gland-specific expression of the polyoma T antigen (PyMT) induces a classical model of breast cancer under the control of the MMTV promoter: the tumors originating in this transgenic model are represented by multifocal adenocarcinomas, with high penetrance and metastatic capacity [428]. The histology of the tumors developing in the time, hyperplasia, adenoma, early, and late carcinoma is, to some extent, comparable to that of human breast cancer [428]. These tumors are originated through the malignant transformation of mammary unipotent LGR6^+^ progenitors [429].

Recent studies have characterized the mutational events occurring in different GEMM models, showing that most of the chromosomal alterations accumulate late during breast tumorigenesis; marked differences in copy number alteration prevalence between mammary tumors initiated with distinct drivers were observed; furthermore, some aberrations are recurrent and they were specific to some GEMMs, suggesting the existence of different driver-dependent routes to tumorigenesis [430]. The genomes of two highly used mouse models of breast cancer were recently analyzed and compared to human genomic data, showing that: (a) MMTV-Neu tumors display CNV involving the genes collagen type 1 alpha 1 (*Col1a*) and chondroadherin (*Chad*), an amplification that was particularly observed in HER2^+^ breast cancers; PyMT tumors displayed a highly conserved mutation in the gene coding the protein tyrosine phosphatase receptor (*ptphr*), resulting in elevated EGFR activity and consequent sensitivity to the EGFR inhibitor erlotinib [431].

Cyclin D1 expression is associated with poor prognosis of ER^+^ breast cancers. In MMTV-Cyclin D1 mice, mammary adenocarcinomas developed by 22 months of age, a phenomenon that was observed in 40% of the transgenic mice [432].

In some instances, MMTV-based GEMMs were used as preclinical tools for predicting drug sensitivity/resistance in the context of a peculiar genetic setting. Thus, Goel and coworkers have developed a transgenic model of HER2-positive breast cancer (MMTV-HER2) and they have used the tumor cells that originated in these transgenic mice to show that cyclin D1/cyclin-dependent kinase 4 (CDK4) mediated resistance to HER2-targeted therapy and that CDK4/6 inhibitors delayed a HER2-positive tumor relapse [433].

A number of studies were focused to the analysis of KRas-driven breast cancers based on different animal models. Thus, Nguyen and coworkers showed that basal cells and luminal progenitors that were isolated from normal human mammary tissue and transduced with the oncogene KRas^G12D^ generate serially transplantable invasive ductal carcinomas 8 weeks after the introduction into immunodeficient mice; these tumor cells rapidly generated secondary clones when passaged into secondary recipient animals [434].

Elevated Ras signaling drives many clinical luminal B cancers and it is associated with poor prognosis [435]; furthermore, KRAS promotes the mesenchymal features of basal-type breast cancer [436]. KRas activation is involved also in the induction of aggressive ER^+^ mammary tumors in the NRL-PRC murine model, defined by mammary-selective transgenic rat prolactin ligand rPrl expression [437].

Most of GEMMs of breast cancers do not reflect the salient features of their human counterparts [437]. To try to meet this request, an improvement of the current GEMMs and implementation of new technologies is strictly necessary [438]. The introduction of the CRIPR-Cas9 genome editing technology has completely revolutionized gene function studies and has offered a unique tool for in vivo modeling of genetic alterations that were observed in human cancers. CRISPR-mediated somatic engineering of the mammary tissue is feasible and particularly effective using intramammary injection of lentivirally encoded single-guide RNAs in female Cas9 kock-in mice [439]. Using this method, double-strand DNA breaks can be induced in situ at the level of specific gene targets and DNA repair processes can result in the formation of insertions or deletions leading to gene disruption [439]. This versatile platform can be used for the in vivo testing of tumor suppressor genes that are involved in various breast cancer subtypes and it was used to define the collaborative role of tumor suppressor genes in driving the development of invasive lobular carcinoma [440] and TNBC [441]. This model was further improved, allowing for the possibility to rapidly engineer breast cancer-associated point mutations *in situ*, thus allowing for the unique opportunity to recapitulate mutations in known and in putative oncogenes in preclinical models [442].

One of the limitations of the GEMMs consists in their incapacity to provide suitable models to functionally understand intertumoral heterogeneity of the breast cancer subtypes. However, recent studies on breast cancer GEMMs have attempted to bypass these important limitations. Thus, Risom and coworkers have shown that the introduction of a deregulated *MYC* into the murine NeuNT model of amplified-*HER2* breast cancer allowed for generating tumors displaying a consistent intertumoral heterogeneity, with generation also of aggressive tumors with metaplastic histology which mimic a subtype of *HER2*-amplified, ER^-^ human tumors by molecular expression [443]. These findings show the preclinical utility of this murine model to explore subtype-specific difference al the level of amplified-*HER2* breast cancers [443].

### 21.2. In Vitro Models

#### 21.2.1. Patient-Derived Xenografts

Patient-derived xenograft (PDX) models are based on a technology that involves the use of cancer cells or tumor tissues directly derived from patients that are injected into immunodeficient mice. These models offer the unique advantage of representing a tool to assess the efficiency of cancer drugs, preserving tumor characteristics that were similar to those of the tumor from which they were derived. An initial study by Zhang and coworkers reported the growth of human breast cancer tissues into highly immunodeficient mice with a successful yield of about 20% [444]. The large majority of tumors yielding xenografts were TNBC and, more rarely, HER2^+^ or HR^+^ tumors [444]. Importantly, serially passaged xenografts remained stable at the genetic and phenotypic level across multiple transplant generations and they exhibit comparable responses to those observed clinically (12 of 13 PDX tumors from 13 patients that were treated with the same drug displayed a matched response with the corresponding clinical response) [444].

Reyal and coworkers have explored the molecular profiling of 18 PDXs that were derived from breast cancer patients. A comparative analysis showed that 14/18 pairs primary tumors-tumor xenografts showed a similarity of >56% of copy number alterations [445]. Gene expression profile analysis showed that less than 5% of genes displayed recurrent variations between primary tumors and PDXs; importantly, these genes mostly corresponded to human stromal compartment genes [445]. Finally, an analysis of different tumors passaged showed that each tumor maintains genomic rearrangements and gene expression profiles [445].

Few studies have reported the isolation of ER^+^ PDX from breast cancer patients. Kabos et al. reported the development of ER^+^ PDX in 8/18 cases; importantly, these ER^+^ PDXs maintained a hormone receptor status that was comparable to that observed in the primary original tumors [446].

Goetz et al. have investigated PDX formation from 113 breast cancer patients, reporting an overall take rate of 27.4%, but with remarkable differences among various breast cancer subtypes: 51% in TNBC, 26% in HER2^+^, 5% in luminal B, and 0% in luminal A tumors [447].

The study of PDXs that were isolated from ER^+^ breast cancers consistently contributed to clarify the role of AR in these tumors. ARs are expressed in 80–90% of ER^+^ breast cancers, but their exact role in breast cancer development remains unclear [448]. The majority of studies suggest that AR expression in ER^+^ breast cancer is associated with a favorable prognosis [449,450]; however, other studies do not support a prognostic and predictive value of AR expression in ER^+^ breast cancer [451]. Recent preclinical studies suggest a bifunctional role of AR in breast cancer, with a growth promoter activity in tamoxifen-resistant cancer, and a growth inhibitor function in tamoxifen-sensitive breast cancer. These evidences were based on the study of subclones of the MCF-7 cell line [452], and on the study of tamoxifen-resistant clinical specimens, showing that a higher AR/ER ratio correlates with aggressive disease and poor prognosis [453]. A recent study by Ponnosamy and coworkers contributed to this important topic through the analysis of PDXs that were isolated from ER^+^ breast cancers. The proliferation of PDXs expressing WT and mutant ER were inhibited by an AR agonist or tissue-selective AR modulator, but not an AR antagonist [454]. The growth inhibitory signals were mediated by a reprogramming of ER and FOXA1 cistrome and by an alteration of phosphokinome signature, resulting in ER inhibition [454]. These observations strongly support a growth suppressive effect of ARs in breast cancer.

HER2-positive breast tumors display a low take rate in PDX assays, and this explains the relatively low number of studies on PDXs that originated from this breast cancer subtype. The use of PDXs was of fundamental importance to define the role of WW-binding protein 2 (WBP2), an oncogenic coactivator, co-amplified with HER2 in 36% of HER2-positive breast cancers [455]. The role and the mechanism of WBP2 in regulating breast cancer response to trastuzumab were elucidated using in vitro PDXs and murine xenograft models [455]. The elevated expression of WBP2 conferred to breast cancer cells a higher response to trastuzumab increasing HER2 downregulation and cell-cycle arrest [455].

HER2^+^/HR^+^ breast cancers form a heterogeneous group of tumors bearing the concurrent activity of HER2 and HR [456]. The optimal treatment of these tumors still remains undefined and recent clinical trials have explored the potential superiority of a dual blocking of both HER2 and ER pathways, showing improvements in progression-free survival, but not in overall survival [96,457]. In this context, particularly interesting was a study by Hsu et al. that reported the use of two PDX models that were isolated from two patients with HER2^+^/ER^+^ breast cancer and used to define a peculiar sensitivity of this tumor subtype to simultaneous suppression of both ER signaling and mTOR pathway, a potential combination treatment for these tumors [458].

AXL is expressed in HER2-positive breast cancers, where its expression correlates with poor patient survival [459]. While using a PDX model, it was shown that interfering with AXL through pharmacological inhibition resulted in an anti-tumor activity with reduced tumor cell invasiveness and reduced tumor metastatic burden [459]. AXL might represent a potential co-therapeutic target in HER2-positive tumors [459].

Vareslija and coworkers have explored gene expression profiling of 21 patient-matched primary breast and their associated brain metastases; bioinformatic analysis for potentially druggable targets displayed recurrent gains in RET expression and HER2 signaling [460]. RET and HER2 inhibition in a PDX model of brain metastases showed a clear significant antitumor response [460].

TNBC is the tumor subtype, from which the take rates of PDXs are highest. Furthermore, the development of models of these tumors is highly needed in view of their scarce response to current treatments. Thus, several studies have reported the use of PDXs of TNBC as a model system for the identification of new therapeutic approaches. El Ayachi and coworkers have identified a WNT10B-dependent β-catenin/HMGA2/EXH2 signaling pathway that was activated in TNBC; the chemical inhibition of WNT signaling in a chemoresistant PDX model that was derived from a TNBC patient exerted an antitumor activity and potentiated the apoptotic effect induced by doxorubicin [461]. Marangoni et al. have isolated PDXs from residual tumors of TNBC patients undergoing neoadjuvant treatment as a tool to identify new treatments and biomarkers [462]. Using PDXs originated from residual tumors, capecitabine was identified as an efficient chemotherapy in TNBC PDX resistant to anthracyclines, taxanes, and platins: RB1 positivity and high expression of TYMP were significantly associated with capecitabine response [462]. These authors have established in the time a large collection of TNBC PDXs, well representative of the molecular and phenotypic heterogeneity of TNBCs [463].

In patients with TNBC, the overexpression of cyclin E and phosphorylated-CDK2 are correlated with poor survival and absence of CDK2 desensitizes cells to inhibition of Wee1 kinase. Chen and coworkers have used TNBC PDX models to support the hypothesis that cyclin E can predict response to the Wee 1 kinase inhibitor AZD1775 [464]. The results of these studies allowed for proposing that cyclin E is a potential biomarker of response for AZD1775 as monotherapy in cyclin E-high TNBC tumors and for sequential combination therapy with CDK2 inhibitor or carboplatin followed by AZD1775 in cyclin E-low TNBC tumors [464].

AXL receptor tyrosine kinase is a key mediator of tumor invasiveness, particularly in TNBC. Leconet et al. demonstrated an inhibitory effect of the anti-AXL monoclonal antibody 20G7-D9: importantly, this inhibitory effect was observed in AXL-positive PDXs, but not in AXL-negative PDXs [393]. The anti-AXL antibody 20G7-D9 might represent a promising therapeutic tool in TNBCs with mesenchymal features [465]. More recently, PDXs with low AXL and high AXL expression were used to demonstrate the anti-tumor activity of DCC-2036, a novel AXL/MET inhibitor [466].

Merino and coworkers have exploited PDX models of TNBC to dissect the clonal dynamics of these tumors during their progression to metastatic disease. It was combined cellular barcoding with PDX models for high-resolution clonal assessment of drug-naïve TNBC, both longitudinally and across multiple sites, to perform these studies [467]. This analysis showed inter-tumoral and intra-tumoral heterogeneity that includes peculiar features of barcoded clones to differentially grow in primary tumors, shed into circulation, seed distal organs following removal of the parental clone in the primary tumor, and contribute to relapse after chemotherapy [467]. Circulating tumor cells are shedders that can be sustained by continuous supply from the primary tumor but do not seem to generate metastases [467]. Luminal androgen receptor (LAR) breast cancers account for about 10% of all TNBCs; anti-AR therapy was proposed for this subset of TNBCs, but the clinical benefit was limited. A recent study reported the study of PDX models of LAR TNBCs resistant to the AR inhibitor enzalutamide, indicating that the *PIK3CA* and *AKT1* are potential therapeutic targets in these tumors, as supported by the observation that mTOR and PI3K inhibitors had marked antitumor activity in vivo in PDXs [468].

Some breast cancer PDX models have contributed to define tumor heterogeneity. Lawson and coworkers, while using a PDX model, showed that progression to a metastatic condition is associated with increased proliferation and *MYC* expression: these events could be in part inhibited by CDK inhibitors [469]. The most metastatic PDXs had the highest proportion of cancer stem-like basal primary tumor cells, while the less metastatic PDXs had the lowest percentages of these cells [469]. According to these findings, it was suggested that the primary tumor contains a rare subpopulation of stem-like cells, and the relative frequency of these cells in the tumor correlates with the metastatic potential [469]. A similar conclusion was reached by Grosselin et al. who, using PDX models of acquired resistance to chemotherapy and targeted therapy in breast cancer, identified a subset of cells within untreated drug-sensitive tumors that display a chromatin structure that was similar to that observed in resistant cells; these cells have lost chromatin marks that were associated with stable transcription repression of genes that are known to promote resistance to treatment [470].

As above discussed, breast cancer PDXs represent a suitable model for exploring drug efficacy and the mechanisms of drug resistance. However, although PDX models are effective platforms for precision medicine, they exhibit some important limitations: (i) not all primary tumors generate xenotransplants when they are injected into an immunocompromised mice; (ii) the development of a PDX model from a patient requires time and often as long as six months are necessary; (iii) human stromal cells originally present in tumor specimens are progressively lost during xenograft growth and are replaced by host stromal cells, thus precluding the study of human tumor-stroma interactions; (iv) the genetic heterogeneity present in the original tumor cannot be always reproduced in the PDX; (v) PDXs have the tendency to select the most malignant cells that are present in the tumor; and, (vi) the need to use immunocompromised mice to avoid tumor rejection [471,472]. The standard PDX models can be used for the evaluation of chemotherapeutic drugs, but they are not suitable for the evaluation of immunotherapeutic drugs. To bypass this limitation, the development of mice with humanized immune systems is required [473]. Interestingly, Rosato et al. have reported the evaluation of an anti-PD-1 antibody in TNBC-derived PDXs engrafted in humanized mouse models, obtained by the injection of human CD34^+^ cells into immunocompromised mice [474]. In this humanized model, TNBC PDXs maintained the capacity to engraft and generate lung metastases; their growth was inhibited by anti-PDX-1 treatment when evaluated in the humanized mouse system, but not in standard immunodeficient mice [474].

#### 21.2.2. Organoids

During the last years, new cell culture techniques to grow tissues in vitro in 3D, as organotypic structures have been developed. Organoids are usually grown in matrices, such as Matrigel, collagen, or peptide hydrogels, which aim to recapitulate the breast microenvironment. These cell cultures originate organoids that can be grown from adult and embryonic stem cells and they are able to self-organize into 3D structures, largely reflecting the tissutal structure of the tissue of origin [475]. Organoid cultures have been established for a variety of normal and tumor-derived human tissue, including normal and mammary cells [475]. Few recent studies have reported attempts to develop organoid cultures of normal mammary tissue. Initial studies were based on the culture of mammary epithelial cells in Matrigel as extracellular medium (ECM): when grown under these conditions, mammary epithelial cells proliferate and form rudimentary spherical glandular structures, consisting of a monolayer of polarized epithelial cells that are separated from the surrounding Matrigel by a basement membrane; the apical side of the cells is oriented toward the center of the acinus [476]. However, the consistent limitations of this system are evident, being related to the incapacity to form a bilayered epithelium and recapitulate the various steps occurring during mammary gland development and regeneration. Thus, more recent studies on mouse or human breast epithelial cells have used advanced cell culture conditions and a three-dimensional (3D) scaffold for the formation of terminal duct lobular unit-like structures (functional units of the mammary gland *in situ*), composed by luminal and basal cells located in the correct orientation [477,478,479,480]. Thus, Zhang et al. reported the growth of estrogen-responsive mouse mammary organoids from single LGR5^+^ cells grown in Matrigel in the presence of a growth factor cocktail containing EGF, Wnt3a, and R-spondin [481]. These cells spontaneously organize into a ductal structure with basal cells around the periphery and luminal cells lining an interior cavity [481].

Jamieson et al. have reported the clonal generation of organoid structures from single-sorted murine basal mammary cells: these organoids comprised an inner compartment of polarized luminal cells with milk-producing capacity and an outer structure that is composed by elongated myoepithelial cells [482].

Reid and coworkers reported the development of large human mammary organoids in hydrogels while using a bioprinting platform: the advantage of the 3D bioprint system consists in its capacity to precisely place cells, allowing for optimal control of organoid formation [483].

Several recent studies have reported the development of organoid cultures from tumoral breast tissue. In this context, fundamental was the study of Sachs et al. reporting an efficient protocol for the generation of breast cancer organoids and the establishment of a representative collection of well-characterized breast cancer organoids that are suitable for drug screening [484]. The methodological improvement was largely related to the addition of Neuregulin-1, a growth factor allowing both the efficient generation and long-term maintenance of breast cancer organoids [484]. Breast cancer organoids recapitulated the histological heterogeneity of breast cancers, with ductal cancers usually giving rise to solid organoids and lobular breast cancers frequently generating disco-adhesive organoids [484]. Organoids retained the expression of hormone receptors, including HER2 receptors; the analysis of genetic alterations, including gene mutations, copy number alterations, and gene expression profile, showed that breast cancer organoids displayed the typical heterogeneous genetic landscape of breast cancer and represent all of the breast cancer subtypes [484]. An additional important feature of breast cancer organoids is the possibility to perform functional high-throughput drug screens [484].

Dekkers and coworkers have shown that breast epithelial organoids can be generated from human reduction mammoplasties, thus generating a precious source of cells that can be used to study the clonal evolution of breast cancer [406]. In order to induce the neoplastic transformation of these cells CRISPR/Cas9 for targeted knock-out of four breast cancer-associated tumor suppressors, such as *P53, PTEN, RB1* and *NF1* in mammary progenitor cells was used [485]. Mutant organoids developed a long-term culturing capacity and formed ER^+^ luminal tumors upon transplantation into mice for 1/6 *P53/PTEN/RB1*-mutated and 3/6 *P53/PTEN/RB1/NF1*-mutated lines [485]. These models responded to endocrine therapy and to chemotherapy, and they can be used to explore the events leading to breast cancer development and the mechanisms of response and resistance to treatment [485].

Mazzucchelli et al. have recently reported a simplified protocol to obtain organoids from breast cancer, either from surgical specimens or tumor biopsies [486]. This methodology allowed for a high success rate (87.5%) and the organoids obtained while using this methodology display histological and biomolecular features similar to primary tumors [486].

It is important to note that organoids obtained from breast cancer samples often also contain normal organoids generated from normal mammary tissue present in the breast cancer specimens [487].

In conclusion, the development of organoid cultures has provided a new tool to model and study tumor samples, allowing for the propagation of breast tumors under conditions mimicking the three-dimensional growth of tumor cells [488]. Organoids formed from patient samples represent a precious tool to characterize molecular alterations and to evaluate drug sensitivity [488]. The comparative analysis of 2D and 3D breast cancer cultures provided evidence that 3D organoid cultures containing multicellular spheroids are a better model than 2D cultures for their capacity to simulate some tumor properties *in vivo*, such as hypoxia, dormancy, anti-apoptotic conditions, and drug-resistance mechanisms [489]. Despite these suitable properties for breast cancer translational research, tumor organoids possess some important limitations that are mainly related to the absence of some key components of the in vivo microenvironment and to the in vitro selection of hyperproliferative cells [490].

#### 21.2.3. Microfluidic Devices as a New Tool for the Development of Breast Cancer Models

Spheroid models, as well as organoid culture technology, are unable to fully reproduce the complexity that is observed in 3D tissue architecture of living organs and cannot incorporate mechanical forces, such as fluid shear stress and hydrostatic pressure, which influence cancer cell biology; furthermore, neither of these models is perfused by blood or growth medium flowing through a vascular system coated by endothelial cells and is able to study the recruitment of tumor cells within the tumors [491]. To bypass these important limitations, a new approach was recently proposed for modeling cancer based on the development of microfluidic devices that allows for better reconstituting in vitro the multicellular architecture, the intra-tissutal cellular interactions, and the microenvironment of tumors growing in humans [492]. These microdevices are commonly called organ chips, because they were initially fabricated while using a manufacturing methodology derived from computer microchip fabrication [493].

The technology of microfluidic vesicles was used to generate models of various stages and types of breast cancer. Thus, Choi et al. have adopted a biomimetic bioengineering approach to develop a human disease model that replicates the three-dimensional architecture of DCIS tumors in a tissue-specific microenvironment [494]. This microsystem allows for microfluidic co-culture of DCIS spheroids with normal mammary human ductal epithelial cells in opposition to human mammary fibroblasts embedded in a three-dimensional extracellular matrix scaffold [494]. This model was used as a screening platform for evaluating the efficacy of anticancer drugs [494]. Some microfluidic-based models offer the unique opportunity to develop in vitro models that are suitable for exploring breast cancer migration and invasion. Jeon et al. developed an organ-specific 3D microfluidic model suitable to study human metastatic breast cancer extravasation into a perfusable human microvascularized bone-mimicking or muscle-mimicking microenvironments. This model allows the flow of breast cancer cells and their adherence and extravasation (metastatization) through human microvascular networks [495]. Extravasation rates and microvascular permeabilities were different in the bone-mimicking microenvironment as compared with myoblast-containing matrices [495]. In this system, A3 adenosine receptors modulate breast cancer extravasation [495].

The organotypic microfluidic breast cancer models are particularly suitable for evaluating the role of environment in breast cancer biology and response to drugs. Thus, Ayuso et al. developed a microfluidic model able to mimic the DCIS structure allowing for capturing the multiple cellular metabolic adaptations to endure hypoxia and nutrient starvation within the mammary duct [496].

Microfluidic platforms that are able to generate breast ductal structures have been used to evaluate the effect of fibroblasts in breast cancer drug response [497] and the impact of adipocytes and obesity in hormonal therapy [498].

## 22. Conclusions

Over the past decades, the knowledge gained in the understanding of the molecular and genetic alterations operating in breast cancer has greatly contributed to better understanding the pathogenetic mechanisms operating in this disease and to define consistent heterogeneity and it has provided at the same time a precious and unique tool for the refinement of treatment options.

Perou and Sarlie related a fundamental step in the study of breast cancer to the classification of these tumors into four groups: luminal A, luminal B, basal-like and HER2-enriched [58]. From this classification it was derived the currently adopted subdivision at clinical level of breast cancer in five subtypes that involve luminal A-like, luminal B-like/HER2-negative, luminal B-like/HER2-positive, HER2-enriched, non-luminal, and triple-negative. Molecular studies have provided a detailed typing of the genetic alterations that characterize these different tumor subtypes.

Considerable progress has been made over the past years in the study and treatment of patients with breast cancer, leading to a 40% decrease in mortality from this disease. This progress was made possible through an improvement of prevention strategies (mainly by early diagnosis) and of a treatment of both early and advanced breast cancer, supported by the understanding that breast cancer is not a single disease but several diseases with different subtypes that are driven by different molecular abnormalities.

In this context, a very notable example is given by HER2-positive breast cancers, which are characterized by the amplification of HER2 with overexpression of its protein product. Thus, the targeting of this membrane receptor with the humanized monoclonal antibody trastuzumab decreased the rate of recurrence and death by 80% in patients with early-stage disease and it was effective in patients with metastatic disease. Several other anti-HER2 agents have been introduced into the clinic, including another monoclonal antibody (pertuzumab), several HER2 tyrosine kinase inhibitors (lapatinib, neratinib, and pazopanib) and an antibody-drug conjugate, trastuzumab emtansine [499].

A recent study showed a remarkable benefit in women with stage I to III HER2-positive breast cancer with residual disease after neoadjuvant chemotherapy plus trastuzumab and treated in the postoperative period with trastuzumab-emtansine when compared to those that were treated with trastuzumab [95]. This important study suggests that neoadjuvant chemotherapy treatment with trastuzumab alone or with pertozumab is the standard of care for patients with primary newly diagnosed HER2-positive breast cancer; patients not achieving a pathological complete response with this regimen, are treated with trastuzumab-emtansine, offering a better opportunity for improving long-term outcome [95]. The idea that a trastuzumab-drug conjugate is a potent and effective therapeutic strategy for HER-positive breast cancers is supported by another recent study that was based on trastuzumab deruxtecan, a drug conjugate with trastuzumab, cleavable peptide-based linker, and potent topoisomerase I inhibitor payload [500]. A recent study showed that, in patients with HER2-positive breast cancer with trastuzumab emtansine-resistant disease, 59.5% of patients achieved an objective response and a progression-free survival of 22 months, with most of responses being durable [500]. These results are particularly remarkable when considering that the responding patients have trastuzumab emtansine-resistant disease and they most progressed on pertuzumab [500]. In line with these findings, Modi et al. observed in 112 HER2-positive breast cancer patients that were previously treated with trastuzumab emtansine and then treated with the recommended dose of trastuzumab deruxtecan 61% of responding patients with a median response duration of 14.8 months and median duration of progression-free survival of 16.4 months [501]. Interestingly, trastuzumab deruxtecan induced objective antitumor responses in 37% of patients with HER2-low-expressing advanced breast cancer, with a median duration of response of 10.4 months [502]. In another recent study, Murthy et al. reported the results of the HER2CLIMB trial, which involved 612 patients with HER2-positive metastatic breast cancer previously treated with trastuzumab, pertuzumab, and trastuzumab emtansine and randomly assigned to receive either trastuzumab and capecitabine alone or in association with tucatinib, an oral HER2 tyrosine kinase inhibitor: tucatinib administration improved the progression-free survival from 5.6 months to 7.8 months, with a median duration of overall survival of 21.9 months in the tucatinib-combination group, when compared to 17.4 months in the placebo-combination group [503]. Despite these improvements, the majority of affected patients with advanced HER2-positive breast cancer unfortunately die from disease; therapies to try to overcome treatment resistance are being actively investigated. One of these strategies consists of targeting the cyclin-dependent kinases 4/6 [504].

HER2-targeting therapies have also modified the care of patients with early stage HER2-positive breast cancer. In fact, the standard-of-care therapy for localized HER2-positive breast cancer patients consists in chemotherapy and one year of adjuvant HER2-targeted therapy. However, some patients relapse to this treatment and, thus, do not benefit from HER2-targeting therapy; for these patients, alternative approaches are under investigation trying to move towards a personalized treatment [505]. This strategy is necessary, because, as discussed above, HER2-positive breast cancers are heterogeneous and the distribution of intrinsic subtypes within HER2-positive tumors greatly differs according to ER status: approximately 75% of HER2^+^/ER^-^ tumors and 30% of HER2^+^/ER^+^ pertains to the HER2-enriched subtype, whereas about 70% of HER2^+^/ER^+^ tumors are luminal A/B. HER2-enriched and HER2-high expressing breast cancers are those with high responsiveness to HER2-targeted therapy [88,91]. However, 20–60% of HER2^+^/HER2-E patients do dot display a complete response following anti-HER2 therapies. The study of phenotypic changes occurring in these tumor cells during and after HER2-blocking therapy showed a switch to a low-proliferative luminal A phenotype, more evident in HR^+^ than in HR^-^ tumors, associated with an increased sensitivity to CDK4/6 inhibitors [506].

The consistent progress that was achieved in the treatment of the most ER-positive breast cancer at various stages of disease development has been recently extended to the treatment of advanced and metastatic stage.

For all ER-positive breast cancer several lines of endocrine-based therapy are usually used until the development of endocrine resistance, unless where a visceral crisis or a clear disease progression occurs. For premenopausal patients, ovarian suppression is required, in addition to another endocrine therapy agent, such as tamoxifen (an aromatase inhibitor) or fulvestrant (selective ER degrader); for postmenopausal patients, endocrine therapy is based on an aromatase inhibitor, fulvestrant, or tamoxifen [505]. However, in spite of the international guidelines recommended endocrine-based therapy as a preferred first-line treatment for HR-positive/HER2-negative advanced breast cancer, various studies have reported that 35–60% patients in Europe and North-America received chemotherapy as a first-line therapy, particularly younger patients with visceral disease [507,508,509,510]. A retrospective analysis was carried out by Netherlands investigators in 520 consecutive HR-positive/HER2-negative metastatic breast cancer patients; 482 of these patients received palliative systemic therapy: about 24% of these patients received initial chemotherapy and they were usually younger, with less comorbidity and bone metastases than patients who received initial endocrine therapy [509]. Median progression-free survival of initial chemotherapy was 5.3 months and of initial endocrine therapy 13.3 months, with a median overall survival of 16.1 and 36.9 months, respectively [509]. Thus, apparently, chemotherapy was associated with a worse outcome, when compared to endocrine therapy, in HR-positive advanced breast cancer patients.

A recent clinical trial, the KCSG-BR15-10 trial, compared palbociclib plus exemestane and a gonadotropin-releasing hormone agonist with capecitabine; 50% of these patients received no previous treatment for metastatic breast cancer, 86% patients relapsed while, or just after, tamoxifen treatment, and 49% had visceral disease [511]. Progression-free survival was significantly longer in the palbociclib arm (20.1 months) than in chemotherapy arm (14.4 months) [511].

The optimal endocrine therapy for metastatic ER-positive breast cancer remains to be determined. In this context, recently, the results of a study involving 674 metastatic HR-positive/HER2-negative breast cancer patients randomized to receive either the aromatase inhibitor anastrozole plus selective ER down regulator fulvestrant with anastrozole alone as the first-line therapy: the median overall survival was 49.8 months in the combination-therapy groups and 42.0 months in the anastrozole alone group [512]. Interestingly, the difference between the two groups of patients was more pronounced for patients who did not receive previous tamoxifen treatment (52.2 months vs 40.3 months), whereas it was similar for patients that were previously treated with tamoxifen (48.2 months vs 43.43 months) [512].

Interestingly, a recent clinical trial (TAILORx, Trial Assigning Individualized Options for Treatment) evaluated on early-stage HR-positive/HER2-negative breast cancer patients, the impact of endocrine therapy as compared to chemoendocrine therapy [513]. This study, which involved the enrollment of more than 10,000 patients, showed the noninferiority of endocrine therapy with respect to chemoendocrine therapy concerning patients who had a recurrence score of 11 to 25 (corresponding to the majority of patients) [513]. Some benefit of chemotherapy was found in some women 50 years of age or younger [513]. In a more recent analysis of this study, the same authors showed that the inclusion of clinical-risk stratification criteria provided prognostic information that, when added to the 21-gene recurrence score, helped to identify premenopausal women who could benefit from more effective therapy: hazard ratios for distant recurrence among women in the high clinical-risk category, as based on tumor size and histologic grade, were 2.5–3 times higher than those among women with a low clinical-risk score, independently of whether they received endocrine therapy alone or chemoendocrine therapy [514]. The introduction of clinical-risk criteria helps to better define the differences between the two treatment groups according to risk category: among women who were 50 years of age or younger with an intermediate recurrence score of 11 to 25 and high clinical risk who had received endocrine therapy alone, the rate of distant recurrence at nine years was 12.3%, as compared with 6.1% among women who had received chemoendocrine therapy; for the same comparison, among women with the low clinical-risk group, the risk difference between the two treatment groups of patients was very limited [514]. On the basis of these data, it was suggested that the addition of ovarian suppression and an aromatase inhibitor might reduce the risk of distance recurrence to the levels that were observed with chemoendocrine treatment.

These observations strongly support the need to integrate data that were related to the clinical-risk status and gene expression profiles to obtain a more accurate classification of ER-positive/HER2-negative breast cancer patients according to their risk of disease recurrence. The importance of clinical parameters, such as tumor diameter and nodal status, tumor grade is clearly supported by studies that were carried out on large sets of patients [515]. The various multiple molecular signatures for managing ER-positive breast cancer may provide additional elements to better predict the risk of distant recurrence of these patients [516,517]. The OncoMasTR Molecular Score (OMm), OMclin1, and OncoMasTR Risk Score (OMclin2) prognostic scores were highly prognostic for early and late disease recurrence in women with early-stage ER-positive breast cancer [518].

The integration of clinical variables was shown to represent a simple tool highly predictive for late disease recurrence of ER-positive/HER2-negative patients who have performed five years of endocrine therapy [519]

An approach that was based on the integration of large-scale gene expression data, genomic copy-number alterations and somatic mutations, and epigenetic data provide a superior tool to predict late recurrence events in ER-positive breast cancers [74].

Importantly, recent studies have also better defined the role of radiotherapy in early-stage, low-recurrence risk, HR-positive breast cancers: radiotherapy slightly decreases the recurrence risk in locoregional recurrence event rates but does not appear to decrease the rate of distant recurrences or death [520].

As above discussed in detail, numerous clinical studies have supported the introduction in therapy and the approval of PIK3 inhibitors and CDK 4/6 inhibitors for the treatment of ER-positive/HER2-negative breast cancers after endocrine-based therapy. Ongoing clinical trials are evaluating these agents in early-stage patients. Interestingly, the results of two of these trials provided initial evidence that: a) in the neoadjuvant setting, some patients with high-risk, early stage, hormone receptor-positive, HER2-negative, luminal B-positive, and breast cancer could achieve molecular down-staging of their disease with CDK4/6 inhibitor and endocrine therapy [521]; (b) in a neoadjuvant setting, the addition of the PI3K inhibitor taselisib to endocrine therapy increased the proportion of patients with ER^+^, HER2^-^ early-stage breast cancer who achieved an objective response [312]. These studies rise also absolute need for the identification of appropriate biomarkers, in addition to *PIK3CA* mutations to predict the response to PI3K inhibitors, particularly when used in combination with other drugs [522].

Recent studies have addressed the important issue of the treatment of high-risk early-stage breast cancer patients who receive neo-adjuvant chemotherapy. About one-fourth of patients with early-stage (II/III) who receive neo-adjuvant chemotherapy relapse within five years. An improvement of their response to neoadjuvant chemotherapy is required. The results of the I-SPY2 show that the addition of Pembrolizumab to standard neoadjuvant chemotherapy more than doubled the pathological complete response rates for both HR-positive/HER2-negative and TNBC patients, thus supporting the view that the checkpoint blockade in women with early-stage, high-risk HER2-negative breast cancer might significantly improve patient outcome [523]. Interestingly, a recent meta-analysis that was carried out on a large set of studies provided clear evidence that patients with a pathological complete response after neoadjuvant chemotherapy displayed a significantly better event-free survival and overall survival, particularly for TNBC and HER2-positive breast cancers; importantly, this association was similar among patients who received, or not, subsequent adjuvant chemotherapy [524].

As discussed above, TNBCs have a highly aggressive clinical course, which is associated with a great metastatic potential and poor clinical outcomes due to a high rate of relapse and reduced overall survival. To date, targeted therapies have scarcely contributed to ameliorating the survival in patients with TNBC and chemotherapy remains the standard of care for this tumor subtype. Some patients with early stages of TNBC may be cured with chemotherapy; however, patients developing advanced/metastatic disease have only a low survival probability with the current treatment options. However, in spite these main limitations, some recent studies have suggested some promising therapeutic options for these patients.

Studies regarding the characterization of genetic alterations in TNBCs have shown that this tumor subtype is heterogeneous, both at the level of genetic alterations and of gene expression profile, thus identifying among these tumors some peculiar subgroups characterized by specific genetic features, such as germline BRCA-associated TNBCs, sporadic TNBCs displaying BRCAness (somatic mutations and epigenetic alterations that inactivate BRCA1/2 and other DNA repair genes) [525]. The identification of these tumor subgroups is important, because it implies sensitivity to PARP inhibitors and two of these agents, oloparib and talozoparib, have been approved for TNBC patients with BRCA mutations [525].

The studies on the molecular stratification of TNBCs have led to identifying some subgroups that are associated with a particularly negative prognosis. Thus, starting from the initial observation that combined deletion of TP53 and PTEN genes in mammary epithelium accelerates the formation of TNBC [526], several studies have identified a subgroup of TNBCs pertaining to the BL1 subtype and characterized by TP53 mutations, PTEN loss, low miR-145 expression, RB1 loss, and high MYC and WNT signaling and associated with an exceedingly poor clinical outcome [527,528,529].

Recent data strongly support the view that the immune system plays a critical role for disease outcome in TNBC. Initial studies that are based on the analyses from adjuvant and neoadjuvant clinical trials in TNBC patients have shown that tumor-infiltrating lymphocytes (TIL), studied through evaluation of hematoxylin and eosin-stained tissue sections, are predictive of response to therapy and they are associated with better prognosis [530,531]. Denkert and coworkers have performed a pooled analysis of 3771 breast cancer patients that were treated with neoadjuvant therapy and pertaining to all breast cancer subtypes; an increased TIL level predicted response to neoadjuvant therapy in all molecular subtypes and it was associated with a survival benefit in HER2-positive breast cancer and TNBC [435]. However, increased TILs were an adverse prognostic factor for survival in luminal-HER2-negative breast cancer [532].

Recent studies have explored the prognostic role of TILs in early TNBC. Loi et al. have performed a pooled analysis of a total of 2,148 patients that were enrolled in nine different studies and reached the conclusion that the number of TILs in early TNBC has strong prognostic value, with an excellent survival of patients with high TILs after adjuvant chemotherapy [533]. The good prognostic impact of stromal TILs was confirmed also in early stage I TNBC patients not undergoing adjuvant chemotherapy treatment [534].

TNBCs with high TIL levels have good prognosis and lower tumor mutation burden and neoantigen counts than lymphocyte-poor TNBCs with poor prognosis did [535].

TNBCs with a high number of TILs contain a population of T CD8^+^ T lymphocytes with features of tissue-resident memory T cell differentiation (T_RM_), and these cells express high levels of immune checkpoint molecules and effector proteins [535]. A CD8^+^ T_RM_ gene signature that developed from RNA sequencing data was significantly associated with patient survival in early TNBC [536].

Dieci et al. have explored and assessed the potential prognostic impact of TILs on residual disease after neoadjuvant chemotherapy in TNBC patients, showing that the presence of TILs in residual disease is associated with better prognosis [537]. Interestingly, in some patients, the tumor microenvironment can be modulated by chemotherapy, with the conversion of some tumors from immunologically “cold” to “hot” [537].

Other studies have explored the expression of the Programmed Cell Death 1 (PD1)/Programmed Cell Death Ligand 1 (PD-L1) immune checkpoint pathway in TNBC and other breast cancer subtypes. PD1 is a cell surface receptor, which is activated by its ligand PD-L1 and PD-L2. Activated T lymphocytes induce PD-L1 expression on various immune cells (T lymphocytes, NK lymphocytes and macrophages) and on tumor cells; PD1 activation induces an inhibitory effect on T lymphocyte activation and attenuates the antitumor immune response. Consequently, a PD1 or PD-L1 pharmacologic blocking strategy was recently developed to restore in vivo an immune antitumor response. PD-L1 expression was explored by immunohistochemistry in breast cancer tissue sections, providing evidence that: (i) PD-L1 is expressed in 20–40% of breast cancers; (ii) PD-L1 expression is significantly higher in invasive breast cancer tissue than in normal mammary tissue; (iii) among the various breast cancer subtypes, TNBC clearly expresses PD-L1 more frequently than other subtypes; and, (iv) in TNBCs, PD-L1 tumor expression is positively associated with stromal TIL levels [538].

A study by Keren and coworkers, using a peculiar methodology, Multiplexed Ion Beam Imaging by Time-of-Flight (MIB-TOF), allowed for evaluating both the expression of multiple proteins related to immune function and immune regulation at sub-cellular level simultaneously [539]. The results of this study, which was carried out in 41 TNBC specimens, showed consistent variability at the level of different patients, in the composition of tumor-immune populations, in line with observed overall immune infiltration and the co-occurrence of immune populations and checkpoint expression [539]. Particularly, spatial enrichment analysis showed the existence of mixed and compartmentalized tumors, which corresponded with the expression of PD1, PD-L1, and IDO in a cell-type and location-specific manner; ordered immune structures along the tumor-immune border were associated with compartmentalized tumors and it is linked to a better survival [539]. Hutchinson and coworkers have explored the changes of immune microenvironment in 54 TNBC patients that were treated with chemotherapy and analyzed TIL levels and gene expression profiles in 43 primary-metastasis pairs [540]. The results of this study showed that the levels of TILs decrease from primary to metastatic disease; a change in gene expression pathways was observed in metastatic tumors with a reduction of immune signatures, particularly a reduction in a signature of interferon gamma signaling; a reduction in some of the T cell subsets and cancer-associated fibroblasts and an increase in endothelial cells and macrophages [540]. These observations have some potential important implications, because they suggest that immunotherapy approaches have more chances to provide benefit in early stage TNBC.

Future therapeutic developments in breast cancer will require a further improvement of the individualization of therapy, reaching a tuning of the treatment escalation or de-escalation to the individual tumor biology and to early therapy response. These required developments aim to optimize the therapy choices at the level of individual patients. A major future effort would be devoted to a better understanding of the mechanisms underlying breast cancer resistance and how to overcome it. In this context, a fundamental contribution could derive through an improvement of the detection methods, such as liquid biopsy, allowing for determining the driving pathways of a tumor at a given moment and, thus, to enable the correct determination of the optimal sequence of therapies [541].

Endocrine resistance mechanisms are heterogeneous in breast cancer subtypes and this indicates that the identification of the resistance mechanisms operating at the level of individual patients will be necessary to adapt adequate individual strategies to bypass the chemoresistance mechanisms. A better definition of the molecular factors that define the endocrine response in ER^+^ breast cancer patients and HER2-targering therapy in HER2-positive breast cancer patients will be essential for the development of an improved treatment and for the selection of patients who may be less responsive to these treatments and more prone to the development of chemoresistance and could be shifted to alternative treatments aiming to prevent the generation of drug resistance. An example of the possible future developments was given by a recent study by Rueda and coworkers [74]. This study, through the retrospective analysis of 3240 breast cancer patients, delineated the spatio-temporal patterns of relapse across immunohistochemical, intrinsic (PAM50), and integrative (IntClust) subtypes [74]. This analysis allowed for defining a model predicting individual risk estimates based on tumor features, clinical, pathological and molecular covariates and disease chronology; importantly, this model is able to evaluate the benefits of adjuvant therapy at diagnosis and to assess also how a patient’s risk of recurrence changes throughout follow-up [74]. The integration of integrative subtypes analysis, allowed for defining very important differences in the rates of recurrence among breast cancer subgroups allowing the identification of ER^+^/HER2^-^ patients having a markedly increased risk of late disease relapse up to 20 years after diagnosis (representing about one-quarter of ER^+^ breast cancers) and the prediction of the risk of distant recurrence after five years in ER^+^/HER2^-^ patients; this analysis allowed also the identification of a subgroup of TNBC patients with a better outcome, remaining largely relapse-free after 5 years [74]. This study addresses one of the contemporary challenges in breast cancer therapy, consisting in the identification of a subset of ER^+^ breast cancer patients having a high-risk of recurrence and tumor biomarkers being more accurate in the prediction of recurrence than standard clinical covariates. These observations support future developments of breast cancer treatment.

The characterization of genomic, transcriptomic, proteomic, epigenomic, and microenvironmental alterations has improved our knowledge of TNBC. Taking advantage of the results of these studies, a major effort is currently made to classify TNBCs into distinct clinical and molecular subtypes that could guide treatment decisions, and this will represent a major objective of future studies.

The neoadjuvant approach to therapy has become the standard for HER2-positive and TNBC. The extension of this approach to groups of early-stage ER^+^ breast cancers, as outlined above [521,522,523] is becoming to give some initial responses about the potential benefits deriving from different therapeutic strategies (e.g., endocrine therapy, chemotherapy, chemoendocrine therapy, and tailored therapy) through the analysis of residual disease detection.

Similarly, other future developments will need to improve our understanding of the molecular mechanisms underlying the onset of endocrine resistance. The molecular alterations and the deregulated pathways that are involved in endocrine resistance are only, in part, known. Current studies have, in part, defined the mechanisms of endocrine resistance involving ER mutations, FOXA1 overexpression, and loss of ER expression [542], but a better understanding of these mechanisms will be necessary for the development of treatments strategies aiming to bypass this resistance.

## Figures and Tables

**Figure 1 medsci-08-00018-f001:**
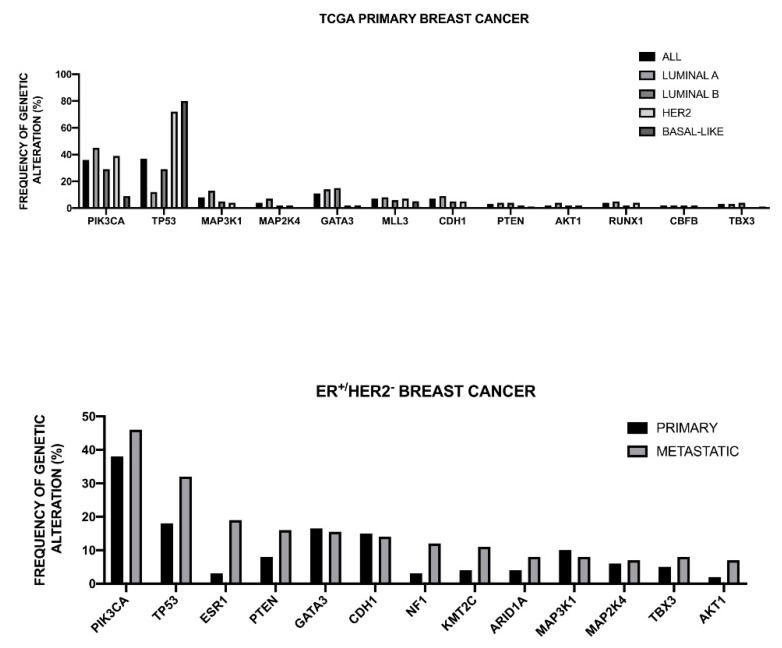
Genomic landscape of recurrent genetic alterations of breast cancer. Top Panel: recurrent genetic alterations in primary breast cancers subdivided according to intrinsic subtype. The data shown in the Figure are reported bt TCGA [46]. Bottom Panel: Most recurrent somatic mutations observed in primary and metastatic ER^+^/HER2^-^ breast cancers. The data shown in the Figure are reported by Angus et al. [47].

**Figure 2 medsci-08-00018-f002:**
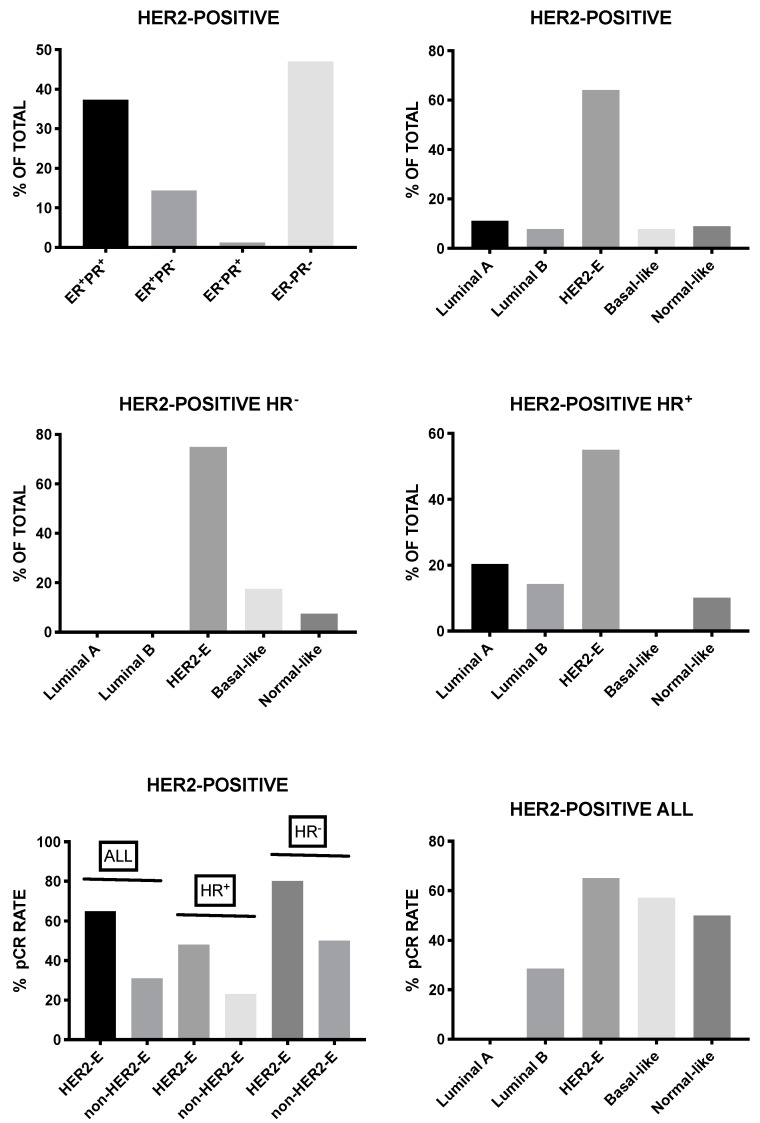
Hormone receptor status (top panel, left) and intrinsic subtype overall (top panel, right) or by hormone receptor status (middle panels) in a population of HER2-positive breast cancer patients undergoing neoadjuvant treatment with trastuzumab plus paclitaxel with or without lapatinib. The bottom panels report the rate of pathological complete response (pCR) observed in all HER2-positive patients (right panel) or in HER2 patients with a HER2-enriched expression subtype, subdivided according to hormonal receptor status (left panel). The data shown in this Figure are reported by Coarey et al. [85].

**Figure 3 medsci-08-00018-f003:**
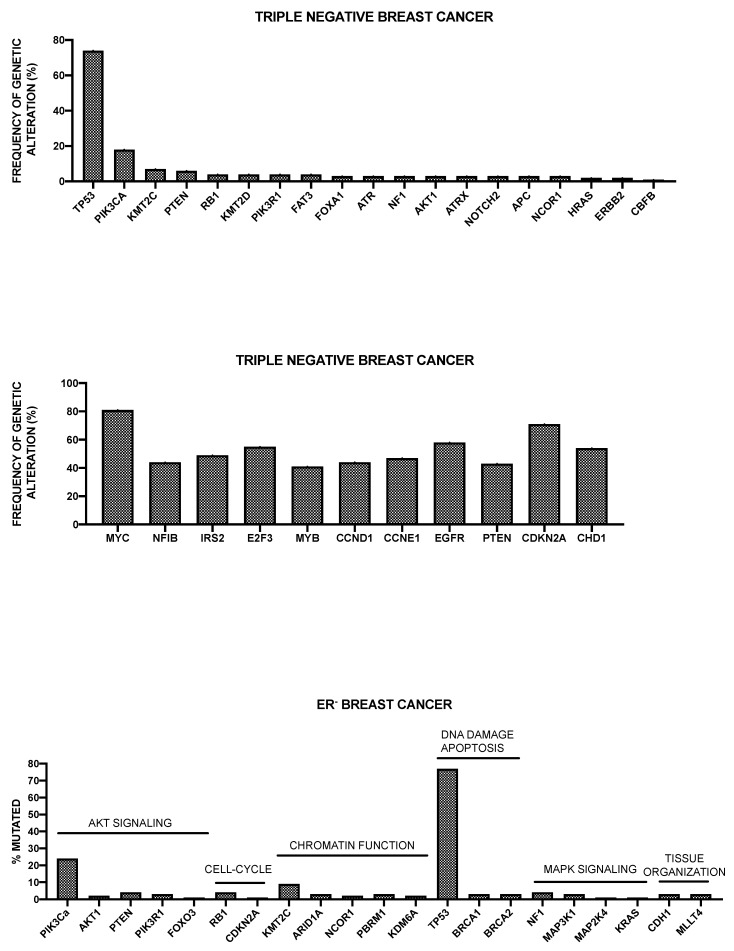
Genetic alterations observed in triple-negative breast cancers (TNBC). Top and middle panels: most recurrent somatic mutations (top panel) and copy number alterations (middle panel) observed in TNBC. The data shown are reported in Jiang et al. 2019 [119]. Bottom panel: most recurrent somatic mutations, subdivided according to functional gene groups, observed in estrogen receptor (ER)-negative breast cancers. The data shown are reported in Pereira et al. 2016 [49].

**Figure 4 medsci-08-00018-f004:**
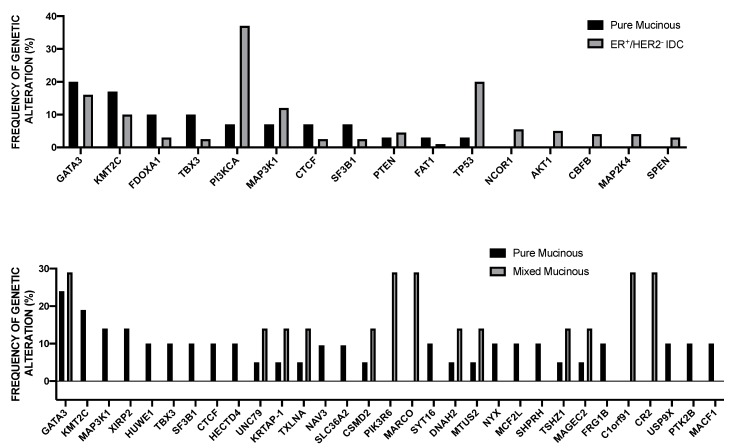
Landscape of somatic genetic alterations observed in mucinous breast cancer. Top panel: comparison of the frequency of most recurrent genetic alterations observed in pure mucinous breast cancers and ER^+^/HER2^-^ invasive ductal breast cancers. Bottom panel: comparison of the most frequent genetic alterations observed in pure mucinous and mixed breast cancers. The data shown in this Figure are reported by Pareja et al. 2019 [214].

**Figure 5 medsci-08-00018-f005:**
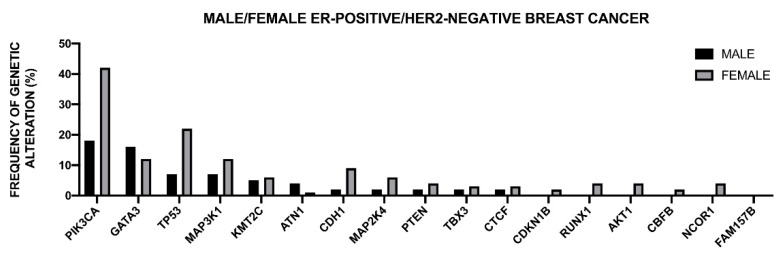
Comparison of the most recurrent somatic mutations reported in male ER^+^/HER2^-^ breast cancers, compared to female ER^+^/HER2^-^ breast cancers. The data shown in this figure are reported by Piscuoglio et al. 2016 [257].

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
