# Peer review of "Breast Cancer: A Molecularly Heterogenous Disease Needing Subtype-Specific Treatments"

_medsci, 2020, doi:10.3390/medsci8010018_

Round 1

Reviewer 1 Report

Due to its high incidence and heterogeneity, the study of target therapies for breast cancer is still a quite relevant and timely topic. Overall, the document herein presented is a broad review, supported by updated/recent literature, covering most (or all) types of breast cancer. However, despite its relevance and systematic approach, several important drawbacks have been identified:

The information is presented in a descriptive form, rather than within an integrated way. Moreover, the manuscript would be much improved if including an overview in each topic. Throughout the text, there is duplication of information (g. information on mutation frequency in the same type of cancer, definition of abbreviations). The review should be more structured. For instance, the paragraphs relative to PDX are demanding to read, due to the “side” information on the different types of disease in which these models are used. This section, should focus on the differences when compared to patient data. The section “Mouse models of breast cancer” includes information of both animal models and human in vitro This should be split in two different sections. Additionally, in such a review one would expect to see the advantages and disadvantages of each model. Importantly, an updated review it would be pertinent to include a section dedicated to microfluidic devices to study breast cancer, and not only 3D cultures.

Finally, the conclusion of this review lists some clinical trials and new therapies. Whereas, there is no overall conclusion or overview on the topic of the review, neither future perspectives.

Author Response

  • The section on animal models was now structured and subdivided into four paragraphs, related to: a) in vivo models, genetically engineered mouse models; b) in vitro models, patient-derived xenografts; c) organoid models; d) microfluidic devices.
  • As above indicated, in the mouse model chapter, a section on microfluidic devices to study breast cancer was now included.
  • The conclusion chapter now includes future perspectives.
  • The English was now revised.

Reviewer 2 Report

This is a well compiled and comprehensive review article. The minor suggestions for improvement are as follows:

The role of various oncogenic signaling cascades involved in breast cancer progression should be discussed. The authors should also highlight pharmacological strategies that are in clinical trials against various types of breast cancer discussed in this article. The authors should provide their own justification and relevance of the study. This will help the readers to understand the importance of the paper. Conclusion section is short and needs elaboration.

Author Response

  • The role of various oncogenic signaling cascades involved in breast cancer progression was now discussed in a dedicated section.
  • Pharmacological strategies that are in clinical trials against various types of breast cancer have been discussed.
  • The main message and justification of this review paper is to provide a wide overview on the recent acquisitions on the molecular pathogenesis of breast cancer and on the development of new therapeutic strategies. These studies strongly support the view that breast cancer is a heterogeneous disease, requiring accurate definition of its molecular identity and dedicated treatments.
  • Conclusion section was now elaborated in more detail.
  • The English was now revised.

Round 2

Reviewer 1 Report

After reading the revised version of the manuscript, some changes were indeed made which improved the quality of the paper, namely a section on microfluidics, 3D cultures and on cell signalling pathways. Nevertheless, some important drawbacks identified in the previous version are still present:

  1. Throughout the text, there is repetition of information (g. information on mutation frequency in the same type of cancer, definition of abbreviations).
  2. The conclusion of this review lists some clinical trials and new therapies. Whereas, there is no conclusion, or overall overview on the topic of the review. Future perspectives are still not clear.